# Probe set selection for targeted spatial transcriptomics

**Louis B. Kuemmerle** [1,2,3,21], **Malte D. Luecken** [1,4,5,21], **Alexandra B. Firsova**[6], **Lisa Barros de Andrade e Sousa**[7], **Lena Straßer**[1], **Ilhem Isra Mekki**[7], **Francesco Campi**[7], **Lukas Heumos**[1,3,8], **Maiia Shulman** [1], **Valentina Beliaeva**[1], **Soroor Hediyeh-Zadeh** [1,3], **Anna C. Schaar**[1,9,10], **Krishnaa T. Mahbubani** [11], **Alexandros Sountoulidis** [6], **Tamás Balassa**[12], **Ferenc Kovacs** [13], **Peter Horvath**[12,14,15], **Marie Piraud**[7], **Ali Ertürk** [2,16,17,18], **Christos Samakovlis**[6,19] & **Fabian J. Theis** [1,3,20] ✉

Targeted spatial transcriptomic methods capture the topology of cell types and states in tissues at single-cell and subcellular resolution by measuring the expression of a predefined set of genes. The selection of an optimal set of probed genes is crucial for capturing the spatial signals present in a tissue. This requires selecting the most informative, yet minimal, set of genes to profile (gene set selection) for which it is possible to build probes (probe design). However, current selections often rely on marker genes, precluding them from detecting continuous spatial signals or new states. We present Spapros, an end-to-end probe set selection pipeline that optimizes both gene set specificity for cell type identification and within-cell type expression variation to resolve spatially distinct populations while considering prior knowledge as well as probe design and expression constraints. We evaluated Spapros and show that it outperforms other selection approaches in both cell type recovery and recovering expression variation beyond cell types. Furthermore, we used Spapros to design a single-cell resolution in situ hybridization on tissues (SCRINSHOT) experiment of adult lung tissue to demonstrate how probes selected with Spapros identify cell types of interest and detect spatial variation even within cell types.

Single-cell transcriptomics has enabled the study of tissue heterogeneity at an unprecedented scale and resolution[1,2]. Recently, spatial transcriptomic technologies have added spatial context to these measurements to describe both tissue composition and organization[3–6]. Yet, this additional information requires a compromise either in spatial resolution or in the number of measured features. While untargeted spatial transcriptomic methods aggregate measurements over multiple cells and thus lack single-cell resolution[7–9], targeted spatial transcriptomic methods measure the expression of a limited number of genes[10–15]. Selecting which genes to target is crucial for successful targeted spatial transcriptomic experiments.

Gene set selection must be guided by analysis goals and the limitations of the experimental technique. Typical analysis goals include the identification of cell types, the description of cell states and transitions and the spatial characterization of cell communication patterns and active multicellular programs. Thus, any selected gene set needs to include cell type marker genes while also capturing general transcriptional heterogeneity beyond cell types. Simultaneously, one must account for technical limitations on the expression levels (for example, due to optical crowding[16,17]) and constraints on probe design and allow users to include prior knowledge such as pre-selected genes that may be relevant to the study of disease.

Selecting gene sets for targeted spatial transcriptomic experiments is a feature selection problem. Typically, expression profiles from dissociated cells are used as a refs. 4,18 (Fig. 1a). A central assumption is that genes that show interesting transcriptional variation in dissociated data will show the same in targeted spatial transcriptomic experiments: cell type markers and highly variable genes (HVGs) from single-cell RNA sequencing (scRNA-seq) data should highlight cell types and genes with interesting spatial patterns. Feature selection applications frequently used in scRNA-seq data analysis include HVG selection, marker gene detection and module detection[19]. Several of these approaches have also been applied to gene panel selection[3,4,6]. Dedicated gene set selection approaches that use a reference scRNA-seq dataset to optimize for cell type classification[20–30] or the capture of transcriptional variation[31–35] have also been proposed. However, few approaches account for both cell type and gene variation, and none of the above methods include technical probe constraints in their selection procedure. As the number of genes that can be selected is limited, probe set selection is a combinatorial problem: an optimal probe set consists of those genes that together optimize multiple objectives simultaneously. Separating gene selection from probe design (Fig. 1b) neglects the combinatorial nature of the problem. Additionally, most available methods are noncombinatorial score-based methods and therefore rather helper tools for laborious manual selections. To tackle the gene set selection problem as a whole, combinatorial selection is required, ideally providing interpretable combinatorial rules for practical downstream analysis.

Here, we present Spapros, a combinatorial probe set selection pipeline that takes into account prior knowledge, technical constraints and probe design while simultaneously optimizing for cell type identification and transcriptional variation. We show that optimizing for these objectives increases the likelihood of finding spatial patterns and cell–cell interaction (CCI)-associated states in spatial measurements. To evaluate Spapros, we developed a suite of evaluation metrics that measure transcriptional variation recovery, cell type identification, redundancy of genes and fulfillment of technical constraints. In our benchmark, Spapros outperforms other methods in both cell type identification and variation recovery. Using Spapros to design a probe set for a SCRINSHOT[13] experiment of human adult lung tissue, we show that Spapros probes identify cell types of interest and detect spatially relevant variation also between cells of the same type. Spapros enables optimal experimental design for targeted spatial transcriptomics and rapid comparison of proposed gene sets through our user-friendly evaluation suite.

## Results

### Quantifying optimal gene set selection

Optimal experimental design can be guided and evaluated by quantifying how suitable candidate gene sets are for downstream analysis of the spatial data. As it is infeasible to perform a new spatial experiment for each proposed gene set, we must find proxies for exploratory analysis success. These proxies should include both the ability to identify known biology (cell type identification) and represent cellular variation that may be found (variation recovery). Following these typical analysis goals, we developed 12 metrics to measure performance in these orthogonal categories as well as gene redundancy, violation of technical expression constraints and run time (Methods and Fig. 1c).

Cell type identification metrics measure cell type classification accuracy and the percentage of captured cell types (Extended Data Fig. 1a) and how well the marker expression of a literature-derived list are captured via marker correlation and cell type-balanced marker correlation (Extended Data Fig. 1b). Variation recovery metrics measure how well cellular variation of the full transcriptome is recovered with only a subset of features. These comprise coarse and fine clustering similarities, which quantify how well the gene set recovers cluster structure at different levels of granularity, and neighborhood similarity, which measures how well the local cell neighborhoods of the gene expression-based $k$-nearest neighbor (knn) graph are preserved (Extended Data Fig. 2a). The amount of redundant genes in the gene set is assessed via gene correlation and the percentage of highly correlated genes (Extended Data Fig. 2b). The low and high expression constraint violation metrics measure how strongly the gene set violates technical expression thresholds. Finally, we measure the computation time of the feature selection methods. The overall performance of a gene set is then computed as the average of the dissociated variation recovery metrics coarse–fine clustering similarity and neighborhood similarity and the cell type identification metrics classification accuracy and percentage of captured cell types, as these are the main objectives we want to optimize for.

As our metrics are mostly run on dissociated single-cell data, we tested how well these translate to spatial variation signals by running existing probe selection methods on a multiplexed error-robust fluorescence in situ hybridization (MERFISH) dataset (Methods). We found that cell type identification and variation recovery metrics translated well to matched spatial data ($r = 0.67$ and $0.68$; Extended Data Fig. 3a). The optimization of fine-grained variation recovery metrics on the dissociated reference was highly correlated with spatial variation recovery in spatial measurements ($r = 0.79$; Extended Data Fig. 3b).

We integrated these metrics and classical gene selection methods into a modular, reproducible Snakemake[36] pipeline for gene set evaluation using our Spapros Python package. The Spapros evaluation pipeline enables large-scale evaluations by automated parallelized high-performance computing usage and can be used to compare gene set selection methods as well as manually selected gene sets.

### Classical feature selection methods optimize different gene set selection objectives

Feature selection approaches are widely used in typical scRNA-seq data analysis pipelines[19]. As such, we investigated whether these approaches are also suitable for gene set selection. We applied our evaluation suite to investigate the performance of several general feature selection methods (based on principal-component analysis (PCA), sparse PCA (SPCA), differential expression (DE) and HVGs; Methods). Additionally, we added random selections and a set of highest-expressed genes as baseline comparisons (Fig. 1c and Extended Data Figs. 1 and 2).

Overall, feature selection methods perform well at particular gene set selection objectives, but no method outperforms others across all metrics. For example, PCA-based feature selection (on unscaled data) clearly outperforms other methods in variation recovery aspects. This difference is most evident when considering finer cellular substructure

---

**Fig. 1 | Probe set selection problem and evaluation of selected gene sets.**
**a**, Schematic of the probe set selection problem. A gene set is selected from scRNA-seq data and used for targeted spatial transcriptomics (ST). The gene set is optimized to identify cell types of interest and to capture cellular variation beyond cell types. **b**, Schematic of the probe design constraint. To measure a specific gene's expression, there must be enough unique probes that can be designed. The unique sequences only occur in at least the expressed isoforms of the targeted gene and not in any RNA of other genes. Sequences that do not have that property are labeled as shared. **c**, Performance comparison for gene sets selected with basic feature selection methods and schematic diagrams of our test suite to evaluate the suitability of selected gene sets for targeted spatial transcriptomic experiments. The test suite includes multiple metrics that are categorized in variation recovery, cell type classification, gene redundancy, computation time and fulfillment of experimental constraints. The aggregated score is the average between variation recovery metrics and the first two cell type classification (classif.) metrics. The red star for DE selected genes indicates that the selection method used cell type annotations in the selection. Acc., accuracy; expr., expression; perc., percent.

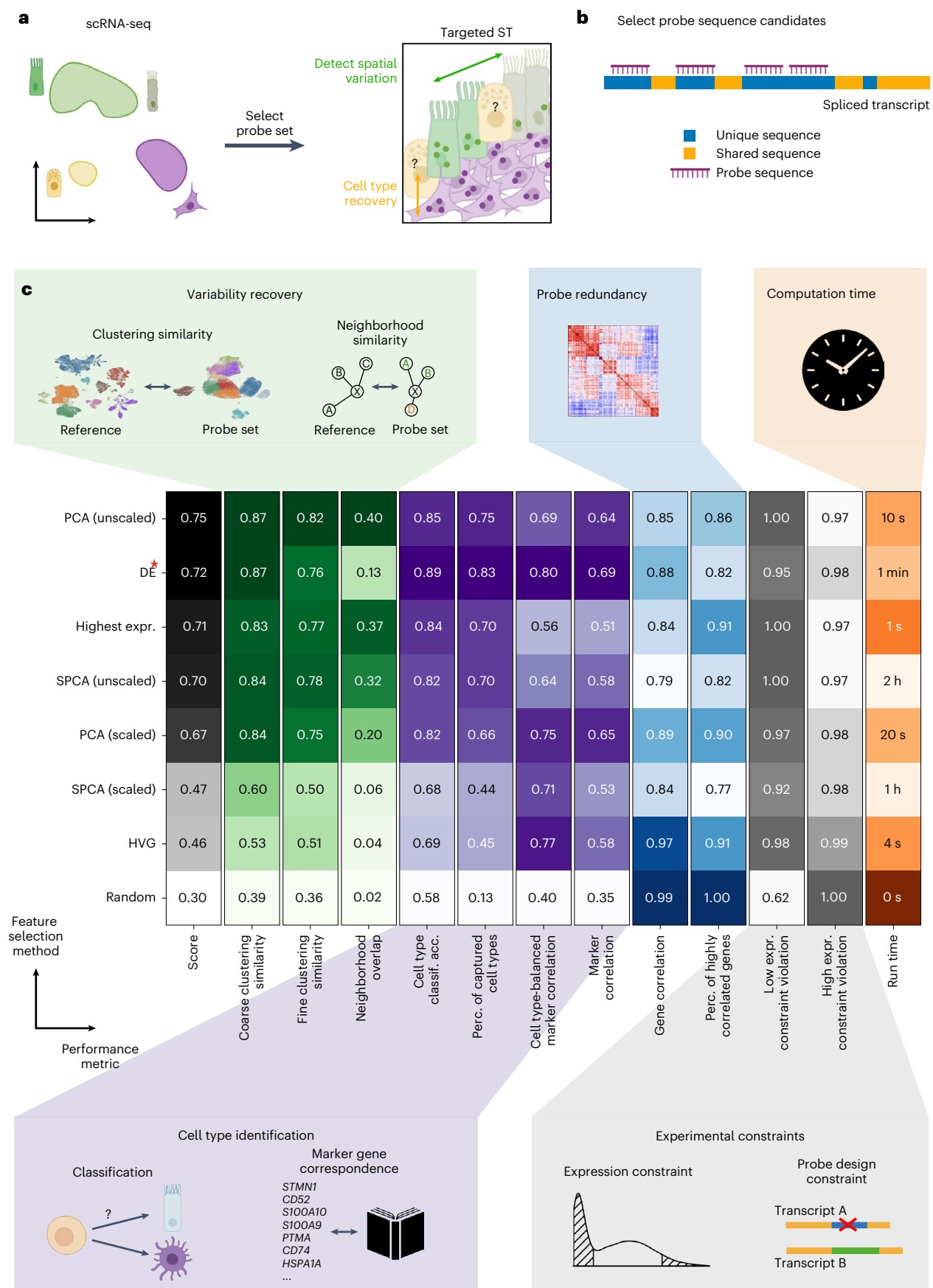

(fine clustering similarity and neighborhood similarity, which measure recovery of cell state variation), which is to be expected given PCA's aim to reconstruct maximal variation. For recovery of coarse effects (coarse clustering similarity), we found similar performance for PCA and DE feature selections. Interestingly, the set of highest-expressed genes ranked second in variation recovery, and PCA-based selection from unscaled data, the top performer in this category, also introduced a natural bias toward highly expressed genes. Considering the performance of highest-expressed genes and PCA-based selection from scaled versus unscaled data, this bias appears to be beneficial for variation recovery in scRNA-seq data (Extended Data Fig. 2a). For cell type identification, differentially expressed genes scored highest among basic feature selection methods. Because PCA-based selection also ranks highly on cell type identification, we observe that optimization for variation recovery and cell type identification can go hand in hand.

As expected, random gene selections exhibited the lowest gene redundancy. Yet, optimizing this metric may not be desirable, as few correlated genes can increase robustness to noise, and medium levels of correlation (-0.3–0.6) between two genes do not preclude that important information is gained by selecting both. By contrast, high levels of correlation (as exhibited by SPCA-based selection; Extended Data Fig. 2b) led to lower information content in the limited probe set. A balance appears to be struck by the best-performing methods (PCA, DE).

Finally, basic feature selection methods do not take into account technical constraints such as gene expression limits due to image saturation or optical crowding and probe design limitations. The expression constraint metrics show that technical constraints are violated by simple feature selection approaches.

Overall, feature selection methods using DE genes or PCA perform well at individual aspects of gene set selection, but no method addresses all objectives at once. Thus, these methods are well suited as components of a larger probe set selection pipeline, which must additionally account for technical constraints.

## End-to-end probe set selection with Spapros

Based on the results of our feature selection benchmark, we built the Spapros pipeline: an end-to-end probe set selection pipeline including PCA-based and DE gene selection as building blocks (Fig. 2a). The Spapros pipeline performs optimized gene selection while designing the probe sequence and accounting for technology-specific technical constraints. These aspects are considered jointly to deliver an optimal combinatorial probe set that can directly be ordered without the need for further gene filtering.

As a first step, Spapros' probe design component is used to filter the full list of possible genes to exclude genes for which probes cannot be designed due to technology-specific technical constraints (Fig. 2b). These constraints include the availability of sufficient unique possible probe sequences as well as sequence properties like GC content and melting temperature requirements. Moreover, binding locations of the final probes for a given gene cannot overlap. Thus, we generate non-overlapping probe sets with optimal thermodynamic and sequence properties with a graph-based search algorithm (Methods). This probe design component supports a range of technologies, including SCRINSHOT[13], MERFISH[15], seqFISH+[37] and HybISS[11], and is easily extensible. Additionally, Spapros' probe design filter can be used

independently of the gene set selection process, making it compatible with other selection methods. After ensuring that all remaining genes represent feasible probe candidates, Spapros selects genes that describe the overall variation in the scRNA-seq reference using a PCA-based selection procedure (Methods) on a pre-selection of HVGs. To ensure that cell types can be recovered using the gene set, Spapros uses the PCA-selected genes to predict cell type labels using a binary classification tree for each cell type (Methods). The genes used in these trees represent candidate cell type marker genes, and the tree itself provides a combinatorial rule, describing how the cell types can be identified in the generated spatial transcriptomics data. To ensure that all user-defined cell types can be identified, Spapros compares the classification performance for each cell type to the performance of reference trees. These trees are generated via a custom approach that iteratively optimizes for classifying similar cell identities. In each iteration, Spapros performs DE selections on critical cell type subsets and retrains the trees on the extended gene pool (Methods). If any discrepancy in performance is found with the DE trees (that represent the optimal performance target), Spapros iteratively adds DE genes to the list of possible genes to improve classification performance. Finally, genes are ranked based on their feature importance in classification trees to allow for a user-defined number of selected genes. To account for technical constraints of expression levels, a smoothed multiplicative penalty kernel is applied to the scores of PCA- and DE-based selections (Supplementary Fig. 1). Based on the final gene set and the non-overlapping probe sequences, our pipeline designs the final probe and detection sequences that need to be used for the experiment (Methods). To facilitate downstream analysis in studies that solely focus on detecting cell type frequencies, it may be of interest to select only genes for cell type recovery rather than detecting additional spatial signals. For this, we provide SpaprosCTo, which exclusively uses DE trees for selection. This contrasts with the standard approach, in which the majority of selected genes originate from PCA-based selection (Supplementary Fig. 2).

While Spapros can select and design probe sets using only a reference scRNA-seq dataset and a list of cell types as input, users can also add prior knowledge and constraints to bias the algorithm toward user-defined genes. This allows users to add particular genes of interest (for example, to test particular hypotheses or capture disease effects) and account for technological constraints (for example, in situ sequencing has limitations on spot detection of highly expressed genes due to optical crowding). This prior knowledge can be incorporated in two ways: (1) as a pre-selection of probe genes, leading to other genes being combinatorially selected around them and (2) as a marker list from which genes are added when respective cell types cannot be adequately classified (Methods).

Overall, Spapros is a flexible, modular gene set selection and probe design tool that selects genes optimized for cell type recovery and cellular variation while enabling users to customize the selection for any experimental design scenario.

## Spapros optimizes multiple probe set objectives simultaneously

We designed Spapros to optimize both cell type identification and recovery of variation beyond cell type annotations. However, by design,

**Fig. 2 | The Spapros probe set selection pipeline. a**, Schematic diagram of the probe set selection pipeline. **b**, Schematic of the transcriptome-wide probe design pipeline. Genes for which not enough probes can be designed are filtered out before gene set selection (first step in **a**). For the selected gene set, technology-specific ready-to-order probes are designed (final step in **a**) (created with https://www.biorender.com). **c**, UMAP comparison of probe sets selected with Spapros for 50 and 150 genes and a reference of 8,000 HVGs for the Madissoon2020 human lung dataset. **d**, Dot plot of probes selected on the lung dataset. Genes are ordered by the Spapros ranking system based on feature importance (Methods). For each cell type, the genes that are important for cell type classification based on the forest classification step are highlighted (Spapros marker). A minimum number of markers per cell type (DE or literature (lit.) gene) defined by the user is selected. For cell types not found in the dataset, genes from a curated marker list are added. KIAA0101 refers to the PCNA clamp associated factor (PCLAF). **e**, Difference of cell type classification confusion matrices between gene sets of Spapros and DE selections. AT1, type I alveolar cell; AT2, type II alveolar cell; DC1, type 1 dendritic cell; DC2, type 2 dendritic cell; NK, natural killer cell; T CD4, CD4+ T cell; T CD8 Cyt, cytotoxic CD8+ T cell.

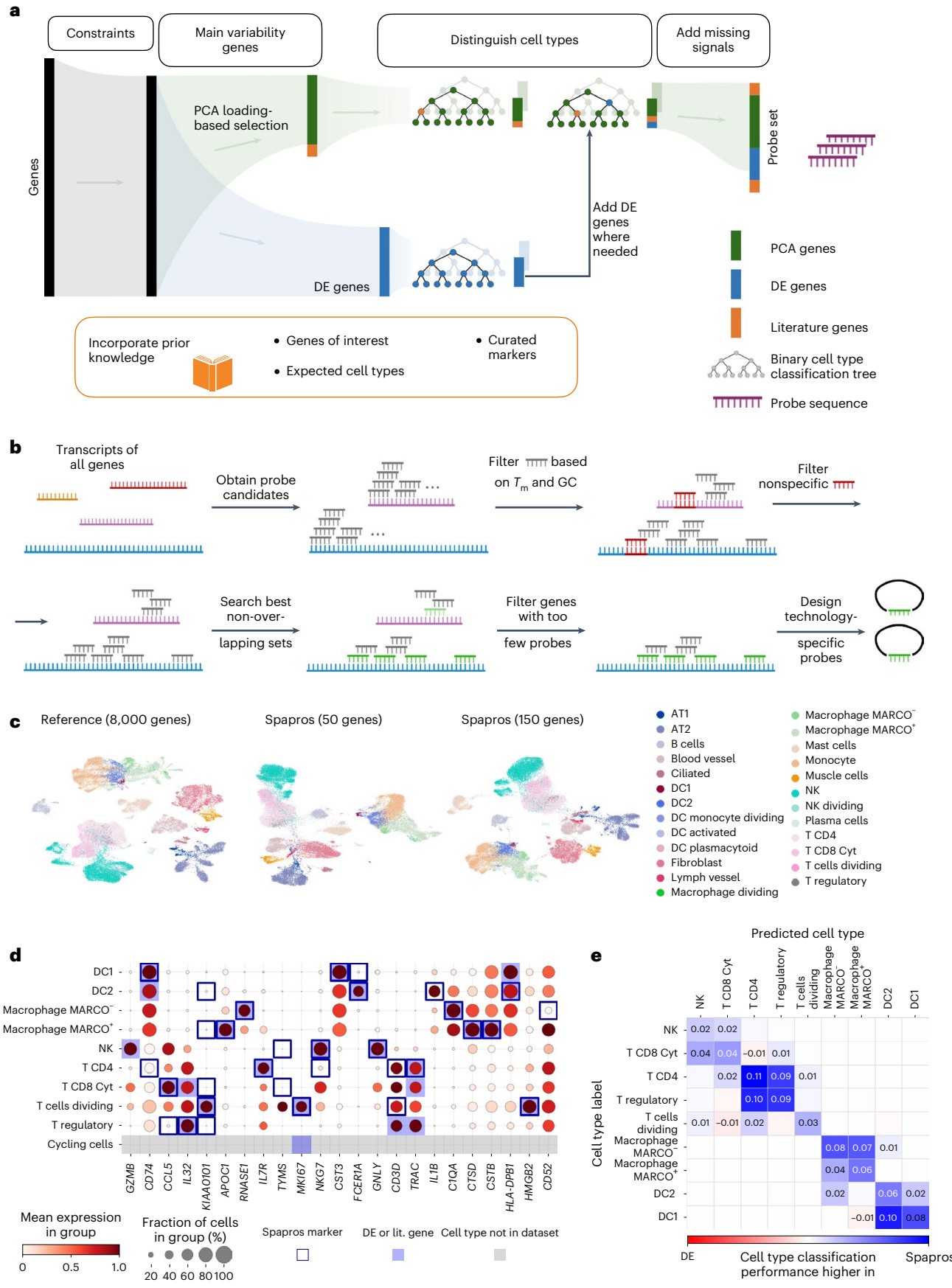

Spapros' first priority is the identification of cell types, while more fine-grained variation is only captured if cell type identification is not affected. We therefore expect that, for low numbers of genes, Spapros mainly captures cell type-level variation and acquires more capacity for fine-grained variation with increasing numbers of genes.

Visually comparing the full gene set to Spapros selections on a uniform manifold approximation and projection (UMAP) embedding showed that cell type variation was strongly conserved using both 50 and 150 genes (Fig. 2c). When comparing cell type classification characteristics between Spapros and DE genes for 50 genes, we observed that similar cell types like type 1 and 2 dendritic cells could be better distinguished by Spapros due to the combinatorial selection of, for example, *CST3*, *FCER1A* and *IL1B* (Fig. 2d,e, Supplementary Fig. 3 and Extended Data Fig. 4). Spapros consistently outperformed the top-performing classical feature selection methods DE and PCA on the prioritized objective of cell type identification and optimized variation recovery to the level of PCA-based selections when increasing the number of selected genes (Extended Data Fig. 4). Thus, the Spapros probe set is optimized for the most relevant signals for any given number of selected genes.

## Spapros selection performs robustly across datasets

When designing a targeted spatial experiment, data generators often have matching scRNA-seq data available from a matching sample. Yet, when this is not the case, the question arises of how similar the transcriptomic reference must be to the spatial sample. Using our evaluation metrics, we can address this question from a computational perspective. We assessed the cross-dataset performance of gene sets selected on three different lung datasets (Extended Data Fig. 5a). Selections are not robust if the cell type classification and variation recovery performance show high variance across datasets. To estimate whether the variance across datasets was high or low, we added selections on each individual donor sample as baseline comparisons.

We observed that the performance variance across datasets and across samples within each dataset was similar (Extended Data Fig. 5b). Thus, selections from one dataset show robust performance in other similar datasets. This is especially pronounced for cell type classification, indicating that there is no cell type identification performance drop for selections on an external dataset from the same tissue. As expected, selection from the full dataset is beneficial for cell type classification compared to selections from individual donors. We also find this trend for variation recovery. Overall, Spapros shows robust performance across different choices of the matched scRNA-seq reference for gene set selection.

## Spapros probe sets identify cell types and spatial variation within cell types in the adult human lung

To show that Spapros' capabilities translate to spatial measurements, we designed and performed a targeted spatial experiment using SCRINSHOT[13] with a 64-gene probe panel generated by Spapros on healthy human lung samples using the Meyer2022 (ref. 38) scRNA-seq reference (Methods). For each cell type, Spapros provides a decision tree that includes the most important genes to robustly identify the given cell type. Leveraging these rules, we detected all targeted cell types in an intralobar section (Fig. 3a,b, Supplementary Fig. 4 and the Methods). Overall, the expression profiles of the cell type clusters matched those of the scRNA-seq reference clusters (Fig. 3a), and the spatial distribution of cell types corresponded to known cellular structures in the lower airways and alveolar space (Fig. 3b). Notably, despite using a relatively small gene panel, we achieved robust cell type identification in our SCRINSHOT experiment. This contrasted with observations from older high-plex spatial technologies, in which a higher number of genes was required to reach similar cell type classification robustness as in dissociated reference data (Supplementary Fig. 5). In such cases, selecting genes based on binarized counts can increase cell type classification performance (Supplementary Note 1 and Supplementary Fig. 6).

To capture variation beyond cell type annotations, Spapros selects genes that exhibit gradients in multiple cell types, as observed, for example, for *FOS* in our panel (Fig. 3c). Although *FOS* is typically associated with dissociation-induced cell stress[39], we tested whether it also exhibits these gradients in non-dissociated tissue. Indeed, we observed such a gradient in airway epithelial cells. Here, *FOS* expression in tracheal basal cells displayed intra-cell type spatial variation. While the basal cell markers *KRT15* and *S100A2* marked the inner (basal) and outer (suprabasal) epithelial layers, respectively (Fig. 3c–e), *FOS* exhibited upregulated and downregulated regions along the epithelium orthogonal to the interior-to-exterior epithelial variation of *KRT15* and *S100A2* (Fig. 3d,e). Investigating FOS variation along the airway epithelium with immunofluorescence (IF) staining validated this finding (Extended Data Fig. 6a). Furthermore, multiple other genes selected by Spapros exhibited spatial intra-cell type variation, such as anti-correlation between *IGFBP7* (associated with larger vessels) and *RGCC* (capillary marker) in endothelial cells as well as between *APOE* and *IFITM1* in macrophages (Extended Data Fig. 6b).

While further experiments are required to interpret these intra-cell type signals, the signals themselves indicate that probes selected by Spapros detect spatial variation of gene expression also beyond differences in broad cellular composition. Thus, Spapros enables identification of cell types of interest while also detecting spatially patterned intra-cell type variation in spatial measurements.

## Spapros outperforms curated gene sets and state-of-the-art methods

We assessed Spapros performance against ten recently proposed gene selection methods, two popular approaches for feature selection (DE and PCA), a curated gene list of airway cell type markers (Methods) and a published probe list for the human heart in a large-scale benchmark using our evaluation pipeline. All methods were run on lung and heart datasets (Methods) to generate both a small (50 genes) and a large (150 genes) gene set (Fig. 4a and Extended Data Fig. 4).

Consistent with the method designs, the results showed that methods could be grouped into categories with differing goals: general variation recovery (SelfE, SCMER, geneBasis, PCA, scPNMF, triku, unsupervised PERSIST (PERSISTus)), cell type identification or selection of cell type-specific markers (spaprosCTo, SMaSH, NS-Forest, ASFS, scGeneFit, COSG, DE, PERSIST) and both of these objectives (Spapros). We found that Spapros was consistently the top-performing method, while the ranking of other methods varied among the different datasets and panel sizes (Fig. 4a,b,d and Extended Data Figs. 4, 7 and 8a).

To provide context for the observed differences in metric scores, we assessed the statistical significance of performance differences between top performers from this benchmark using a bootstrapping approach across 12 datasets (Fig. 4c, Extended Data Fig. 8b and the Methods). We found that Spapros showed significantly higher performance on our aggregated score for 50 and 150 genes than that of all other methods (except for $n = 50$, no significant difference with geneBasis). SpaprosCTo performed best in cell type classification, followed by Spapros and NS-Forest (Extended Data Fig. 8b), especially for low numbers of genes (Fig. 4a and Extended Data Figs. 4 and 8). By contrast, as expected, the variation recovery methods (SelfE, SCMER, PCA, geneBasis) achieved the highest variation recovery metric scores (Extended Data Fig. 8b). However, this did not translate to a performance significantly different from that of Spapros. In sum, when investigating pareto-optimal method performance in terms of cell type classification and variation recovery, Spapros has a unique position on the pareto front (Fig. 4d and Extended Data Fig. 7). By contrast, cell type identification methods do not optimize for variation recovery and variation recovery methods do not reach the cell type classification performance of Spapros.

Spatial transcriptomics is typically used to investigate cellular localization and interactions in tissue. To assess whether the

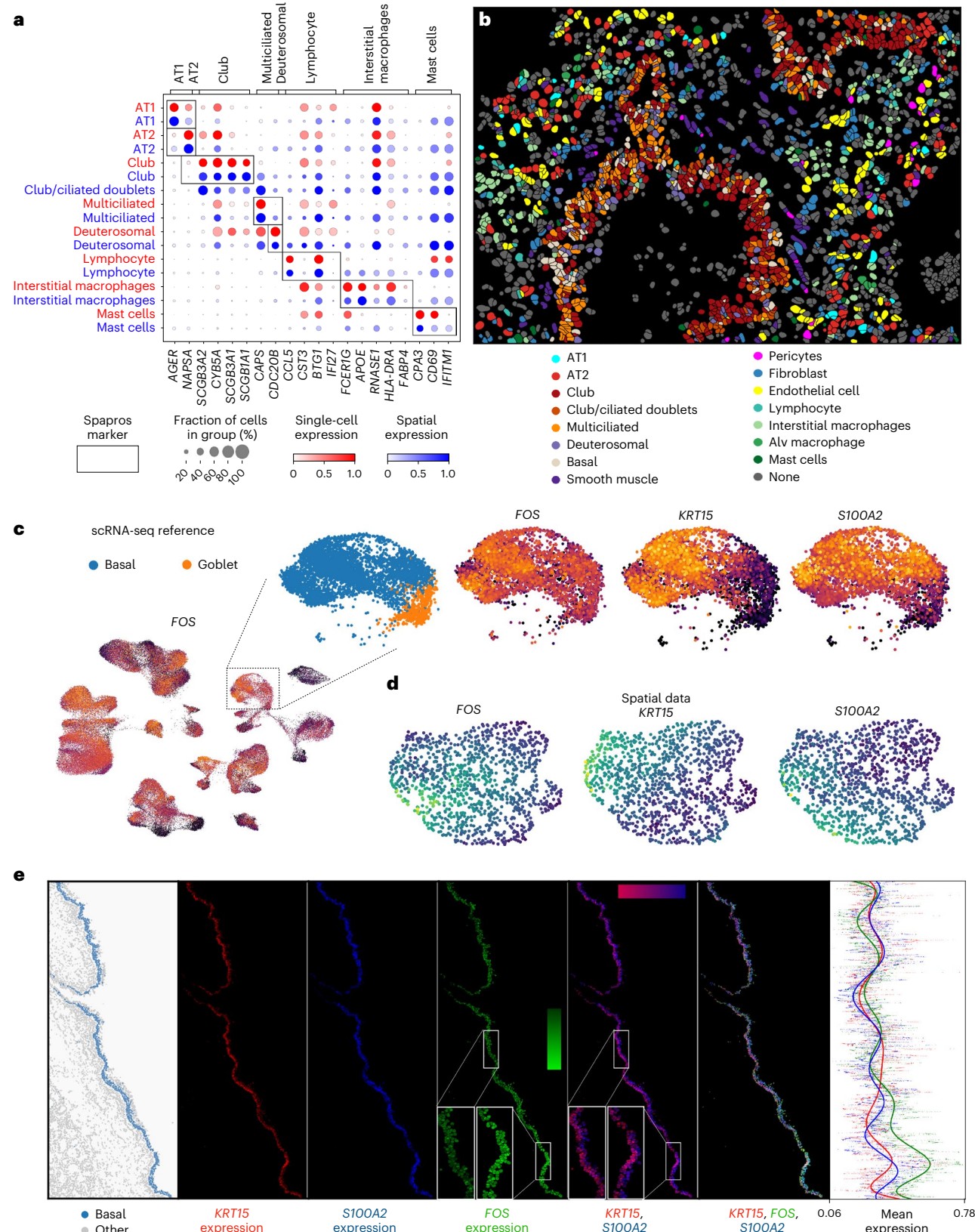

**Fig. 3 | Spapros probe sets identify cell types and spatial variation within cell types.** Spatial lung data measured with SCRINSHOT technology for a probe set selected with Spapros. **a**, Mean expression in spatial cell types in an intralobar lung sample (blue) and in cell types from the single-cell reference (red). The shown genes are identified as the most important genes for cell type identification in the Spapros selection. **b**, Annotated cell types in the intralobar lung sample. **c**–**e**, Spatial distribution of two orthogonal variation axes within tracheal basal cells. Alv, alveolar. **c**, *FOS* expression in the UMAP of the scRNA-seq reference dataset and expression of *FOS*, *KRT15* and *S100A2* in the magnified basal and goblet subset. **d**,**e**, UMAP (**d**) and spatial distribution (**e**) of *FOS*, *KRT15* and *S100A2* in basal cells in a tracheal lung sample.

performance differences in our benchmark translate to improved utility of Spapros to address biological questions, we investigated how well cellular interactions are recovered in different probe sets. Specifically, we compared the gene set selection methods on recovery of cell state variation associated with CCIs on a matched single-nucleus RNA sequencing (snRNA-seq) and MERFISH brain dataset (Methods). Overall, we found a high correlation between variation recovery and CCI recovery ($r = 0.85$), with Spapros among the top performers (Fig. 4e). Moreover, Spapros gene sets captured cell type interaction pairs in an unbiased fashion: we detected interactions between all cell type lineages. Furthermore, CCIs detected by Spapros were also able to distinguish between similar cell subtypes. We found distinct CCIs for similar excitatory neuronal cell identities with a third cell type (for example, SST[+] inhibitory neurons (iSST) or layer 4/5 intratelencephalic excitatory neurons (eL45IT); interactions iSST–eL6ITCAR3 versus iSST–eL5IT and eL45IT–eL23IT versus eL45IT–eL6ITCAR3; with CAR3+ layer 6 (eL6ITCAR3), layer 5 (eL5IT) and layer 2/3 (eL23IT) intratelencephalic excitatory neurons; Extended Data Fig. 9).

As mentioned above, experimental design for targeted spatial transcriptomics involves both selecting optimal gene sets and designing probes for these gene sets. Yet, aside from Spapros, no other method considers the technical probe design constraints in the selection, leading to a significant number of genes lacking adequate probes (Extended Data Fig. 10a). Removing genes for which probes cannot be designed from the selected probe pool for each method significantly reduces the performance of these methods (Fig. 4f) and again requires manual adaptation of the probe set, leading to non-optimized experimental designs. Indeed, when comparing Spapros with manual feature selection based on literature markers, Spapros outperformed the manual list for both heart[3] and lung[38] scenarios. The method surpasses manually selected probe sets and curated marker lists on cell type identification and variation recovery (Supplementary Fig. 7).

Finally, we compared Spapros' scalability to classical gene set selection methods (Fig. 4a and Extended Data Fig. 4). Spapros showed computation times comparable to those of other fast methods with high performances on our evaluation scores (Fig. 4 and Extended Data Fig. 4). Importantly, in contrast to methods like SelfE, ASFS and geneBasis, it also maintained constant time consumption for varying gene set size. Instead, Spapros scaled linearly with the number of cell types in both memory and time (Extended Data Fig. 10b–d). Thus, Spapros can be run locally, especially for selections on coarse cell type annotations. Overall, we find that Spapros outperforms classical gene set selection approaches for targeted spatial transcriptomic experimental design across evaluation criteria. It does so by uniquely optimizing simultaneously for cell type identification and variation recovery, being the only method that considers probe design constraints and providing a scalable user-friendly code base (Supplementary Table 7).

## Discussion

We present Spapros, a probe set selection and design pipeline for experimental design of targeted spatial transcriptomics experiments. Spapros optimizes probe set selection in a combinatorial fashion by optimizing both design and selection of genes simultaneously while taking into account prior knowledge and technical constraints. With these features, our method enables end-to-end probe set selection. It is also a method that optimizes simultaneously for both identification of cell types and recovery of transcriptomic variation beyond cell types.

Due to a reduced number of measured features and processing challenges, annotating cell types in targeted spatial transcriptomics data is typically more challenging than in scRNA-seq data. Consequently, current spatial transcriptomic studies and probe selection methods predominantly focus on cell type identification. Spapros facilitates this by not only optimizing probe sets for cell type identification but also by providing annotation rules based on the decision trees used during probe selection. Additionally, Spapros highlights probes that are candidates for capturing spatially variable patterns across cell identities by scoring them on variation recovery and their importance for cell type separation. Capturing intra-cell type variation is valuable because meaningful spatial signals extend beyond differing cell type compositions, for example, indicating local tissue niches, viral spread, inflammation and more. However, including such more continuous transcripts may reduce separation of cell types when clustering cells. Thus, such genes introduce an additional challenge for downstream analysis and may be excluded for cell type clustering. Spapros' rule-based annotation output and labeling of gene characteristics enable a quick reference-based spatial mapping of cell types and identification of new spatial patterns of cell state continuums. As spatial transcriptomic protocols and analysis methods continue to improve, these spatial state continuums will become increasingly of interest.

A central assumption in the current implementation of Spapros and probe set selection in general is that the transcriptomic signal measured by scRNA-seq is representative of the signal measured in targeted spatial transcriptomics. As shown in our evaluations on spatial data, we are able to use dissociated single-cell reference signals to find cell types of interest with high recall, find matching orthogonal expression variation within these cell types and detect relevant CCI variation. However, the equality assumption between spatial and scRNA-seq data is currently not strictly met, and we find discrepancies between the modalities. Gene distributions are notably different, and some genes are uncorrelated with the signal expected from scRNA-seq, possibly due to nonfunctional probes or individual tissue section anomalies. These challenges are particularly evident in data from older high-plex spatial technologies, in which limitations in data quality and probe performance can hinder robust cell type classification. This has been reported to improve with newer spatial technologies such as Xenium[40,41]. We also find that spatial robustness constraints like PERSIST's binarization of counts can enhance performance on such spatial data (Supplementary Note 1). When anticipating lower data quality, prioritizing cell type identification over variation recovery during gene selection (as implemented in SpaprosCTo) can enhance the experimental design's robustness. There are multiple reasons that lead to discrepancies between the modalities: different RNA measurement techniques lead to the exclusion of reads from highly similar paralogs that are mapped to multiple regions of the genome in droplet-based scRNA-seq, different sample processing between scRNA-seq and targeted spatial transcriptomics leads to the enrichment or exclusion of certain cell types and states, and technology-specific effects like optical crowding can result in imprecise measurements. Quantifying these

---

**Fig. 4 | Spapros outperforms classical selection strategies and state-of-the-art methods. a**, Table showing mean performances of Spapros and other methods, based on 20 bootstrap samples for selecting 50 genes from the Madissoon2020 lung dataset. Methods that use cell type information are annotated with a red star. **b**, P values from two-sided t-tests comparing the aggregated scores of these methods on the bootstrap samples in **a**. Methods are ranked by mean performance. **c**, Two-sided paired t-test P values for the mean aggregated scores across 12 datasets on 50-gene selections. **d**, Pareto front showing the tradeoff between variation recovery and cell type classification scores for 50-gene selections from the Madissoon2020 lung data. **e**, Correlation between variation recovery scores on dissociated data and CCI recovery on spatial data, using matched snRNA-seq and MERFISH data from the human brain. Data are presented as mean values ± s.d. over selections on seven bootstrap samples of the snRNA-seq reference for selecting 50 genes. **f**, Performance benefit of probe design constraint: comparison of the aggregated scores for different methods after excluding genes failing probe design criteria, using 50-gene selections from 20 bootstrap samples of the Madissoon2020 data. Center line, median; box limits, upper and lower quartiles; whiskers, 1.5× interquartile range; points, outliers.

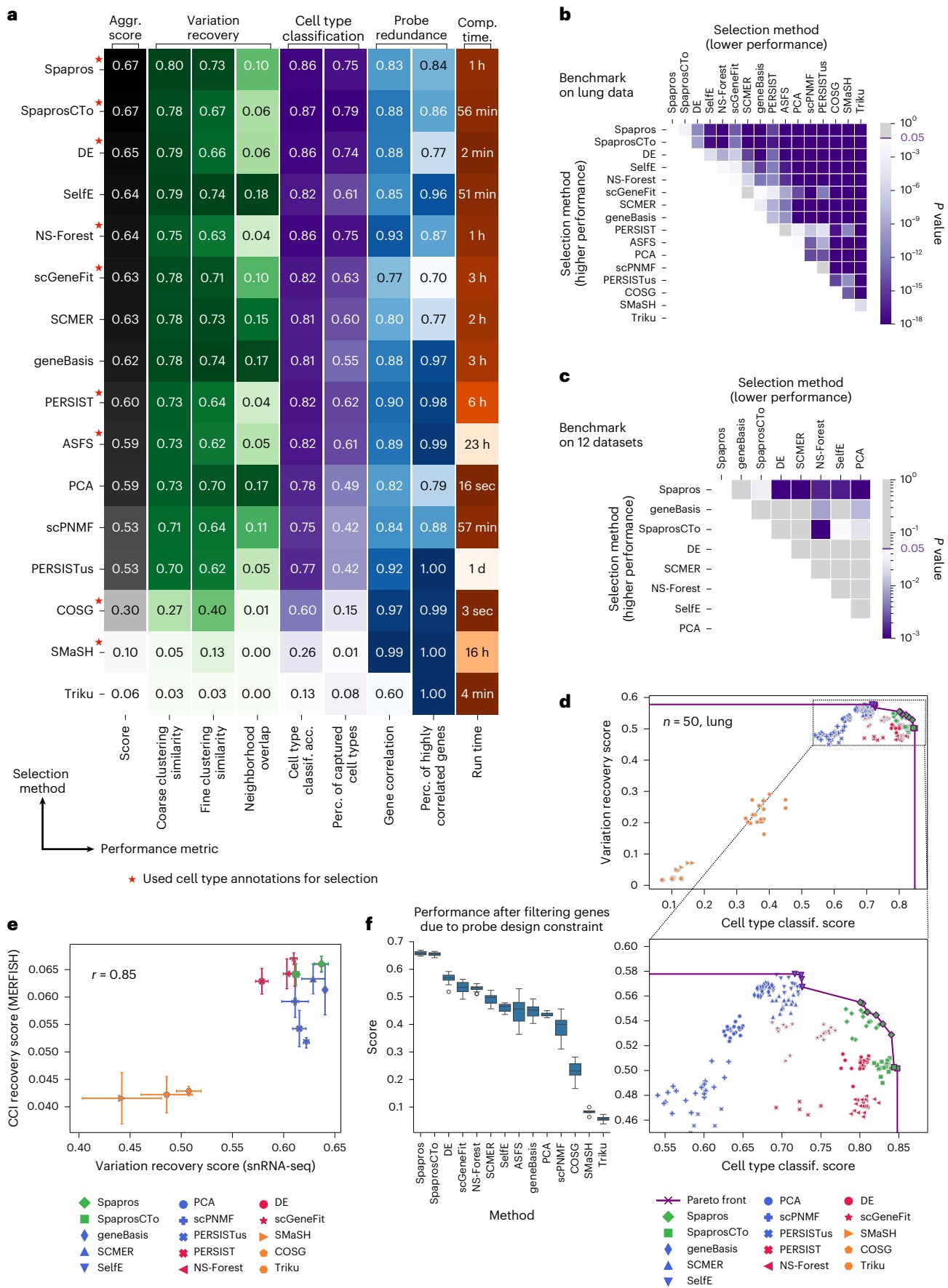

differences and constructing additional robustness constraints are important future directions for probe set selection and reference-based downstream analysis. These robustness constraints also depend on the processing of targeted spatial transcriptomics data for which there are currently no best practices. Normalization, further preprocessing and cell segmentation including the disentanglement of overlapping cells will affect how well these modalities match. To design ideal robustness constraints, future studies need to investigate these effects as well as tissue parameters like organ type, cell density and tissue quality.

While Spapros is a flexible pipeline that can be tuned to emphasize subtypes of a specific cell type, it inherently focuses on capturing the major sources of variation within a dataset. However, the strength of these signals does not always correspond to their relevance to a given scientific question. This can be particularly true in disease studies, in which subtle transcriptional differences may be of greater interest than global sources of variation. In these cases, incorporating prior knowledge through Spapros' pre-selection feature can be especially beneficial. This approach enables users to specify genes to focus on the exploration of disease mechanisms and the identification of relevant spatial cell niches and their associated CCIs. These different application modes highlight how Spapros can be used flexibly to maximize the chances of uncovering biologically important signals that are directly relevant to the scientific question being investigated.

Further improvements to probe set robustness can be derived from integrated reference atlases. As more scRNA-seq datasets are becoming available, these datasets are being integrated into comprehensive reference atlases (Sikkema et al.[42]) that contain consensus signatures of rare cell types and subtle state differences learned across studies. Yet, the large cell number also poses additional methodological challenges such as batch effects and unbalanced label distributions. Spapros tackles unbalanced label distributions by balanced sampling strategies over cell type labels to robustly capture rare cells. Yet, strong batch effects are still a challenge for Spapros and other probe selection methods. Especially selection methods that optimize for variation recovery will also select genes aligned to batch effect variation. While we showed that Spapros can select probe sets and project these across datasets, we did find that also dataset-specific variation (potentially due to batch effects) was captured. Users who choose to select probes using reference atlases that contain robust consensus cell type annotations such as the HLCA[42] may wish to consider testing whether the selected probes are consistent across batches. The optimization for biological variation disentangled from technical variation is therefore an interesting direction for future work. Further extensions to Spapros include experimental design for other modalities like spatial proteomic measurements (for example, CODEX[43]) and extended probe design schemes of combinatorial gene probes that target multiple genes contributing to the same cellular program.

With Spapros, we introduced new concepts for optimally selecting probe sets in targeted spatial transcriptomics: our approach combines gene set selection and probe design to enable combinatorial selection and optimizes simultaneously for the dual objective of cell type identification and recovery of transcriptomic variation. Spapros will thus enable optimal experimental design while guiding downstream analysis. Our gene set selection approach is broadly applicable to all imaging-based spatial transcriptomic methods that use gene subsets, with a pronounced advantage for those with smaller gene panels. Additionally, our evaluation suite sets a reproducible and robust standard for quality assessment of spatial probe sets and can be readily extended toward additional metrics. Spapros is available as a Python package enabling easy and flexible probe set selection, evaluation of probe sets and probe design. With Spapros, we aim to enable users to maximize their success in future exploratory spatial studies to find new spatial cellular variation.

## Online content

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

[1]Institute of Computational Biology, Helmholtz Zentrum München, Neuherberg, Germany. [2]Institute for Tissue Engineering and Regenerative Medicine, Helmholtz Zentrum München, Neuherberg, Germany. [3]School of Life Sciences Weihenstephan, Technical University of Munich, Freising, Germany. [4]Institute of Lung Health & Immunity, Helmholtz Munich, Member of the German Center for Lung Research (DZL), Munich, Germany. [5]German Center for Lung Research (DZL), Gießen, Germany. [6]SciLifeLab and Department of Molecular Biosciences, Stockholm University, Stockholm, Sweden. [7]Helmholtz AI, Helmholtz Zentrum München, Neuherberg, Germany. [8]Institute of Lung Biology and Disease and Comprehensive Pneumology Center, Helmholtz Zentrum München, German Center for Lung Research (DZL), Munich, Germany. [9]TUM School of Computation, Information and Technology, Technical University of Munich, Munich, Germany. [10]Munich Center for Machine Learning, Technical University of Munich, Munich, Germany. [11]Department of Surgery, University of Cambridge and Cambridge NIHR Biomedical Research Centre, Cambridge, UK. [12]Synthetic and Systems Biology Unit, Biological Research Centre, Eötvös Loránd Research Network, Szeged, Hungary. [13]Single-Cell Technologies Ltd, Szeged, Hungary. [14]Institute of AI for Health, Helmholtz Zentrum München, Neuherberg, Germany. [15]Institute for Molecular Medicine Finland (FIMM), University of Helsinki, Helsinki, Finland. [16]Institute for Stroke and Dementia Research, Klinikum der Universität München, Ludwig-Maximilians University Munich, Munich, Germany. [17]Munich Cluster for Systems Neurology (SyNergy), Munich, Germany. [18]School of Medicine, Koç University, İstanbul, Turkey. [19]Cardiopulmonary Institute, Justus Liebig University, Giessen, Germany. [20]School of Computation, Information and Technology, Technical University of Munich, Munich, Germany. [21]These authors contributed equally: Louis B. Kuemmerle, Malte D. Luecken. ✉e-mail: fabian.theis@helmholtz-munich.de

## Methods

### Probe set evaluation metrics

We assessed selection method performance or probe set quality via multiple metrics of the following categories: cell type identification, variation recovery, gene redundancy, fulfillment of technical constraints and computation time. For calculation of the metrics, a reference dataset (typically scRNA-seq) is used. The reference is reduced to a pre-selection of HVGs (8,000 in our use cases; scanpy's highly_variable_genes function with the cell_ranger flavor option). In the following, the metrics of each category are described.

**Variation recovery.** For the clustering similarity metrics, Leiden clusterings for different numbers of clusters $n_c$ are calculated via binary search over Leiden resolutions. This way, sets of clusterings for $n_c \in \{7, 8, \ldots, 60\}$ are produced for the 8,000 reference genes and the gene set that is evaluated. For each $n_c$, the similarity of the clusterings between the reference and the gene set are measured via normalized mutual information (NMI):

$$\mathrm{NMI}(U, V) = \frac{\mathrm{MI}(U, V)}{\mathrm{mean}(H(U), H(V))},$$

with $U$, set of sets of cells in each cluster $U = \{U_1, U_2, \ldots, U_{n_c}\}$; $V$, like $U$ for the second clustering; MI, mutual information, which is defined as

$$\mathrm{MI}(U, V) = \sum_{i=1}^{|U|} \sum_{j=1}^{|V|} \frac{|U_i \cap V_j|}{N} \, \log\left( \frac{N|U_i \cap V_j|}{|U_i||V_j|} \right),$$

with $N$, number of cells; $i/j$, the identifier of each cluster of the clusterings $U/V$; note that $|U| = |V| = n_c$ in our comparisons; $H(U)$, entropy of $U$, which is given by

$$H(U) = -\sum_{i=1}^{|U|} P(i) \log(P(i)) = -\sum_{i=1}^{|U|} \frac{|U_i|}{N} \, \log\left( \frac{|U_i|}{N} \right).$$

The final clustering similarity metrics are given by

$$\text{coarse clustering similarity} = \mathrm{AUC}(\mathrm{NMI}(n_c) | n_c \in \{7, 8, \ldots, 20\})$$

and

$$\text{fine clustering similarity} = \mathrm{AUC}(\mathrm{NMI}(n_c) | n_c \in \{21, 22, \ldots, 60\}),$$

with $\mathrm{NMI}(n_c') := \mathrm{NMI}(U, V)|_{n_c = n_c'}$ and $\mathrm{AUC}(f(x) | x \in \mathrm{I})$, the area under the curve of $f(x)$ over the interval $I$. Due to the nature of the Leiden algorithm, sometimes certain $n_c$ for a given dataset cannot be found. Missing values due to this are imputed by linear interpolation between $\mathrm{NMI}(n_c - 1)$ and $\mathrm{NMI}(n_c + 1)$ or the closest existing values in case of multiple missing data points.

For the neighborhood similarity metric, knn graphs are obtained for the 8,000 reference genes and the gene set that is evaluated. knn graphs are calculated for $k \in \{5, 10, 20, 30, 50\}$ on the PCA space of the gene expression. The neighborhood similarity is given as

$$\text{neighborhood similarity} = \mathrm{AUC}\left( \frac{1}{N} \sum_{i}^{N} \frac{\left| \mathcal{N}_{i,\mathrm{ref}}^{(k)} \cap \mathcal{N}_{i,\mathrm{set}}^{(k)} \right|}{k}, |, k \in \{5, 6, \ldots, 50\} \right),$$

with $N$, number of cells and $\mathcal{N}_{i,ref/set}^{(k)}$, set of neighbors of cell $i$ in the knn graph of the reference (ref) or gene set (set) for a given $k$.

**Cell type identification.** To assess how well cell types can be recovered with a gene subset, we train gradient-boosted forests (Brownlee[44]) for multiclass cell type classification and measure the test set performance. To achieve a robust performance readout, fivefold cross-validation is performed with five different seeds, summing up to $N_m = 25$ models per evaluated gene set. The test set classification confusion matrix of each model is obtained and normalized by the ground truth cell type count. Based on the normalized confusion matrix $\mathrm{CM}_m$ per model $m$, the summary metrics are given as

$$\text{cell type classification accuracy} = \frac{1}{N_c N_m} \sum_{i}^{N_c} \sum_{m}^{N_m} \mathrm{CM}_{m,ii}$$

and

$$\text{percentage of captured cell types} = \frac{1}{N_c} \sum_{i}^{N_c} \theta\left( \frac{1}{N_m} \sum_{m}^{N_m} \mathrm{CM}_{m,ii} \right),$$

with the linearly smoothed step function $\theta(x)$ from 0.75 to 0.85 (that is, $\theta(x \leq 0.75) = 0$ and $\theta(x \geq 0.85) = 1$) and the number of cell types $N_c$.

The metrics marker correlation and the cell type-balanced marker correlation measure how well marker signals of a literature-derived marker list are captured with the selected gene set. Based on the maximal Pearson correlation $r_m^{(\mathrm{max})}$ with the gene set for each marker $m$, the summary metrics are given by:

$$\text{marker correlation} = \frac{1}{N_m} \sum_{m}^{N_m} r_m^{(\mathrm{max})}$$

and

$$\text{cell type-balanced marker correlation} = \frac{1}{N_c} \sum_{c}^{N_c} \max_{m \in M_c} r_m^{(\mathrm{max})},$$

with the number of markers $N_m$, the number of cell types $N_c$ and the set of markers $M_c$ for each cell type $c$.

**Gene redundancy.** Based on Pearson correlations $r_{ij}$ of gene pairs $(i, j)$, we assess the redundancy in a gene set with the overall

$$\text{gene correlation} = 1 - \frac{2}{N_g(N_g - 1)} \sum_{i}^{N_g} \sum_{j>i}^{N_g} |r_{ij}|$$

and the

$$\text{percentage of highly correlated genes} = 1 - \frac{1}{N_g} \sum_{i}^{N_g} \theta(\max_{j \neq i} |r_{ij}|),$$

with the number of genes $N_g$ in the gene set and the linearly smoothed step function $\theta$ (see above).

**Expression constraint violation.** We penalize a gene for too low expression if it is expressed below a lower expression threshold in at least 90% of cells where the gene is expressed >0, that is, the 0.9 expression quantile. Similarly, we penalize too high gene expression if the 0.99 expression quantile over all cells is above an upper threshold. Expression thresholds were obtained from expert experience on too lowly and too highly expressed reference genes (for example, *MALAT1*). Based on cpm log-normalized data, the thresholds were set to 2.3 and 5. Because our data were scran normalized, thresholds were transferred by mapping mean expressions to 1.78 and 4.5. These values are technology specific and can only be roughly estimated. To not set strict thresholds, smoothed penalty functions $P(q)$ over the gene's quantile $q$ with a Gaussian decay below (low expression) and above (high expression) the thresholds were introduced. To assess how strongly a gene set violates the expression constraints, we compute the means over $P^{\mathrm{low}}(q)$ and $P^{\mathrm{high}}(q)$ as the low and high expression constraint violation metrics, respectively.

## Spatial probe set evaluation metrics

We assessed the selection method performance or probe set quality related to spatially relevant measures via two metrics: spatial variation (based on Moran's $I$) and CCI recovery. The metrics were calculated on MERFISH human brain data for selections on matched snRNA-seq data ('Datasets for probe set selection and evaluation'). In the following, the two metrics are described.

**Spatial variation (Moran's $I$).** Moran's $I$[45] measures spatial autocorrelation and can be leveraged to assess whether a gene shows spatial patterning or a random distribution over space. For each gene $g$ in the selected gene panel, Moran's $I$ is calculated over all cells as

$$I_g = \frac{n}{S_0} \times \frac{\sum_{i=1}^{n}\sum_{j=1}^{n} w_{ij}\left(x_i^{(g)} - \bar{x}^{(g)}\right)\left(x_j^{(g)} - \bar{x}^{(g)}\right)}{\sum_{i=1}^{n}\left(x_i^{(g)} - \bar{x}^{(g)}\right)^2},$$

with gene expression $x_i^{(g)}$ of cell $i$, mean expression $\bar{x}^{(g)}$, the number of cells $n$, the normalization factor

$$S_0 = \sum_{i=1}^{n}\sum_{j=1}^{n} w_{ij}$$

and the spatial weights defined as

$$w_{ij} = \begin{cases} 1 \text{ if } (i \neq j) \wedge (i \text{ neighbor of } j) \\ 0 \text{ otherwise.} \end{cases}$$

The neighborhood graph was constructed with six nearest neighbors. The graph and Moran's $I$ were both calculated using Squidpy[46] version 1.2.2. The final metric for a given gene set is calculated as the sum of Moran's $I$ values across the gene panel

$$\text{spatial variation} = \sum_{g} I_g,$$

providing a measure of the overall gene expression variation associated with spatial patterning.

**Cell–cell interaction recovery.** For the validation of the ability of a gene set to maintain information about CCIs in tissues, node-centric expression models (NCEM)[47] (version 0.1.5), a method using graph neural networks to analyze and infer cell communication patterns from spatial molecular profiling data, was used. NCEM identifies key genes in cell–cell communication by analyzing molecular signatures that underpin these interactions, including direct contact and indirect mechanisms. NCEM infers latent cell communication events within tissues by examining the co-occurrence of ligand and receptor expression and gene expression signatures, using spatial cell graphs and molecular profiling assays for a detailed understanding of cellular interactions. For the purpose of validation of the selected gene panels, linear NCEM, a simplified form of NCEM that uses linear graph neural networks to effectively infer cell communication patterns, was used. The model was applied to the human brain MERFISH data ('Datasets for probe set selection and evaluation') with a spatial connectivity graph of neighbors within 20 μm computed with Squidpy. Based on Wald tests, NCEM provides a significance table $p_{g,c_1,c_2}$ that describes whether gene $g$'s expression is affected by the presence of a specific cell type interaction pair $(c_1, c_2)$. Our score for CCI recovery, which counts the number of captured interactions of a given gene set, is defined as

$$\text{CCI recovery} = \sum_{g}\sum_{c_1}\sum_{c_2} \Theta(0.05 - p_{g,c_1,c_2}),$$

with the Heaviside step function $\Theta$.

## Classical feature selection methods

The evaluated classical feature selection methods include DE gene selection, a PCA-based selection, HVG selection, an SPCA-based selection and selection of highly expressed genes.

DE genes were scored with $t$-tests using scanpy's[48] rank_genes_groups function. The highest-scored genes per cell type were selected until the set encompassed $n$ genes. For PCA-based selection, genes were scored according to the sum over the loadings of the first 20 PCs, and the top $n$ genes were selected. HVG selection was performed with scanpy's highly_variable_genes function and the cell_ranger flavor option. For SPCA-based selection, scikit-learn's sparsePCA class was used. With the $\alpha$ argument of sparsePCA, the sparsity of the loading matrix can be controlled. To select a set of $n$ genes, a binary search over $\alpha$ values was conducted to find a setting in which only $n$ genes have loadings >0. For the selection of highly expressed genes, we scored genes based on the mean expression.

## Spapros selection pipeline

Spapros enables an end-to-end probe set selection, which includes probe design constraint-based gene filtering, gene panel selection and probe design for the selected gene panel. Therefore, Spapros encompasses a probe design pipeline and a gene set selection pipeline. We discuss them separately in the following.

**Gene panel selection.** Spapros uses a log-normalized count matrix of all genes or a pre-selection of HVGs and cell type annotations as input. The first step consists of a PCA-based ('Classical feature selection methods') prior selection (100 genes per default). This prior selection biases the next steps to use genes that capture a high degree of variation in the dataset. Next, we train decision trees on the prior selected genes for binary cell type classification for each cell type (that is, cell type of interest versus all other cell types). These trees are highly regularized by setting the max_depth to 3. Thus, robust and interpretable rules are learned for the classification of each cell type. For each tree optimization, a training set of 1,000 cells per cell type are sampled (oversampling if cell count is too low), and performance is assessed on a test set sample of 3,000 cells per cell type. The uniform sampling ensures cell type-balanced training. Additionally, we use class proportion weights for the binary classification of the two classes 'cell type of interest' versus 'other'. Per default, we train 50 trees per cell type each with a different training set and choose the best tree based on the $F_1$ score. To increase classification performance on cell types that are difficult to distinguish from the target cell type, we train additional secondary trees on a subset of cell types. This subset is identified based on the specificities $s(c) = \text{TN}_c/N_c$ of the primary tree of each cell type $c$ in class 'other', with $\text{TN}_c$, number of true negative cells of cell type $c$ and $N_c$, number of cells from cell type $c$ in the test set. Cell types are considered for the secondary tree training if their specificity is either below 0.9 or 1 s.d. below the mean of all specificities but at least 0.02 below the mean. Spapros has a hyperparameter to set the number of further secondary trees that are trained based on previous secondary trees in the same manner. The default is 3, that is, one primary tree and two secondary trees. The trees trained on PCA-selected genes often can be improved, as important genes in the pool are missing. Therefore, we also train trees on DE genes. Trees are trained in the same manner except that additionally an iterative adding of genes from specific DE tests is performed: After tree training, cell types that are difficult to distinguish are identified via specificities (best over primary and secondary trees) in the same manner as that by which cell types were selected for secondary trees. Next, a DE test is performed between the cell type of interest and the identified cell types. The top two of those DE genes per cell type that need optimization are added to the DE pool, and tree training is repeated until an early stopping criterion is reached or up to 12 times. By comparing the performance of trees on PCA genes and trees on DE genes, we identify those cell types that are better

distinguished with DE genes and iteratively add missing genes from DE trees with the highest feature importance to the PCA pool and retrain trees until the same performance is reached. The genes that occur in the final trees are ranked by feature importance and used to build the gene panel. Exact labels for cell type or state clusters are not necessary; once a cluster is identified as interesting, it can be included in the cell type classification task with a generic label to ensure that it is distinguishable from other cell types or states.

Spapros incorporates prior knowledge as a pre-selection of genes that is added to PCA and DE pools; therefore, genes are selected around them. Furthermore, a list of marker genes can be provided. After tree-based selection, the pipeline checks whether at least a user-defined number of markers per cell type are captured (correlation > 0.5; see marker correlation metrics in 'Probe set evaluation metrics').

Experimental expression constraints are incorporated by multiplying PCA and DE scores ('Classical feature selection methods') with a penalty kernel (see expression constraint violation metrics in 'Probe set evaluation metrics').

**Probe design pipeline.** To design probes for a set of given genes, we developed three custom probe design pipelines for the spatial transcriptomic protocols: SCRINSHOT (and HybISS), MERFISH and SeqFISH+. The probe design pipelines were developed with the Oligo Designer Toolsuite (ODT) package[49], which is a Python framework for the development of custom pipelines to design experiment-specific oligonucleotides. Each pipeline has four major steps: (1) probe generation, (2) probe filtering by sequence property and binding specificity, (3) probe set selection for each gene and (4) final probe sequence generation.

For the probe generation step, the user has to define a reference from which the probes should be extracted. The user can choose between different reference sources, that is, NCBI, Ensembl or a custom reference. If a custom reference is chosen, a GTF file with gene annotations and a fasta file with the genome sequence have to be provided. When choosing an NCBI or Ensembl reference, the annotation and genome sequence will be downloaded automatically via FTP from the respective servers using the FtpLoader from the ODT package. Therefore, the user has to define the species, annotation release and taxon (only for NCBI) for the reference. Once the annotation files are loaded, the CustomGenomicRegionGenerator from the ODT package is used to extract user-defined genomic regions from the given annotation files. The genomic regions are stored in a memory-efficient format, which eliminates duplicated sequences stemming from common exons of different gene isoforms while preserving the isoform information. The user can choose from a predefined list of genomic regions, that is, intergenic, gene, CDS, exon, intron, 3′ UTR, 5′ UTR and exon–exon junctions. Those genomic regions are used in two steps of the pipelines: as a background reference for the binding specificity filters and as a reference for the generation of probe sequences. The probe sequences are generated using the OligoSequenceGenerator from the ODT package. Therefore, the user has to define the probe length (can be given as a range) and optionally provide a list of gene identifiers (matching the gene identifiers of the annotation file) for which probes should be created. If no gene list is given, probes are created for all genes in the reference. The probe sequences are generated in a sliding window fashion from the DNA sequence of the noncoding strand, assuming that the sequence of the coding strand represents the target sequence of the probe. The generated probes are stored in a fasta file, where the header of each sequence stores the information about its reference region and genomic coordinates. In a next step, this fasta file is used to create an OligoDatabase, which is the underlying data structure of the ODT package that allows combining different filter and selection functionalities in a custom fashion. When the probe sequences are loaded into the database, all probes of one gene having the exact same

sequence are merged into one entry, saving the transcript, exon and genomic coordinate information of the respective probes. Creating the database that contains all possible probes for a given set of genes concludes the first step of each probe design pipeline. The standard probe generation parameters for the three spatial transcriptomic protocols can be found in Supplementary Table 6.

In the second step, the number of probes per gene is reduced by applying different sequence properties and binding specificity filters. The applied filters differ between the three spatial transcriptomic protocols. For the SCRINSHOT protocol, the following sequence property filters of the ODT package are applied: removal of probes that contain unidentified nucleotides (HardMaskedSequenceFilter), that have a GC content (GCContentFilter) or melting temperature (MeltingTemperatureNNFilter) outside a user-specified range, that contain homopolymeric runs of any nucleotide longer than a user-specified threshold (HomopolymericRunsFilter) or that have no suitable ligation site to form two padlock arms with the user-defined melting temperature (PadlockArmsFilter). For the MERFISH protocol, the following filters are applied: removal of sequences that contain unidentified nucleotides (HardMaskedSequenceFilter), that have a GC content (GCContentFilter) or melting temperature (MeltingTemperatureNNFilter) outside a user-specified range, that contain homopolymeric runs of any nucleotide longer than a user-specified threshold (HomopolymericRunsFilter) or that contain secondary structures like hairpins below a user-defined free energy threshold (SecondaryStructureFilter). For the SeqFISH+ protocol, the following filters are applied: removal of sequences that contain unidentified nucleotides (HardMaskedSequenceFilter), that have a GC content (GCContentFilter) outside a user-specified range, that contain homopolymeric runs of any nucleotide longer than a user-specified threshold (HomopolymericRunsFilter) or that contain secondary structures like hairpins below a user-defined free energy threshold (SecondaryStructureFilter). After removing probes with undesired sequence properties from the database, the probe database is checked for probes that potentially cross-hybridize, that is, probes from different genes that have the exact same or similar sequence. Those probes are removed from the database to ensure uniqueness of probes for each gene. Cross-hybridizing probes are identified with the CrossHybridizationFilter from the ODT package, which uses a BLASTn alignment search to identify similar sequences and removes those hits with the RemoveByBiggerRegionPolicy, which sequentially removes the probes from the genes that have the bigger probe sets. Next, the probes are checked for off-target binding with any other region of a provided background reference. Off-target regions are sequences of the background reference (for example, transcriptome or genome) that match the probe region with a certain degree of homology but are not located within the gene region of the probe. These off-target regions are identified with the BlastnFilter from the ODT package, which removes probes for which a BLASTn alignment search found off-target sequence matches with a certain coverage and similarity, for which the user has to define thresholds. For the SCRINSHOT protocol, the coverage of the region around the ligation site of the probe by the matching off-target sequence is used as an additional filtering criterion. Filtering the probe database for off-target binding concludes the second step of each probe design pipeline. The standard probe-filtering parameters for the three spatial transcriptomic protocols can be found in Supplementary Table 6.

In the third step of each pipeline, the best set of non-overlapping probes is identified for each gene. The OligoSetGenerator from the ODT package is used to generate ranked, non-overlapping probe sets in which each probe and probe set is scored according to a protocol-dependent scoring function. For the SCRINSHOT and MERFISH protocols, the sets are scored by the distance to the optimal GC content and melting temperature, weighted by the number of targeted transcripts of the probes in the set. For the seqFISH+ protocol, the sets are scored by the distance to the optimal melting temperature

penalized if located in a 5′ UTR of the probes in the set. The identification of the best-scored non-overlapping set of probes for each gene concludes the third step of each pipeline. After this step, all genes with an insufficient number of probes (user defined) are removed from the database and stored in a separate file for user inspection.

In the last step of each probe design pipeline, the ready-to-order probe sequences containing all additional required sequences are designed for the best non-overlapping sets of each gene. For the SCRINSHOT protocol, the padlock backbone is added to each probe, and a detection oligonucleotide is created for each probe by cropping the probe with even nucleotide removal from both ends, exchanging thymines for uracils and placing the fluorescent dye at the side with the closest uracil as described by Sountoulidis et al.[13] For the MERFISH and seqFISH+ protocol, two and four readout sequences are added to the probe, respectively, creating the encoding probes. A pool of readout probe sequences is created from random sequences with equal per-base probability that have a GC content (GCContentFilter) within a user-specified range and no homopolymeric runs of three or more G nucleotides (HomopolymericRunsFilter). Additionally, the readout probes are checked for off-target binding (BlastnFilter) against the transcriptome and cross-hybridization (CrossHybridizationFilter) against other readout probe sequences, from which hits are removed with the RemoveByDegreePolicy, which iteratively removes readout probes with the highest number of hits against other readout probes. The readout probes are assigned to the probes according to a protocol-specific encoding scheme described for MERFISH by Wang et al.[50] and for SeqFISH+ by Eng et al.[37] In addition, one forward primer and one reverse primer are provided. The reverse primer is the 20-nucleotide T7 promoter sequence (TAATACGACTCACTATAGGG), and the forward primer is created from a random sequence with equal per-base probability that fulfills the following criteria: GC content (GCContentFilter) and melting temperature (MeltingTemperatureNNFilter) within a user-specified range, CG clamp at the 3′ terminal end of the sequence (GCClampFilter), no homopolymeric runs of any nucleotide longer than a user-specified threshold (HomopolymericRunsFilter) and no secondary structures below a user-defined free energy threshold (SecondaryStructureFilter). Furthermore, the forward primer sequence is checked for off-target binding (BlastnFilter) against the transcriptome, the encoding probes and the T7 primer. The standard final probe sequence generation parameters for the three spatial transcriptomic protocols can be found in Supplementary Table 6.

### Datasets for probe set selection and evaluation
Our experiments and analyses comprise three human lung scRNA-seq datasets: Madissoon2020 (ref. 51), Krasnow2021 (ref. 52), Meyer2022 (ref. 38), an scRNA-seq dataset and an untargeted spatial transcriptomics dataset of the developing human heart, Asp2019 (sc/ST)[3], and an scRNA-seq–snRNA-seq adult human heart dataset, Litvinukova2020 (ref. 53). The datasets are all publicly available. Cell type annotations were obtained from the original publications. For fair comparisons, the annotations were filtered or pooled to coarse annotations in those analyses when necessary (Supplementary Table 1). All single-cell and single-nucleus datasets were preprocessed in the same manner: raw counts were normalized with scran[54] using Leiden clusterings[55] with a resolution of 0.5 on a temporary log normalization to $10^6$ counts per cell. The logarithm of the scran-normalized data plus one pseudocount was taken. Features were reduced to the top 8,000 HVGs selected with scanpy's highly_variable_genes function (flavor, cell_ranger). A detailed summary of which dataset and annotation were used in each analysis is given in Supplementary Table 1. For some datasets, we only used a subset of cells due to different reasons: for Meyer2022, we only used single cells and not nuclei because we used it for our SCRINSHOT experiment and assumed that single-cell expression is closer to the observation in a targeted spatial transcriptomic experiment than that of single nuclei. Only the developmental stage at 6 weeks from Asp2019

spatial transcriptomics was used because the selected in situ sequencing (ISS) panel in their study was also selected on that subset. The heart atlas Litvinukova2020 was reduced to single-nucleus observations and a maximum of 2,000 cells per cell type to reduce the computation time of our evaluations (56 cell types, including 58,966 cells).

**Datasets for method benchmark.** Aside from the within-dataset benchmarks on the Madissoon2020 and Litvinukova2020 datasets, we benchmarked performances across multiple datasets including ten additional datasets. Specifically, this included the Tabula Muris Senis atlas[56], the Human Lung Cell Atlas[42], an immune cell atlas[57] and datasets of liver[58], thymus[59], primary motor cortex[60], PBMCs[61] and lung[52], a bone marrow atlas[62–64] and a separate bone marrow dataset[65]. The additional datasets were processed as described above, except that we normalized by total counts instead of applying scran normalization. The number of cell types in the datasets varied from 10 to 122 (median = 40.5). Datasets with more than 100,000 cells were subsampled by reducing the number of cells for the most abundant cell types.

**Matched snRNA-seq and MERFISH data for spatial metrics.** To assess scores for the spatial metrics as well as for the translation of the dissociated metrics to spatial measurements, a public matched snRNA-seq and spatial dataset of the human brain was leveraged. The spatial dataset from Fang et al.[66] included five samples from the human middle temporal gyrus (MTG) region collected by spatially resolved single-cell profiling of 4,000 genes using MERFISH. MERFISH measurements were performed using samples from the human MTG and the superior temporal gyrus from freshly frozen neurosurgical and postmortem brain. The single-nucleus profiling SMART-seq dataset of the MTG from Hodge et al.[67] that was used in the comparative analysis of Fang et al. was leveraged as a reference for our selections. From 4,000 genes in the published MERFISH dataset, 3,491 genes were used for selections, as only those were present in the snRNA-seq reference. Cell types of the MERFISH dataset that were not present in snRNA-seq data were filtered out to make the datasets comparable.

### SCRINSHOT experiment
**Samples and histology.** Samples were obtained from deceased transplant organ donors by the Cambridge Biorepository for Translational Medicine with informed consent from the donor families and approval from the NRES Committee of East of England−Cambridge South (15/EE/0152). Lung biopsies (~2 cm³) were freshly frozen in OCT (Leica Surgipath, FSC22) and shipped to Stockholm University on dry ice. Quality control was carried out by evaluating histopathological condition (sections stained with hematoxylin and eosin were analyzed by the pathologist) and RIN value analysis. Healthy samples with RIN values above 4 were selected for SCRINSHOT. The samples include a biopsy from the distal lung of a 28-year-old male (smoker) and a biopsy from the trachea of a 61-year-old male (non-smoker), and both donors had no notable reported lung or tracheal conditions. Sections of lung tissues were cut at a thickness of 10 μm and placed on poly-lysine slides (Thermo, J2800AMNZ) and then stored frozen at −80 °C for further use.

**Probe design.** At the time when the probe set for the SCRINSHOT experiment was selected with the Spapros gene panel selection, the probe design pipeline was not finished; therefore, probes were designed manually. A detailed description of the padlock probe design is provided in previous publications[13,68]. Sequences for probes (38−45 nucleotides in length) were selected using the PrimerQuest online tool (Integrated DNA Technologies) for the targeted mRNA of 64 genes in the gene selection list. These sequences were then interrogated against the targeted organism genome and transcriptome with the BLASTn tool (NLM) to guarantee their specificity. Two to four specific sequences per gene were selected for further padlock design. An extra sequence was selected to create a unique barcode for the detection

probe, which was reused for each padlock of the same gene with several T nucleotides replaced with U, as described previously[13,68]. All detection probes were then interrogated against all padlock probes using the BLASTn tool to ensure no overlapping sequences and avoid unspecific detection probe binding. An overlap of nine or more nucleotides was avoided by modification of the detection barcode by replacing one to two nucleotides. One RCA primer sequence was used for all padlock probes, taking into account the preceding gene expression level before selection. Sequences for padlock and fluorophore-labeled detection probes are provided in Supplementary Table 2. Both types of probes (248 padlock and 64 detection probes) were ordered from Integrated DNA Technologies.

**SCRINSHOT procedure.** The SCRINSHOT procedure was followed exactly as described previously[13,68] with extra-stringent detection probe incubation (30 °C) in 30% formamide and an increased (20%) formamide concentration in the washing buffer in the following step to avoid unspecific binding of detection probes. After a trial experiment, *SCGB1A1* and *SCGB3A1* probe concentration was reduced to one padlock per gene to avoid dot crowding. Probes were applied in sets of five per hybridization cycle for a total of 13 cycles to detect all 64 genes in each sample (Supplementary Table 3). After each cycle, the whole slide was imaged as a *Z* stack with 11 steps of 0.8 µm (to cover the whole 10-µm thickness) using a widefield microscope (Zeiss Axio Observer Z.2, Carl Zeiss Microscopy, with a Colibri LED light source, equipped with a Zeiss Axiocam 506 mono digital camera and an automated stage) at ×20 magnification. Maximum intensity orthogonal projection was then used for further analysis as described previously[13,68]. One sample from an area corresponding to the alveolar parenchyma collected from the upper part of the left lobe (location 5, Luecken et al.[69]) with a substantial amount of signal from most of the probes was selected for gene pre-selection evaluation.

**SCRINSHOT image analysis.** Images were originally taken, projected (maximum intensity projection) and stitched using ZEN software (ZEN 2.3 Lite). Channels were then exported as TIFF files for further processing. Images were aligned using the DAPI channel, followed by manual nuclear segmentation for the intralobar region and automated nuclear segmentation for the tracheal region. For automated segmentation, a Mask R-CNN convolutional deep neural network model was used as part of the nucleAIzer pipeline. The final model is trained so that the annotated image set was augmented with artificially created ones[70]. The training set contained 50,000 single nuclei manually annotated by experts on ×40 magnification microscopy images. The trained network was integrated into Biological Image Analysis Software (BIAS)[70,71] and is available at http://single-cell-technologies.com/download/. Automated dot detection using CellProfiler version 3.1.9 was performed as described previously[13,68]. All detected dots were assigned to each cell ROI in Fiji (ImageJ1.53c) (https://github.com/AlexSount/SCRINSHOT/blob/master/automated_stitching_dot_counting_v1_19genes.ijm). The resulting dataset containing dots per ROI was used for further analysis.

**SCRINSHOT analysis.** Cells with less than ten counts were filtered out. Counts were normalized by segmentation area and then logarithmized. The cells were clustered with the Leiden algorithm, and cell types were annotated by comparing expression profiles of Spapros markers for each cell type with those of the Meyer2022 scRNA-seq reference. Inclusion of some genes affected clustering in a worse separation of cell types. Those genes were therefore left out for clustering (Supplementary Table 4). These genes include broadly expressed genes with intra-cell type variation like *FOS*. They were identified based on PCA scores in the Spapros selection and manual inspection of mean expression over cell type clusters.

For the tracheal sample, only genes that were relevant for identification of basal cells were included in the clustering (Supplementary Table 4). We searched for genes with orthogonal intra-cell type variation in *KRT15* and *S100A2* based on low prediction scores of a linear regression on *KRT15* and *S100A2* and high abundance in basal cells. *FOS* was revealed as a strong candidate in comparison with all genes. For the smoothed spatial expression profile of *KRT15*, *S100A2* and *FOS* along the epithelium, scikit-learn's B-spline fit with ten knots, degree 10 and $L_2$ regularization with $\alpha = 10^{-3}$ were used.

**Immunofluorescence.** IF for FOS in tracheal basal cells was performed using closely associated serial sections from the same tracheal sample as was used for SCRINSHOT. Freshly frozen sections were fixed with 4% paraformaldehyde and washed with PBS. Antigen retrieval was performed using Tris-EDTA for 30 min at 80 °C, followed by blocking with 5% donkey serum and 0.2% Triton X-100 and overnight incubation with primary antibodies (anti-FOS, rabbit, Novus Biologicals, NBP1-89065 at 1:200; anti-KRT5, chicken, BioLegend, 905901 at 1:1,000; anti-MUC5AC, mouse, Thermo Fisher Scientific, MA5-12178 at 1:100). Donkey anti-mouse Cy5, anti-rabbit Cy3 and anti-chicken AF488 (Jackson ImmunoResearch) secondary antibodies were applied at 1:400. Sections were counterstained with DAPI and imaged as described for SCRINSHOT. Nuclear segmentation was performed using BIAS as for SCRINSHOT, and KRT5-positive MUC5AC-negative cells were selected for analysis. Levels of FOS were detected per nucleus using mean fluorescence intensity measurements in Fiji (ImageJ).

**Immunofluorescence analysis.** For validation of the spatial intra-cell type variation of FOS along the epithelium of the human trachea, IF and SCRINSHOT signals were compared to each other. To correlate the FOS IF and SCRINSHOT signals along the epithelia of the two adjacent slides, the two epithelia were first manually registered as follows. Based on the fluorescence measurements, the paths along the epithelia were annotated with napari[72] (version 0.4.17). Additionally, based on the major epithelial folds in the slice, six landmarks along the paths were annotated to match the two paths. With scikit-image's[73] (version 0.21.0) profile_line function, the mean intensity along the path was quantified using a width of 101 pixels (22 µm). To check the robustness of the measured correlations, the results were double checked with varying widths, showing the same results. As the measured profiles between the landmarks slightly differ in lengths between the two tissue slices, the shorter profile between landmark pairs was linearly interpolated to generate two profiles with the same number of points. Finally, the Pearson correlation of the mean FOS intensity profiles of the two measurements was calculated.

**Selection method benchmark**
**Curated marker list and ISS panel.** The curated lung marker list (Supplementary Table 5) was provided by lung experts (Acknowledgements) and is a collection of airway wall markers from various publications. We reduced the number of genes in the marker list to 155 by only allowing up to ten genes per cell type and from those the ones that occur in the 8,000 HVGs from the Meyer2022 dataset ('Datasets for probe set selection and evaluation').

For our comparison with an ISS panel, we took the original gene set from Asp2019 (ref. 3), which contains 69 genes that were selected based on an scRNA-seq dataset and an untargeted spatial transcriptomics dataset. To generate a comparable selection with Spapros, we selected 34 genes in the untargeted dataset and used these as prior knowledge selection ('Spapros selection pipeline', 'Gene panel selection') for a selection of 69 genes in the scRNA-seq dataset.

**Method benchmark.** We leveraged our Snakemake pipeline to run the method benchmark. To assess statistical significance for metric performance, multiple selections were conducted on bootstrap samples of the datasets. Bootstrap sampling was applied per cell type of a given dataset to retain cell type frequencies. Furthermore, Gaussian noise

with $\sigma = 0.001$ (that is, relatively small compared to the log-normalized count values) was added to the duplicated cells. Statistical significance was measured with $t$-tests when comparing within-dataset performance and paired $t$-tests when comparing across datasets. All methods were benchmarked on the datasets Madissoon2020 and Litvinukova2020 ('Datasets for probe set selection and evaluation' and Supplementary Table 1) with 20 bootstrap samples (exception, PERSIST with five samples due to time reasons). Based on the results of the initial comparisons, the top performers Spapros, SpaprosCTo, DE, NS-Forest, geneBasis, SCMER, SelfE and PCA were compared for statistical significance over 12 datasets in total ('Datasets for probe set selection and evaluation'). In this extended comparison, paired $t$-tests between methods were run over the mean performances of the 12 datasets. The mean performance was calculated based on five bootstrap samples for each dataset.

For all selections, 12-CPU cores and 64 GB of memory were allocated except for the method PERSIST, for which 6 CPU, 64 GB and 1 GPU were allocated. As recommended for the method SCMER, subsampling the dataset is necessary to run the method in a reasonable time. We followed their recommendations of subsampling to 10,000 cells. As the methods SelfE and scPNMF exhibit long computation times, we applied the same subsampling scheme for them as for SCMER. If a selection took longer than 2 d, it was interrupted and not added (SelfE and ASFS for 150 genes).

**External selection methods.** We compared Spapros with 11 other methods dedicated to gene selection. These methods are described as follows.

NS-Forest[22] is a marker gene selection algorithm based on random forest importance scores combined with a binary expression scoring approach to select markers that are specifically upregulated in the cell type of interest but not in other cell types. NS-Forest is available as a repository of Python functions.

SMaSH[27] is a general computational framework for extracting marker genes. Different base classification models can be used: three different forest-based ensemble learners and a neural network. Gini importance and Shapley values are used for scoring genes for the forest models and the neural network, respectively. As the authors describe that the XGBoost base model performs consistently excellently in terms of yielding low marker gene classification rates, we chose this configuration for our comparisons. SMaSH is available as a Python package on PyPI.

scGeneFit[20] selects gene markers that jointly optimize cell label recovery using label-aware compressive classification methods. The method finds a projection to the lowest-dimensional subspace for which samples with different labels remain farther apart than samples with the same label, while the subspace dimensions are individual genes. The optimization is formulated as a linear program. The method not only finds marker genes that are specifically expressed in single cell types but also genes that reflect the hierarchical structure of cell types. scGeneFit is available as a Python package.

The ASFS (or ActiveSVM)[26] selection procedure generates minimal gene sets from single-cell data by employing a support vector machine classifier. The method iteratively adds more genes by identifying cells that were misclassified. ASFS is available as a Python package on PyPI.

COSG[28] is a cosine similarity-based method for marker gene selection. It is fast and scalable and particularly designed for selection of marker genes in large datasets. COSG is available as Python and R packages.

SelfE[31] aims to select a subset of genes that is optimized for prediction of all remaining genes as linear combinations. The gene subset is constructed iteratively, and each step the gene that minimizes the $L_2$ error over genes is added. SelfE is available as an R package.

SCMER[32] selects a set of genes that reconstructs a pairwise similarity matrix between cells and therefore preserves the manifold of the scRNA-seq data. To find that sparse set of features, a binary search on the $L_1$ regularization parameter is performed. Similar to SelfE, this method optimizes for general variation opposed to the previously described marker gene-focused and cell type classification-focused methods. The method is available as a Python package.

Triku[35] selects genes that are locally overexpressed in groups of neighboring cells, which aims to recover cell populations and general variation in scRNA-seq data. The method is available as a Python package.

scPNMF[21] selects genes based on non-negative matrix factorization in an unsupervised manner with additional filtering steps. The sparse matrix factorization aims to capture the variation of the dataset with a reduced number of genes, while the filtering step aims to increase the likelihood of selecting genes with, for example, relevance for cell type classification or robustness to batch and/or modality effects.

geneBasis[33] selects gene panels in an unsupervised iterative approach, in which each newly added gene captures the maximum distance between the true manifold represented by a knn graph and the manifold constructed using the currently selected gene panel.

PERSIST[34] is a flexible deep learning framework that identifies a set of informative genes to either reconstruct gene expression or predict cell type labels based on binarized counts. The method leverages custom layers enabling feature selection and tailored loss functions to account for the intricacies of single-cell transcriptomics. In our comparisons, we call the unsupervised version PERSISTus and the supervised version PERSIST.

## Statistics and reproducibility

In the SCRINSHOT experiment, the sample size was limited to a single sample from each of two lung regions. This was considered sufficient because our aim was to show translatability of signals from reference scRNA-seq data to spatial data, rather than examining interindividual variability. No data were excluded from this analysis. While we observed reproducibility within the same donor via IF labeling on adjacent tissue slides, our study does not make claims about the frequency of these signals across a broader patient population. Randomization and blinding were not applicable, as no comparative groups were analyzed.

For the selection method benchmark, a comprehensive statistical assessment was conducted to evaluate the performance of different gene selection methods, including Spapros. The methods were compared using various metrics. We used multiple datasets to ensure robustness and generalizability of the results. No data were excluded from these analyses. Selections on bootstrap samples of the datasets were employed to assess the statistical significance of performance differences, with paired $t$-tests used for comparisons across datasets. No statistical method was used to predetermine sample size of bootstrap samples. No formal randomization or blinding procedures were applied, as the primary focus was on computational comparisons rather than experimental interventions.

## Reporting summary

Further information on research design is available in the Nature Portfolio Reporting Summary linked to this article.

## Data availability

SCRINSHOT expression data are included as an Excel file. IF images and SCRINSHOT segmentations are available on Zenodo[74]. All scRNA-seq, snRNA-seq, untargeted spatial transcriptomics and the MERFISH brain datasets are publicly available[3,38,42,51–53,56–60,62–67]. The datasets used for diverse analyses include Madissoon2020, available at https://www.tissuestabilitycellatlas.org, Krasnow2021 at https://www.synapse.org/#!Synapse, Meyer2022 at https://5locationslung.cellgeni.sanger.ac.uk, Asp2019 ISS at https://doi.org/10.6084/m9.figshare.10058048.v1 (ref. 75), Asp2019 single-cell and spatial transcriptomics at https://data.mendeley.com/datasets/mbvhhf8m62/2, Litvinukova2020 at

https://www.heartcellatlas.org/v1.html and HLCA at https://cellxgene.cziscience.com/collections/6f6d381a-7701-4781-935c-db10d30de293. Other datasets used for benchmarking include the Tabula Muris Senis atlas, accessible via NCBI GEO accession GSE132042; the immune cell atlas at https://www.tissueimmunecellatlas.org; liver datasets available via NCBI GEO accession GSE115469; thymus data at https://developmental.cellatlas.io/thymus-development; primary motor cortex data (10X_v2, 10X_v3 and SMART samples) available from https://assets.nemoarchive.org/dat-ch1nqb7; PBMC data at https://www.10xgenomics.com/datasets/10-k-pbm-cs-from-a-healthy-donor-v-3-chemistry-3-standard-3-0-0; and bone marrow data from https://figshare.com/projects/Single-cell_proteo-genomic_reference_maps_of_the_human_hematopoietic_system/94469 (ref. 76) and GEO accession numbers GSE201333, GSE134355 and GSE192616. The matched brain MERFISH and dissociated data used for the evaluation of spatial metrics can be accessed at https://doi.org/10.5061/dryad.x3ffbg7mw (ref. 77) and https://portal.brain-map.org/atlases-and-data/rnaseq/human-mtg-smart-seq, respectively.

## Code availability

The Spapros package and the probe design pipeline are publicly available at https://github.com/theislab/spapros and https://github.com/HelmholtzAI-Consultants-Munich/oligo-designer-toolsuite, respectively. The end-to-end selection, which combines panel selection and probe design, is described in tutorials of the Spapros package. The Snakemake evaluation pipeline is available at https://github.com/theislab/spapros-smk. Code to reproduce the analyses is available at https://github.com/theislab/spapros_reproducibility.

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

## Acknowledgements

We are grateful to all members of the Theis and Ertürk laboratories as well as the discovAIR consortium for frequent discussions of the project. We thank E. Madissoon and K. Meyer for provision and discussion of the scRNA-seq lung reference datasets. We thank

P. Barbry for provision of the airway marker list. We thank J. Theelke for testing the probe design pipeline. We thank X. Abalo for helping with tissue sectioning and tissue quality control. We thank W. Timens for histopathological tissue evaluation. This work was supported by the project 'Virological and immunological determinants of COVID-19 pathogenesis — lessons to get prepared for future pandemics (KA1-Co-02 'COVIPA')', a grant from the Helmholtz Association's Initiative and Networking Fund. This project has received funding from the European Union's Horizon 2020 Research and Innovation Programme under grant agreement 874656.

## Author contributions

M.D.L. conceived the project. L.B.K. and M.D.L. developed the Spapros method and metrics. L.B.K. conducted the computational experiments. M.D.L., F.J.T., C.S., M.P., A.E. and P.H. supervised the project. A.S. and C.S. selected the parameters for gene and probe selection. L.B.d.A.e.S., I.I.M., F.C. and L.B.K. set up the probe design pipeline. A.S. and A.B.F. revised the probe design pipeline. K.T.M. collected the tissue samples for SCRINSHOT. A.B.F. prepared samples for SCRINSHOT and conducted the SCRINSHOT experiment and IF. T.B. and F.K. conducted the automated nuclear segmentation design of the tracheal sample; the algorithm training dataset of manually drawn nuclei was provided by A.B.F. and A.S. L.B.K. and A.B.F. analyzed the SCRINSHOT and IF data. M.S. and A.C.S. trained NCEM models for the CCI recovery metrics. L.B.K., A.B.F. and S.H.-Z. investigated intra-cell type variation signals. L.B.K., L.S. and L.H. contributed to the Spapros package. L.S. and L.B.K. implemented the external selection methods. L.B.K. and V.B. generated the method comparison table. L.B.K., M.D.L., A.B.F., L.B.d.A.e.S., F.J.T. and M.S. wrote the manuscript. All authors reviewed the manuscript

## Funding

## Competing interests

F.J.T. consults for Immunai Inc., CytoReason Ltd, Cellarity and BioTuring Inc., and has an ownership interest in Dermagnostix GmbH and Cellarity. M.D.L. contracted for the Chan Zuckerberg Initiative and received speaker fees from Pfizer and Janssen Pharmaceuticals. P.H., F.K. and T.B. acknowledge support from TKP2021-EGA09, Horizon-BIALYMPH, -SYMMETRY, -SWEEPICS, -Fair-CHARM and OTKA-SNN 139455.

## Additional information

**Extended data** is available for this paper at https://doi.org/10.1038/s41592-024-02496-z.

**Correspondence and requests for materials** should be addressed to Fabian J. Theis.

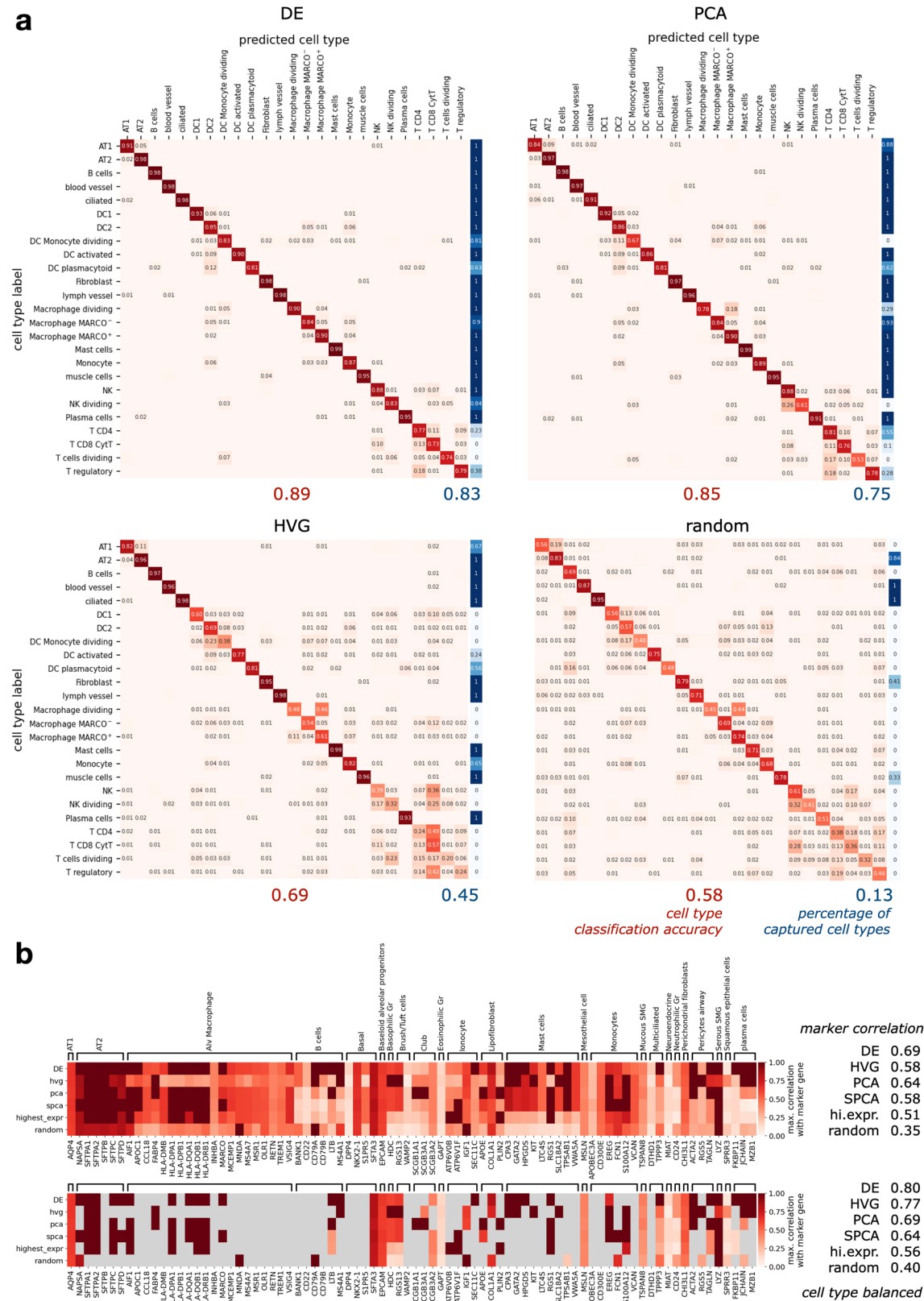

**Extended Data Fig. 1 | Spapros evaluations show cell type specific classification performance.** Evaluations on the Madissoon2020 dataset.
**a**, Normalized cell type classification confusion matrices (red color scale) for gene sets of 150 genes selected with DE, PCA, HVG, and random selection, and linearly smoothed step function of the diagonal elements at 0.8 (blue color scale). The summary metrics *cell type classification accuracy* and *percentage of captured cell types* are the means of the diagonal and the thresholded values

respectively. **b**, Maximal Pearson correlation of marker genes from a curated marker list and gene sets selected with DE, HVG, PCA, SPCA, as well as highest expressed and randomly selected genes. In the bottom heatmap values below the maximum correlation of each cell type are masked (gray). The summary metrics *marker correlation* and *cell type balanced marker correlation* are the row means of all genes (top heatmap) and per cell type (bottom heatmap) respectively.

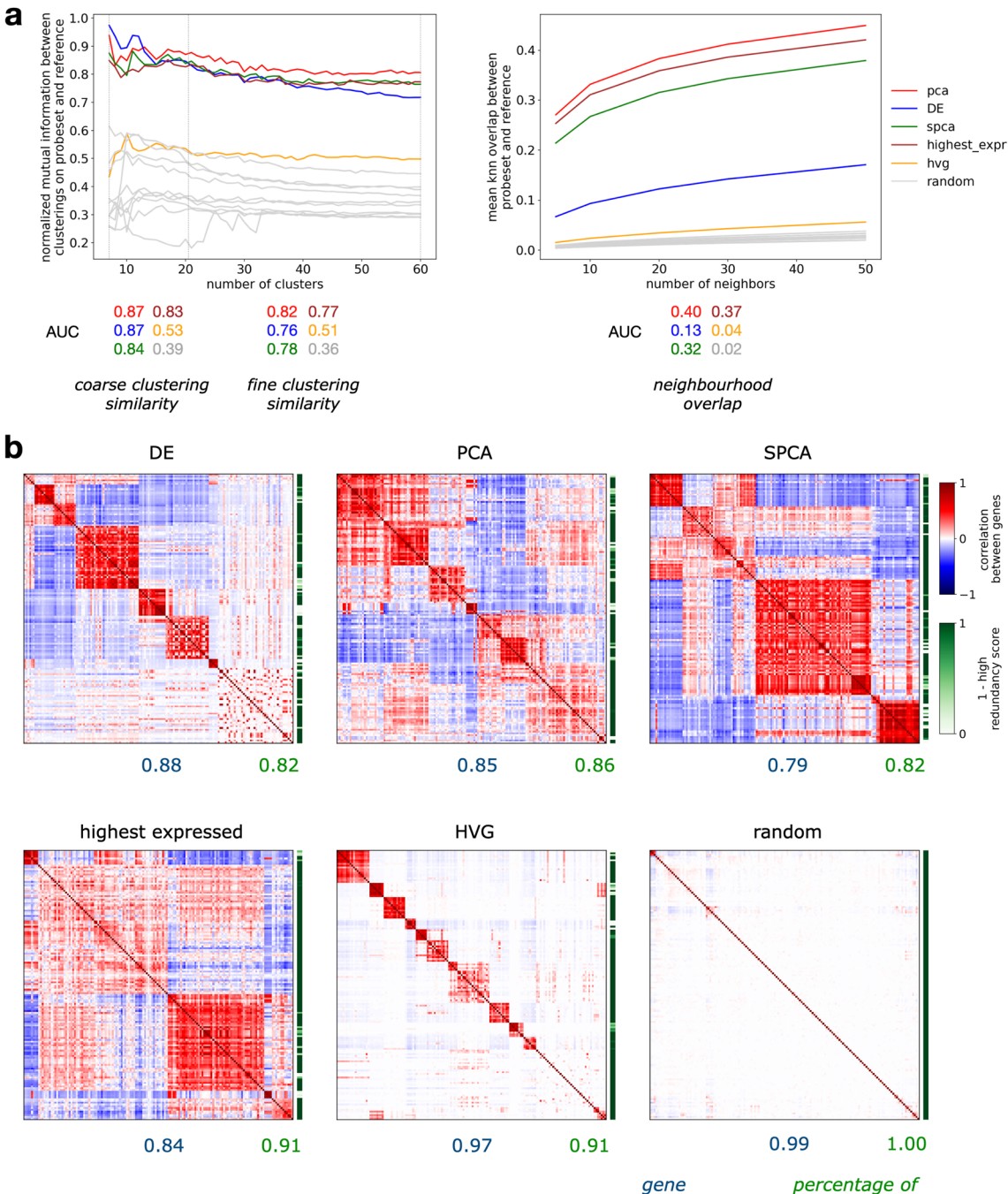

**Extended Data Fig. 2 | Variation recovery metrics for different granularity levels and correlation evaluations. a**, Clustering similarity and neighborhood overlap metrics evaluated on the Madissoon2020 dataset of gene sets with 150 genes selected with PCA, DE, SPCA, HVG, as well as highest expressed genes and random selection. The summary metrics *coarse* and *fine clustering similarity* are the AUCs of the normalized mutual information in the intervals [6,20] and [21,60] respectively, and *neighborhood overlap* is the AUC of knn overlaps over multiple k's. **b**, Gene correlation on the Madissoon2020 dataset of gene sets with 150 genes selected with DE, PCA, SPCA, HVG, as well as highest expressed genes and random selection. The redundancy score is a linearly smoothed step function at 0.8 of the maximal correlation of each gene. The summary metrics *gene correlation* and *percentage of highly expressed genes* are the AUCs of the normalized mutual information in the intervals [6,20] and [21,60] respectively, and *neighborhood overlap* is the AUC of knn overlaps over multiple k's.

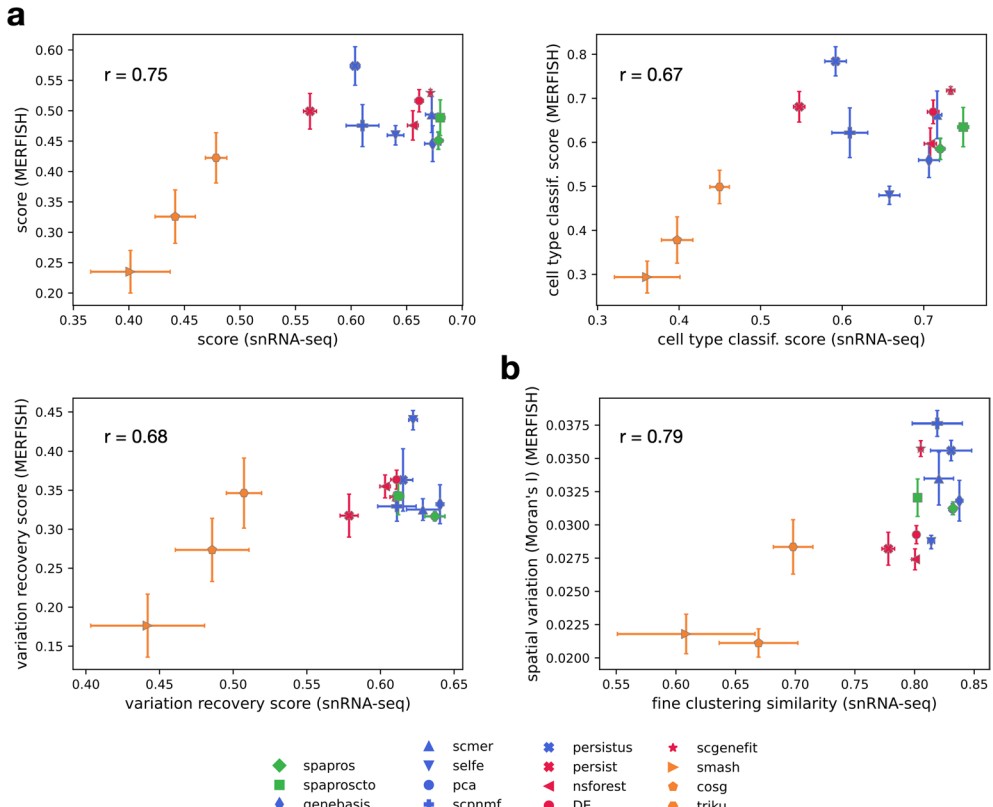

**Extended Data Fig. 3 | Correspondance between dissociated and spatial evaluations. a**, Correlation between performance metrics on dissociated and spatial data based on matched snRNA-seq and MERFISH human brain data. Data are presented as mean values ± SD over selections on 7 bootstrap samples of the snRNA-seq reference for selecting 50 genes. **b**, Correlation between spatial variation metric on the MERFISH data and fine clustering similarity on the snRNA-seq data. Same error bars as in (**a**).

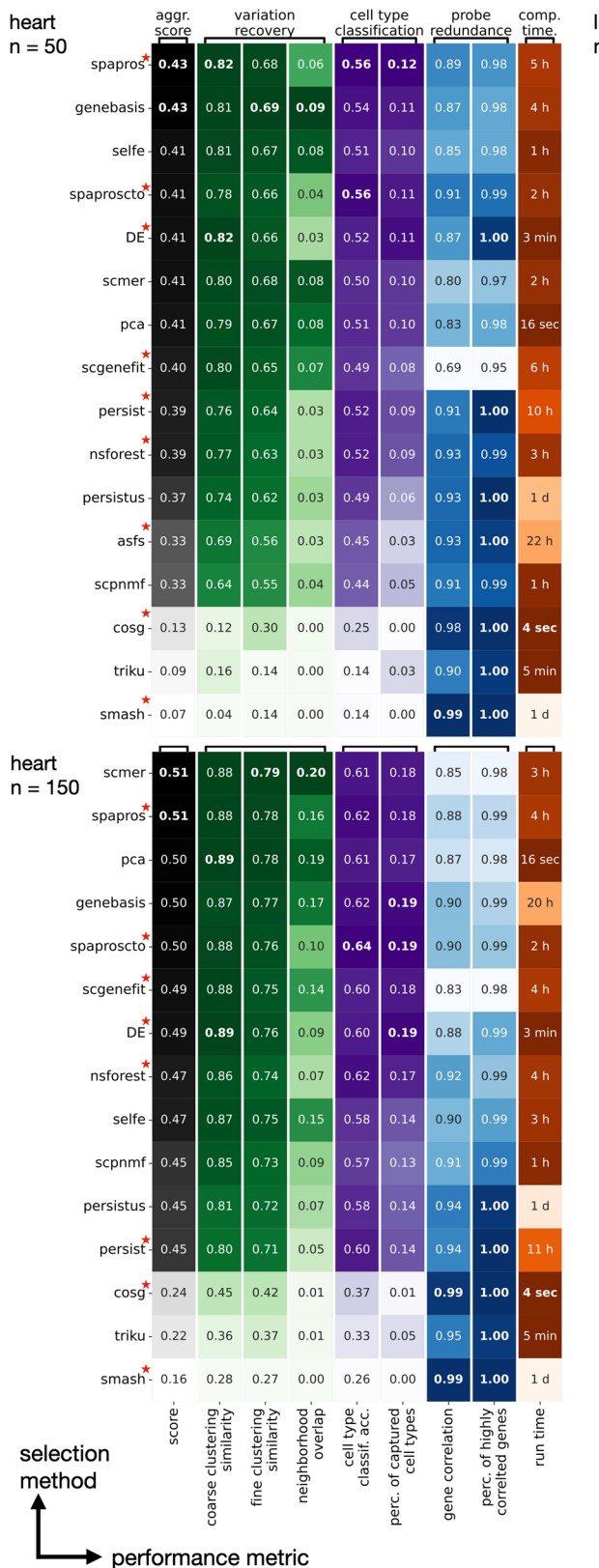

**Extended Data Fig. 4 | Spapros outperforms state-of-the-art methods.**
Heatmap of our evaluation metrics comparing Spapros with recently published methods as well as DE, and PCA-based selections. We compared selections of 50 and 150 genes for lung and heart data sets. Methods are sorted and ranked by the aggregated score of variation recovery and cell type classification. Methods that use cell type information are annotated with a red star.

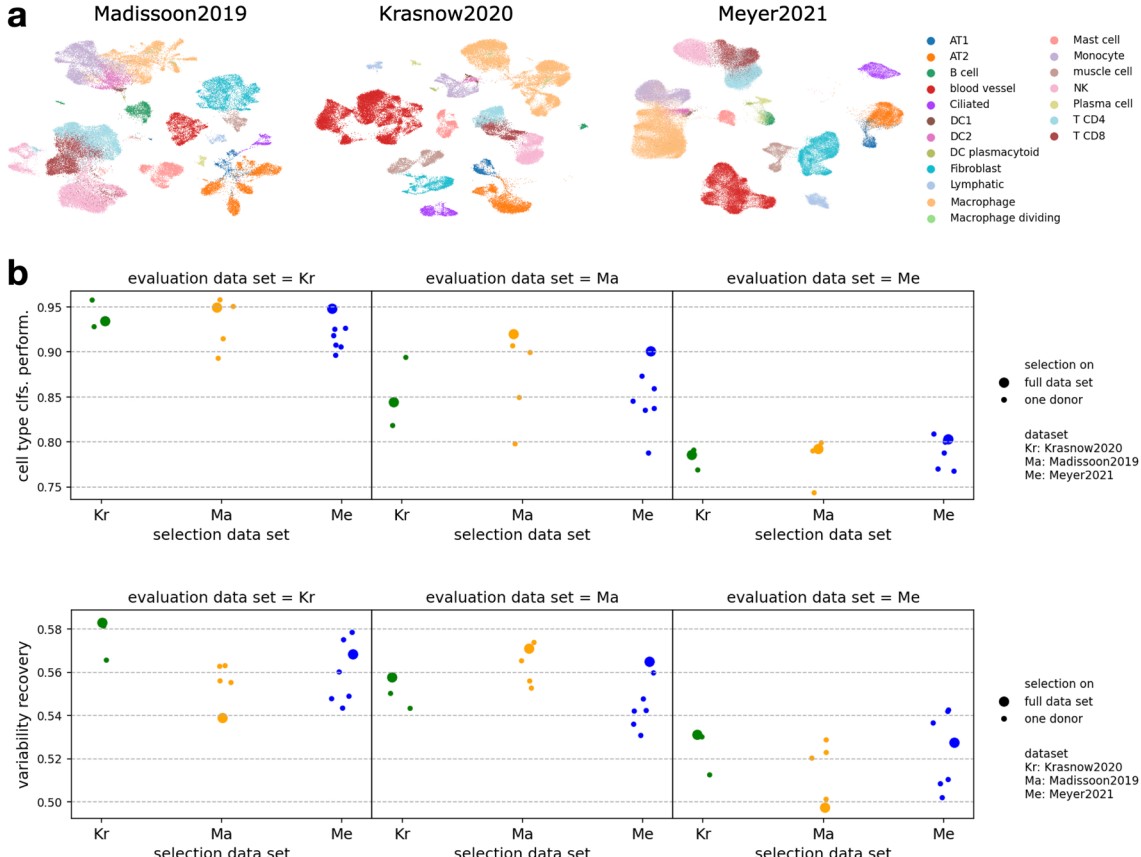

**Extended Data Fig. 5 | Spapros selections show robust cross dataset performance. a**, UMAPs of the three lung datasets with unified cell type annotations for cross dataset evaluation. **b**, Cross dataset evaluations of selections on the lung data sets and on the donor samples within each data set. *Cell type clfs. perform.* is the average of the metrics *cell type classification accuracy* and *percentage of captured cell types*. Variability recovery is the average of the metrics *coarse* and *fine clustering similarity*, and *neighborhood overlap*.

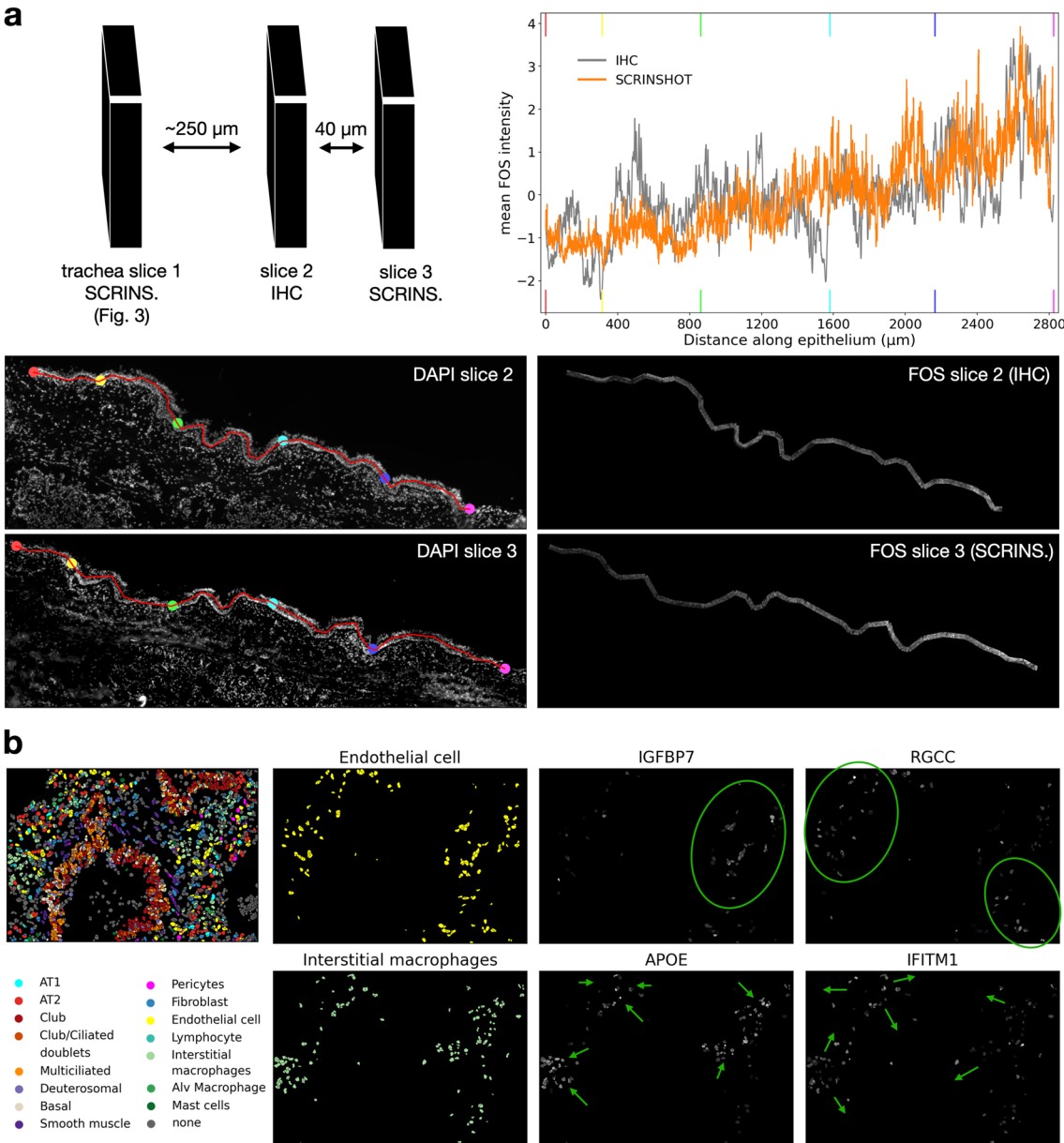

**Extended Data Fig. 6 | Intra-cell type variation and validation with IF. a**, Validation of the spatially variable FOS signal in tracheal basal cells. FOS expression of adjacent IF and SCRINSHOT samples are correlated along the registered annotated tracheal epithelium. **b**, Spatial intra-cell type variation of genes in the intralobar SCRINSHOT lung sample.

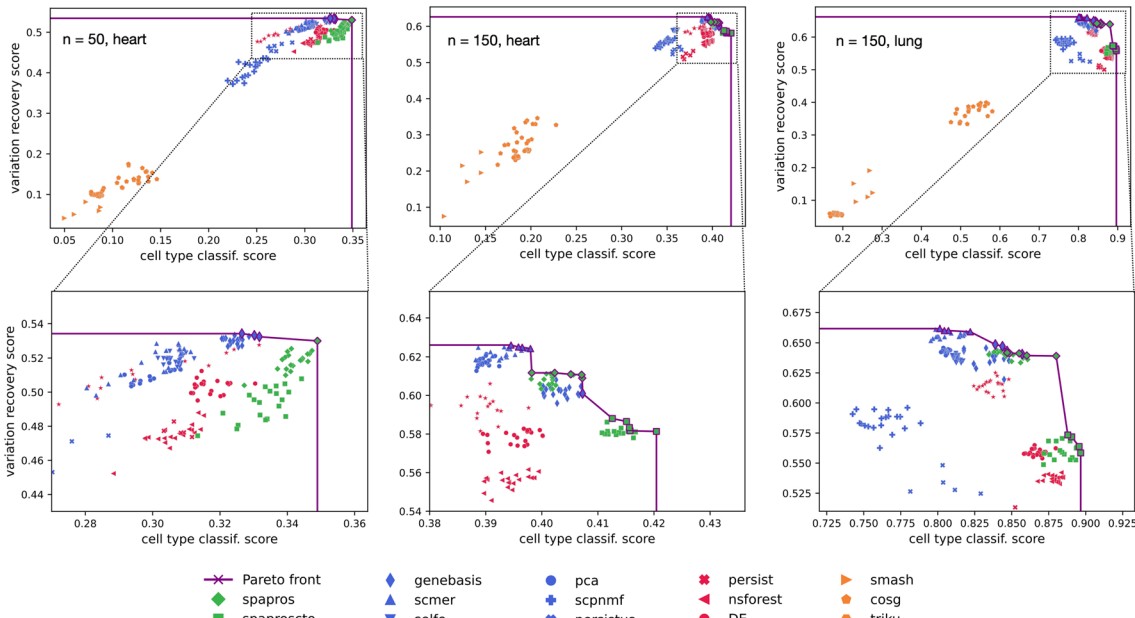

**Extended Data Fig. 7 | Uniqueness of Spapros on balancing performance metrics.** Pareto fronts, showing the trade-off between variation recovery and cell type classification scores for 50- and 150-gene selections from the Madissoon2020 lung and Litvinukova2020 heart data.

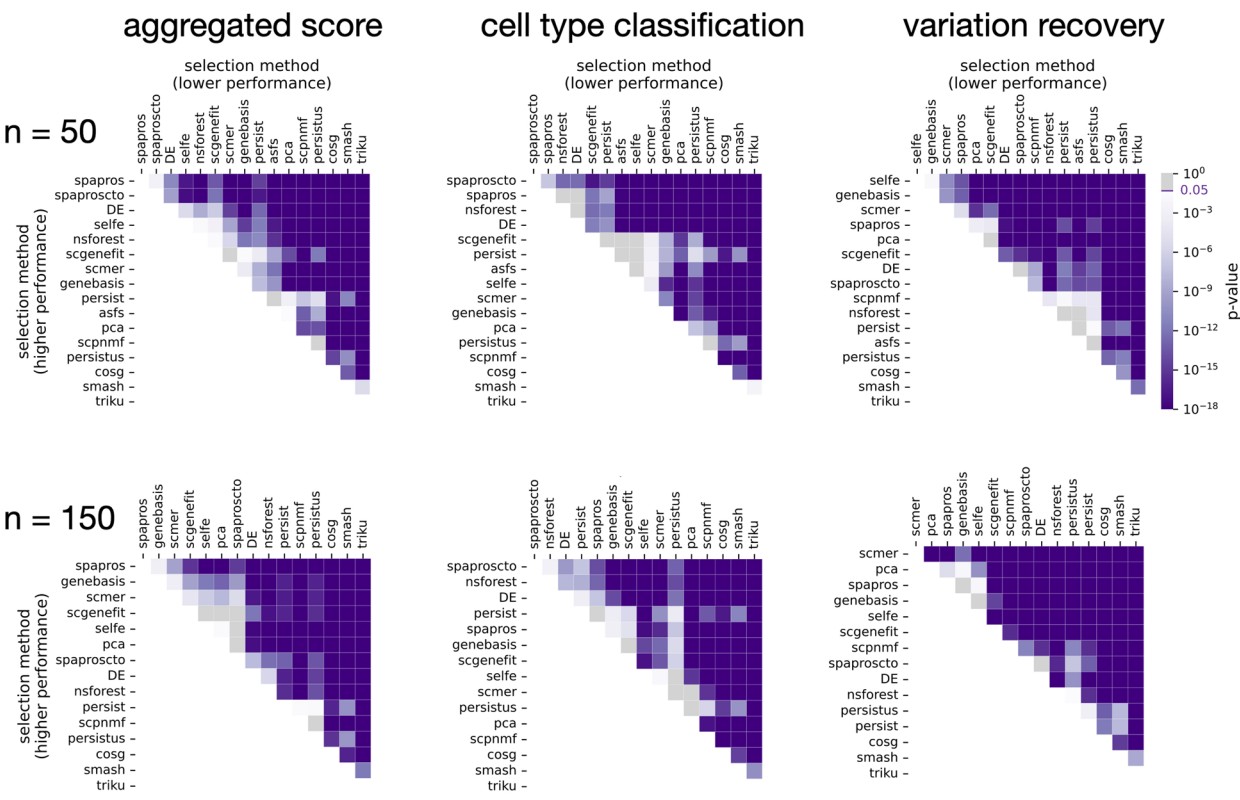

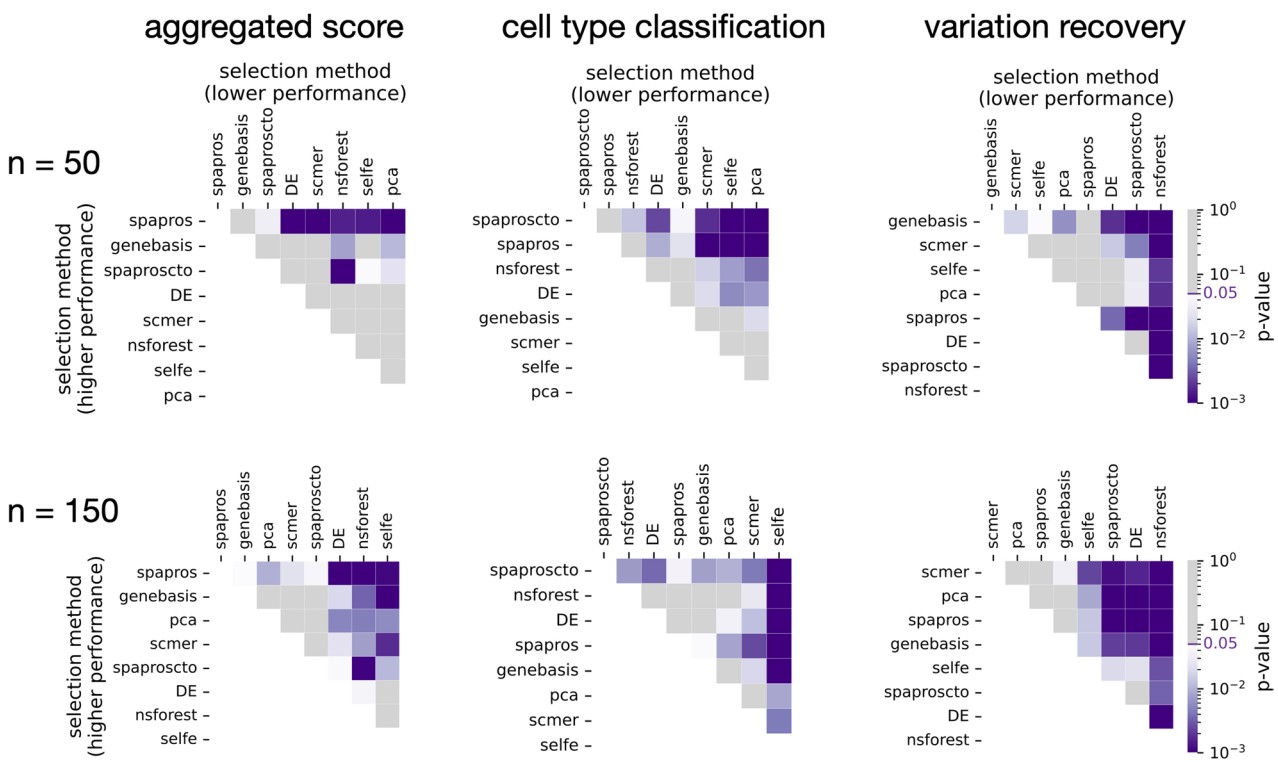

**Extended Data Fig. 8 | Method benchmark significance tables. a**, P-values from two-sided t-tests comparing cell type recovery, variation recovery, and the aggregated score of the different selection methods on selections on bootstrap samples of the Madissoon2020 lung data for 50- and 150-genes selections.

Methods are ranked by mean performance. **b**, Two-sided paired t-test P-values for the mean scores of the same metrics across 12 datasets on 50- and 150-gene selections.

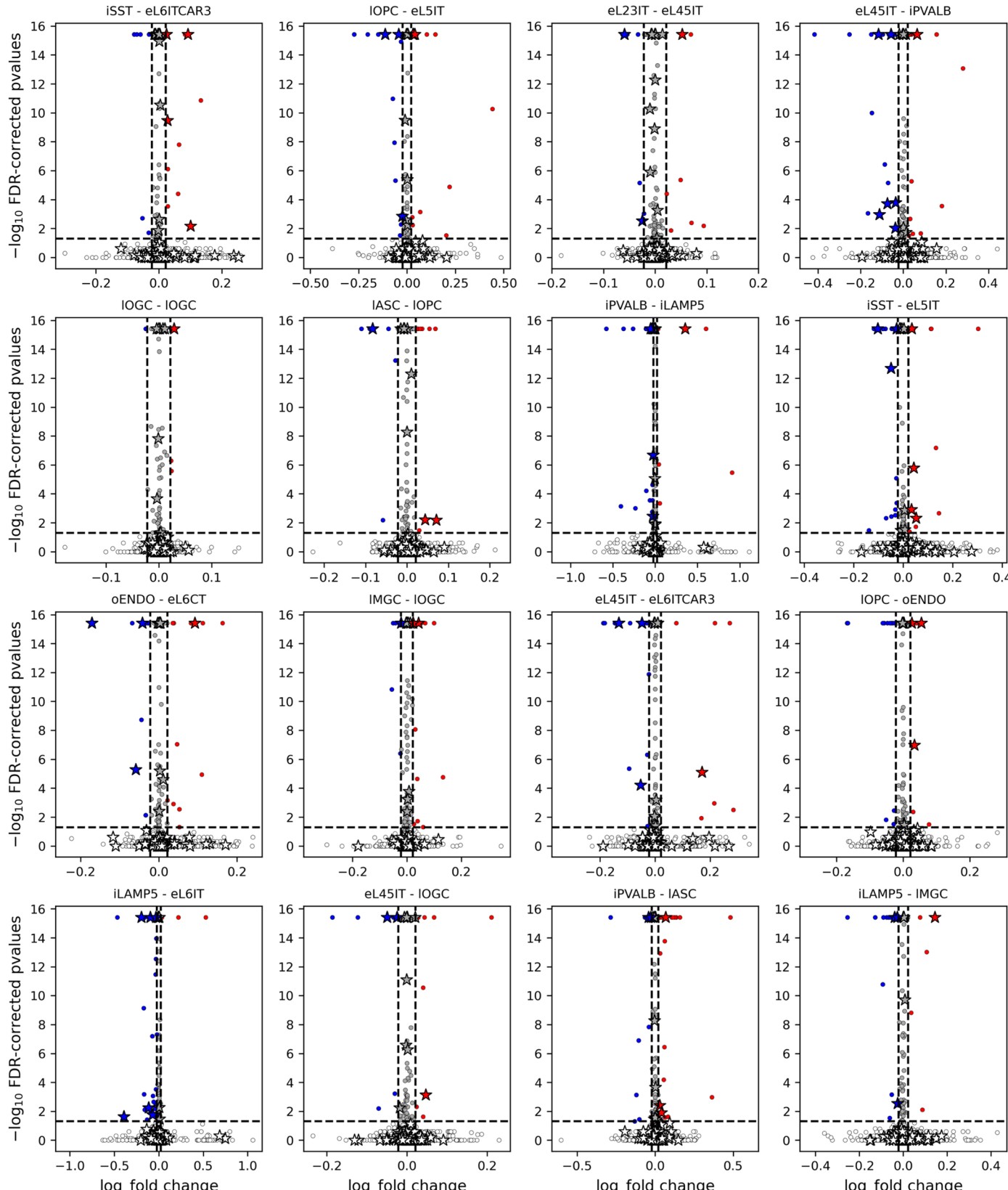

☆ Gene in 150 genes Spapros panel ● Gene up-/ ● down-regulated by the presence of the interaction pair

**Extended Data Fig. 9 | Cell-cell interactions of selected gene sets for MERFISH human brain data.** Volcano plots of the 16 cell type interaction pairs with the highest number of significant genes affected by cell-cell interaction of the given cell type pair (based on two-sided Wald-tests of the NCEM model on MERFISH human brain data). Significant hits are shown for a 150 genes Spapros selection on snRNA-seq human brain data). Genes of the selected gene set are highlighted by star symbols. P-values of 0 were set to the minimal non-zero observed p-value of $-10^{-16}$.

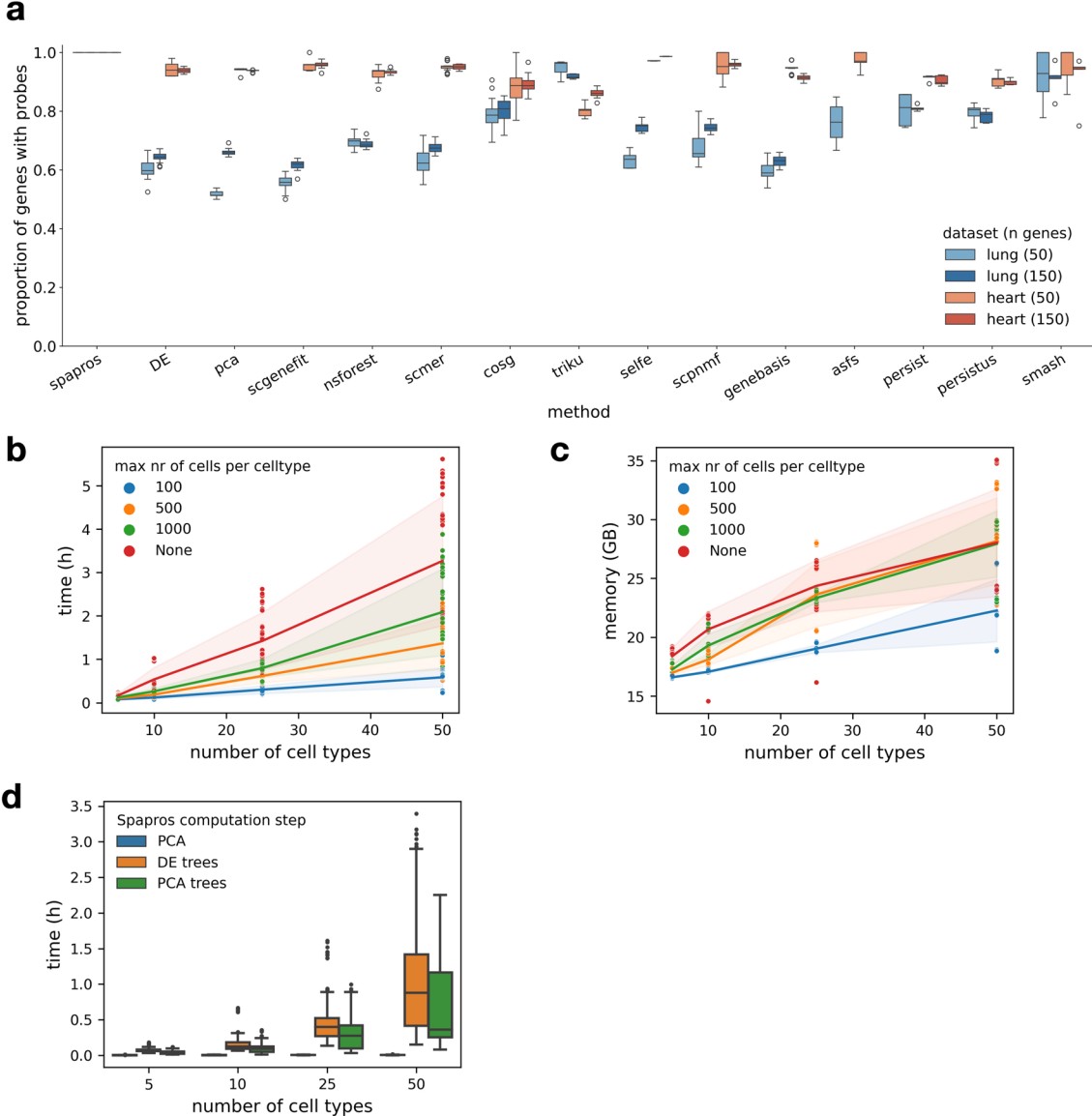

**Extended Data Fig. 10 | Proportion of probe design filtered genes and technical aspects of Spapros and selections. a**, Proportion of genes that pass the SCRINSHOT probe design constraints for the same datasets. Center line, median; box limits, upper and lower quartiles; whiskers, 1.5x interquartile range; points, outliers. **b**, Computation time and **c**, memory of Spapros selections for datasets with different numbers of cell types and cells per cell type. The filled area shows the standard deviation. **d**, Computation time of different steps in the Spapros gene set selection. Center line, median; box limits, upper and lower quartiles; whiskers, 1.5x interquartile range; points, outliers. Each box comprises typical selection scenarios of 100 selections with different numbers of sampled cells per cell type over 5 datasets (same selections as for (**b**) and (**c**)).

# Reporting Summary

Nature Research wishes to improve the reproducibility of the work that we publish. This form provides structure for consistency and transparency in reporting. For further information on Nature Research policies, see our Editorial Policies and the Editorial Policy Checklist.

## Statistics

For all statistical analyses, confirm that the following items are present in the figure legend, table legend, main text, or Methods section.

| n/a | Confirmed | |
|---|---|---|
| ☐ | ☒ | The exact sample size ($n$) for each experimental group/condition, given as a discrete number and unit of measurement |
| ☐ | ☒ | A statement on whether measurements were taken from distinct samples or whether the same sample was measured repeatedly |
| ☐ | ☒ | The statistical test(s) used AND whether they are one- or two-sided<br>*Only common tests should be described solely by name; describe more complex techniques in the Methods section.* |
| ☒ | ☐ | A description of all covariates tested |
| ☒ | ☐ | A description of any assumptions or corrections, such as tests of normality and adjustment for multiple comparisons |
| ☐ | ☒ | A full description of the statistical parameters including central tendency (e.g. means) or other basic estimates (e.g. regression coefficient) AND variation (e.g. standard deviation) or associated estimates of uncertainty (e.g. confidence intervals) |
| ☐ | ☒ | For null hypothesis testing, the test statistic (e.g. $F$, $t$, $r$) with confidence intervals, effect sizes, degrees of freedom and $P$ value noted<br>*Give P values as exact values whenever suitable.* |
| ☒ | ☐ | For Bayesian analysis, information on the choice of priors and Markov chain Monte Carlo settings |
| ☒ | ☐ | For hierarchical and complex designs, identification of the appropriate level for tests and full reporting of outcomes |
| ☐ | ☒ | Estimates of effect sizes (e.g. Cohen's $d$, Pearson's $r$), indicating how they were calculated |

*Our web collection on statistics for biologists contains articles on many of the points above.*

## Software and code

Policy information about availability of computer code

| Data collection | No software was used. |
|---|---|
| Data analysis | For the analysis we used our custom python package "spapros" (https://github.com/theislab/spapros) and additional standard python packages. The environment configuration and all used versions are provided in environment.yaml files in https://github.com/theislab/spapros_reproducibility. For the benchmarked external selection methods code or packages in python or R used. The respective code/package and environments are provided in https://github.com/theislab/spapros_reproducibility. The benchmark was run with our snakemake pipeline https://github.com/theislab/spapros-smk. Spapros' probe design pipeline is given in our separate package https://github.com/HelmholtzAI-Consultants-Munich/oligo-designer-toolsuite. For the analysis of the spatial metrics (CCI score and Moran's I) the packages squidpy (version 1.2.2) and NCEM (0.1.5) were used. For the processing of the SCRINSHOT data the softwares Zen (2.3 Lite), Fiji (ImageJ 1.53c), Cell Profiler (v.3.1.9), and BIAS (version 1.0, available at http://single-cell-technologies.com/download/) were used. For the analysis of the IF samples napari (version 0.4.17) and scikit-image (0.21.0) were used. |

For manuscripts utilizing custom algorithms or software that are central to the research but not yet described in published literature, software must be made available to editors and reviewers. We strongly encourage code deposition in a community repository (e.g. GitHub). See the Nature Research guidelines for submitting code & software for further information.

## Data

Policy information about availability of data

All manuscripts must include a data availability statement. This statement should provide the following information, where applicable:

- Accession codes, unique identifiers, or web links for publicly available datasets
- A list of figures that have associated raw data
- A description of any restrictions on data availability

The data generated during this study are included in the supplementary information files and on Zenodo at https://doi.org/10.5281/zenodo.10731614. All sc/snRNA-seq, untargeted spatial transcriptomics, and MERFISH datasets are publicly accessible. The datasets used for diverse analyses include Madissoon2020, available at https://www.tissuestabilitycellatlas.org; Krasnow2021 at https://www.synapse.org/#!Synapse:syn21041850; Meyer2022 at https://5locationslung.cellgeni.sanger.ac.uk; Asp2019 ISS at https://doi.org/10.6084/m9.figshare.10058048.v1; Asp2019 single-cell and spatial transcriptomics at https://data.mendeley.com/datasets/mbvhhf8m62/2; Litvinukova2020 at https://www.heartcellatlas.org/v1.html; and HLCA at https://cellxgene.cziscience.com/collections/6f6d381a-7701-4781-935c-db10d30de293. Other datasets used for benchmarking include the Tabula Muris Senis Atlas, accessible via NCBI GEO accession (GSE132042) https://www.ncbi.nlm.nih.gov/geo/query/acc.cgi?acc=GSE132042; the immune cell atlas at https://www.tissueimmunecellatlas.org; liver datasets available via NCBI GEO accession (GSE115469) https://www.ncbi.nlm.nih.gov/geo/query/acc.cgi?acc=GSE115469; thymus data at https://developmental.cellatlas.io/thymus-development; primary motor cortex data (10X_v2, 10X_v3, and SMART samples) available from https://assets.nemoarchive.org/dat-ch1nqb7; PBMC data at https://www.10xgenomics.com/datasets/10-k-pbm-cs-from-a-healthy-donor-v-3-chemistry-3-standard-3-0-0; and bone marrow data from https://figshare.com/projects/Single-cell_proteo-genomic_reference_maps_of_the_human_hematopoietic_system/94469, and GEO accession numbers (GSE201333) https://www.ncbi.nlm.nih.gov/geo/query/acc.cgi?acc=GSE201333, (GSE134355) https://www.ncbi.nlm.nih.gov/geo/query/acc.cgi?acc=GSE134355, and (GSE192616) https://www.ncbi.nlm.nih.gov/geo/query/acc.cgi?acc=GSE192616. The matched brain MERFISH and dissociated data used for the evaluation of spatial metrics can be accessed at https://doi.org/10.5061/dryad.x3ffbg7mw and https://portal.brain-map.org/atlases-and-data/rnaseq/human-mtg-smart-seq, respectively.

# Field-specific reporting

Please select the one below that is the best fit for your research. If you are not sure, read the appropriate sections before making your selection.

☒ Life sciences          ☐ Behavioural & social sciences          ☐ Ecological, evolutionary & environmental sciences

For a reference copy of the document with all sections, see nature.com/documents/nr-reporting-summary-flat.pdf

# Life sciences study design

All studies must disclose on these points even when the disclosure is negative.

| | |
|---|---|
| Sample size | The sample size of each lung region equals 1. As we optimized the selected gene set for cell type recovery and variation recovery over multiple reference samples our method aims to recover major variation common across the population. A single sample that shows that variation in space is therefore representative of the observed cell type patterns and major variation axes in healthy human lungs. |
| Data exclusions | No data was excluded. |
| Replication | Our measurements of human lung SCRINSHOT samples with Spapros probesets aimed to show that signals observed in the reference scRNAseq translate to spatial data. In the two measured samples of different lung regions we showed this translatability. Each sample contains several thousand cells and therefore provide a very high effective sample size to show the within sample effects of cell type and intra-cell type variation. Further, our Immunofluorescence labelling on adjacent tissue slides showed within-donor reproducibility of the observed signal. However, with our low sample size regarding number of donors we only show the evidence that these signals do translate to spatial data, we can not make claims about the frequency of the occurrence of the observed signals over patients or show consistent reproducibility of the signal over patient populations. |
| Randomization | Not relevant to our study as no groups were compared. |
| Blinding | Not relevant to our study as no groups were compared. |

# Reporting for specific materials, systems and methods

We require information from authors about some types of materials, experimental systems and methods used in many studies. Here, indicate whether each material, system or method listed is relevant to your study. If you are not sure if a list item applies to your research, read the appropriate section before selecting a response.

## Materials & experimental systems

| n/a | Involved in the study |
|-----|----------------------|
| ☐ | ☒ Antibodies |
| ☒ | ☐ Eukaryotic cell lines |
| ☒ | ☐ Palaeontology and archaeology |
| ☒ | ☐ Animals and other organisms |
| ☐ | ☒ Human research participants |
| ☒ | ☐ Clinical data |
| ☒ | ☐ Dual use research of concern |

## Methods

| n/a | Involved in the study |
|-----|----------------------|
| ☒ | ☐ ChIP-seq |
| ☒ | ☐ Flow cytometry |
| ☒ | ☐ MRI-based neuroimaging |

# Antibodies

| | |
|---|---|
| Antibodies used | 1. MUC5AC<br>supplier name: Thermo Fisher Scientific (Invitrogen)<br>catalog number: MA5-12178<br>clone name: 45M1<br>lot number: XC3530341<br><br>2. KRT5<br>supplier name: Biolegend<br>catalog number: 905901<br>clone name: Polyclonal, Poly9059<br>lot number: B29722<br><br>3. c-FOS<br>supplier name: Novus Biologicals<br>catalog number: NBP1-89065<br>clone name: Polyclonal<br>lot number: G119139 |
| Validation | anti-MUC5AC, mouse monoclonal (Thermo Fisher Scientific, MA5-12178).  Manufacturer provides information and citation(s) regarding the species-reactivity and usage on tissue sections for immunohistochemistry/immunofluorescence.<br><br>anti-Cytokeratin 5, chicken polyclonal (Biolegend, 905901). Manufacturer provides information and citation(s) regarding the species-reactivity and usage on tissue sections for immunohistochemistry/immunofluorescence.<br><br>anti-c-FOS, rabbit, polyclonal (Novus Biologicals, NBP1-89065).  Manufacturer provides information and citation(s) regarding the species-reactivity and usage on tissue sections for immunohistochemistry/immunofluorescence. |

# Human research participants

Policy information about studies involving human research participants

| | |
|---|---|
| Population characteristics | Two samples were obtained from lungs of deceased transplant organ donors. The samples include a biopsy from the distal lung of a 28-year-old male (smoker), and a biopsy from the trachea of a 61-year-old male (non-smoker), both donors having no significant reported lung or tracheal conditions. |
| Recruitment | Samples were obtained from deceased transplant organ donors by the Cambridge Biorepository for Translational Medicine (CBTM) |
| Ethics oversight | the NRES Committee of East of England – Cambridge South (15/EE/0152) |

Note that full information on the approval of the study protocol must also be provided in the manuscript.

