## [Peer Review File · Nature Methods]

Probe set selection for targeted spatial transcriptomics

Corresponding Author: Professor Fabian Theis

A version of this paper was originally rejected for publication by Nature Methods, however that decision was reconsidered after appeal by the authors.

Version 0:

Decision Letter:

14th Oct 2022

Dear Fabian,

Your Article entitled "Probe set selection for targeted spatial transcriptomics" has now been seen by 3 reviewers, whose comments are attached. While they find your work of potential interest, they have raised serious concerns which in our view are sufficiently important that they preclude publication of the work in Nature Methods, at least in its present form.

As you will see, the reviewers raise concerns about the technical validation experiments as well as a lack of sufficient benchmarking. We agree with the referees that it will important to demonstrate the advance and novelty in the context of spatial transcriptomics data.

Should further experimental data allow you to fully address these criticisms we would be willing to look at a revised manuscript (unless, of course, something similar has by then been accepted at Nature Methods or appeared elsewhere). This includes submission or publication of a portion of this work somewhere else. We hope you understand that until we have read the revised paper in its entirety we cannot promise that it will be sent back for peer-review.

If you are interested in revising this manuscript for submission to Nature Methods in the future, please contact me to discuss your appeal before making any revisions. Otherwise, we hope that you find the reviewers' comments helpful when preparing your paper for submission elsewhere.

Sincerely,
Madhura

Madhura Mukhopadhyay, PhD
Senior Editor
Nature Methods

Reviewers' Comments:

Reviewer #1:

Remarks to the Author:

Novel contribution of the work is not clear. There are already work that aimed at capturing variance (gradient) within cell clusters.

Also unclear if it is possible or beneficial to streamline the process as a computational pipeline, as there are a wide variety of needs under various design/study scenario, many of which desire to be handled in flexible, human-intelligent ways.

Comparison with other programs were not performed comprehensively (mostly against the "basic methods"). In the 2 datasets (lung and heart) where more comparisons were performed against the state-of-the-art tools, the results do not indicate/support

a clear advance/benefit of Spapros (even under n=50 condition, n=150 is essentially a tie).

The criteria for judgement/comparison are not well reasoned.

- The criteria were tailored towards scRNA-seq data with limited consideration (e.g., spatial covariation of gene expression) towards spatial omics data, which are becoming widely available.
- Why should the “aggregate score” be the overall basis of judgement of the approaches? “The aggregated score is the average between variation recovery metrics and the first two cell type classification metrics”, why does aggregate score focus on only cell-type classification and variation recovery, both are well covered by other approaches. What about other metrics, for example “neighborhood similarity”, which is relatively novel in the context of spatial data? In Fig. S8, if the KNN column corresponds to “neighborhood similarity”, it appeared that Spapros did not outperform existing methods.
- It is not novel to demonstrate the identification of transcription factors such as FOS from within-cluster variation. Other programs such as SCMER (Nature Comp. Sci.) have already demonstrated TFs including FOS as examples (Fig. 3).
- The comparison appears unfair in that some of the methods being compared also supports preselected genes, but the DE genes was only given to Spapros.
- Is there any particular reason that the authors did not include geneBasis (ref 33) in the comparison?
- The method largely relies on clustering/annotation accuracy of the single-cell data, e.g., selection of DE genes. In Fig 2c, the left panel: the bottom right blue cluster is a complex cell type with several subclusters. With 150 genes, the work did not appear to resolve them, likely due to limited cell-type clustering/annotation quality/resolution to begin with.

At the end, about what percentage of genes are from PCA and DE, respectively? Do the percentages change significantly with different datasets? I would guess DE will take larger portion if the annotations are finer.

The longest time Spapros needed on the test data was 11h (Fig S8). What kind of machine (CPU & RAM) was used? Can the authors mention which step is the most time consuming, and do the authors envision ways (e.g., parallel computing) to accelerate it? The authors may also want to briefly discuss scalability in general.

Reviewer #2:

Remarks to the Author:

The manuscript describes the method Spapros, an end-to-end pipeline for probe selection for targeted single-cell spatial transcriptomics experiments. This is done by first identifying the technologically feasible probes combined with computationally optimal probes that preserve cell type markers, variation, and reduced redundancy. The authors demonstrate that standard algorithms for feature selection are inconsistent across priority metrics and their proposed method generally out-performs recent state-of-the-art approaches, although only slightly in some cases. Their results are supported by public data as well as an experimental validation dataset. The paper is well-written, with appropriately scoped introduction and discussion sections. Overall, the method seems useful and the package relatively user-friendly.

Major:

1. As mentioned, the technical probe selection step is currently tailored to SCRINSHOT. It would be helpful to highlight specifically which steps are most likely to be technology specific in a workflow/tutorial so that users are very aware of that.
2. The tutorial for the end-to-end pipeline incorporating the technical considerations does not exist yet.
3. Is it worth looking at the top 50 or 150 genes in Figure S8 selected by the pca or scmer methods and show how many of those genes/probes would be deemed technologically infeasible/suboptimal? Or even how many of those are ruled out due to high correlation by spapros? This could certainly help bolster the significance of the method.
4. Given the long computation time in some cases, how to you recommend using the package? Was it done on a single standard laptop? Or should it be on a HPC system? Can steps be parallelized by the user?
5. The 10X protocol for targeted spatial experiments appears to just enrich target genes for sequencing after performing an untargeted capture. Is spapros still relevant for use in that scenario?

Minor:

1. I don't see anywhere on the github on documentation pages that explicitly say the package is Python based.

Reviewer #3:

Remarks to the Author:

The study reported in the manuscript entitled “Probe set selection for targeted spatial transcriptomics” by Kuemmerle et al., introduces a novel computational method for selecting a probe set to analyze spatial transcriptomics data from different tissues. This method combines unsupervised and supervised combinatorial selection features to account for cell type identification and gene expression variation within a cell type, which resolve its spatial and temporal location within a tissue. The authors compared their method to the existing methods aiming at targeting key features within tissue and described the advantages of Spapros over these methods. To ensure that Spapros produces credible results, the authors performed validation experiments using the in situ spatial method SCRINSHOT on two tissues: the lung and the heart. Specifically, Spapros was utilized to

identify not only the lung's cellular subsets but also to account for expression variation and technical noises that emerged from the single cell data sets obtained from different studies. By utilizing their new probe set of 50 or 150 genes, the authors showed their new method's ability to capture the cell type landscape of the tissue while also measuring their expression variability in different locations, using the FOS gene as proof of concept in tracheal basal cells. Lastly, the method is well described, the code is well written, and the open-source software works fairly quickly with a friendly interface. Indeed, Spapros, a new spatial transcriptomics probe selection, fulfills the open and accessible source so that it can be adopted by the community around the world.

Spapros was evaluated by selecting probes from 6 different single cell datasets and utilizing 12 quality metrics to show its superior selection in cell type and expression variation recovery. Of note, Spapros relies heavily on single cell genomics analysis of tissues. Therefore, it might be biased toward abundant cell subsets and may be guided by false signals arising from cell dissociation. To overcome that, the authors utilized only the in situ-based SCRINSHOT assay for validation and showed FOS expression variation along the basal cell spatial organization. Is this an actual signal? Why do basal cells express FOS? Therefore there is a need to test the success rate of Spapros prediction on other in situ-based spatial datasets such as Slide-seq. This examination will reduce noise and technical issues associated with dissociation protocols used in scRNA-seq. Next, it is imperative to include basic guidelines to perform Spapros analysis, e.g., how many single-cell datasets should be used? Is it important to include at least one in situ hybridization-based dataset? In addition, the authors claim that their method is broadly applicable to different tissues. However, I was not convinced that this method is robust enough to account for cell-cell interactions from different cellular lineages (e.g., epithelial-immune and fibroblast-immune interactions). In addition, a step-by-step validation of this algorithm by showing its specificity and success rate in a controlled environment (known interactions between cells) is missing from the paper. Overall, Spapros provides an elegant solution method for choosing a targeted gene set for spatial transcriptomics by utilizing single-cell RNA-seq datasets. However, the method's robustness and validations are still missing and need more work to pinpoint the robustness of the technique in identifying cell-cell interaction signals within tissues as other non-supervised spatial transcriptomics methods.

Specific comments:

1. Cell-cell interactions might change the gene expression of the interacting cells (changing the cell state) so that cells would have a different expression profile than their singlet constituents. How does the method deal with these changes, and how can the algorithm assess this? This is one of the most challenging issues that Spapros should deal with.
2. Related to the previous comment, the strength of this method should be in trying to understand the cellular program changes occurring due to cell-cell interactions in health and, most importantly, in disease, as many of the single cell datasets report these transcriptional changes but with no spatial context.
3. The authors should also address the possibility of two similar cell subsets (same kind) interacting with another cell type (such as T-B or T-DC interactions). Could Spapros identify this?
4. The authors show FOS expression in tracheal basal cells. FOS expression may be due to low-quality single cell measurements and therefore represent technical noise expression. Could the authors utilize the protein levels to validate their results? And show a different gene with a similar variation in basal cells?
5. It will be important to show gene expression variation to other cell types from a different compartment. For example, expression variation in stromal or immune cells?
6. The authors should state more clearly the limitations of their method. It would be important to know if Spapros may not work well in deciphering cell-cell interactions. It is important to expand on this and other limitations so that researchers who adopt the Spapros method are aware of possible biases or limitations.
7. The authors should provide a guideline on how many genes are necessary to identify most cell subsets and their expression variation within a tissue.
8. Is it possible to create combinatorial gene probes that are not unique but will combine a few genes to account for cellular programs such as consensus non-negative matrix factorization (cNMF)?

** For Nature Portfolio general information and news for authors, see <http://npg.nature.com/authors>.

Version 1:

Decision Letter:

23rd Jan 2023

Dear Fabian,

Thank you for your letter asking us to reconsider our decision on your Article, "Probe set selection for targeted spatial transcriptomics". After careful consideration we have decided that we are willing to consider a revised version of your manuscript as you have outlined in your plan.

- * include a point-by-point response to our referees and to any editorial suggestions
- * please underline/highlight any additions to the text or areas with other significant changes to facilitate review of the revised manuscript
- * address the points listed described below to conform to our open science requirements
- * ensure it complies with our general format requirements as set out in our guide to authors at www.nature.com/naturemethods
- * resubmit all the necessary files electronically by using the link below to access your home page

Link Redacted

We hope to receive your revised paper within 8 weeks. If you cannot send it within this time, please let us know. In this event, we will still be happy to reconsider your paper at a later date so long as nothing similar has been accepted for publication at Nature Methods or published elsewhere.

OPEN SCIENCE REQUIREMENTS

REPORTING SUMMARY AND EDITORIAL POLICY CHECKLISTS

When revising your manuscript, please submit reporting summary and editorial policy checklists.

DATA AVAILABILITY

CODE AVAILABILITY

Please include a "Code Availability" subsection in the Online Methods which details how your custom code is made available. Only in rare cases (where code is not central to the main conclusions of the paper) is the statement "available upon request" allowed (and reasons should be specified).

MATERIALS AVAILABILITY

SUPPLEMENTARY PROTOCOL

To help facilitate reproducibility and uptake of your method, we ask you to prepare a step-by-step Supplementary Protocol for the method described in this paper. We [encourage authors to share their step-by-step experimental protocols](https://www.nature.com/nature-research/editorial-policies/reporting-standards#protocols) on a protocol sharing platform of their choice and report the protocol DOI in the reference list. Nature Portfolio's Protocol Exchange is a free-to-use and open resource for protocols; protocols deposited in Protocol Exchange are citable and can be linked from the published article. More details can found at www.nature.com/protocolexchange/about.

ORCID

Sincerely,
Madhura

Madhura Mukhopadhyay, PhD
Senior Editor
Nature Methods

Version 2:

Decision Letter:

Our ref: NMETH-A49853B

15th May 2024

Dear Fabian,

Thank you for submitting your revised manuscript "Probe set selection for targeted spatial transcriptomics" (NMETH-A49853B). It has now been seen by the original referees and their comments are below. The reviewers find that the paper has improved in revision, and therefore we'll be happy in principle to publish it in Nature Methods, pending minor revisions to satisfy the referees' final requests and to comply with our editorial and formatting guidelines.

We are now performing detailed checks on your paper and will send you a checklist detailing our editorial and formatting requirements within two weeks or so. Please do not upload the final materials and make any revisions until you receive this additional information from us. Please address the remaining reviewer concerns and include a point-by-point rebuttal with your final files.

TRANSPARENT PEER REVIEW

Please note: we allow redactions to authors' rebuttal and reviewer comments in the interest of confidentiality. If you are

concerned about the release of confidential data, please let us know specifically what information you would like to have removed. Please note that we cannot incorporate redactions for any other reasons. Reviewer names will be published in the peer review files if the reviewer signed the comments to authors, or if reviewers explicitly agree to release their name. For more information, please refer to our [FAQ page](https://www.nature.com/documents/nr-transparent-peer-review.pdf).

ORCID

Sincerely,
Madhura

Madhura Mukhopadhyay, PhD
Senior Editor
Nature Methods

Reviewer #1 (Remarks to the Author):

General remarks:

The authors clarified the novelty of their work, and it looks like the uniqueness of this work is still being an ensemble of multiple smaller steps, which remains as a concern. The authors added more benchmarking and validation datasets, which is appreciated.

Below are comments in details:

1. The updated manuscript seems to restrict the application of this method to a few sequencing-based technologies. This makes the context clearer. It would be appreciated if the authors can state more clearly about which technologies they recommend applying this method to (and which are not)
2. The authors showed an advantage of optimizing gene selection and probe design simultaneously and many genes selected by other methods does not make good targets because they cannot be well-probed. This point is well taken. The question is: is it feasible that people can filter-out these genes beforehand and only retain the ones that can be probed, and apply other methods?
3. While discussing cell type classification and variation preservation, the authors need to clarify under what circumstances each aspect is important. In many cases, even cell types cannot be well-defined.
4. The authors did not address the question regarding the requirement of clustering/annotation of the data for Sparos.
 - a. The authors discussed that Sparos cannot deal with batch effects as a limitation. However, with only a single-batch dataset, one could hardly reach high cell type resolution (given the small cell number).
 - b. What about cases that cells are in transitioning/development/tumor that are hard to annotate? In developmental data, for example, progenitor cells are possibly in different stages to differentiate but shall we annotate all as progenitor cells?

Reviewer #3 (Remarks to the Author):

The revised version of the manuscript offers novel analysis and statistical confirmations and, as such, enhances the method's credibility for the selection of specific probes to in situ-based ST. The authors have addressed my main concerns and modified their conclusions accordingly.

Overall, this study provides an important guideline for spatial analysis probe selection, which could be of interest to the broad scientific community and may help uncover additional layers of complexity within cells found in tissues under homeostasis and disease.

Version 3:

Decision Letter:

30th Sep 2024

Dear Fabian,

I am pleased to inform you that your Article, "Probe set selection for targeted spatial transcriptomics", has now been accepted for publication in Nature Methods. The received and accepted dates will be 19 Jul 2022 and 30 Sep 2024. This note is intended to let you know what to expect from us over the next month or so, and to let you know where to address any further questions.

Over the next few weeks, your paper will be copyedited to ensure that it conforms to Nature Methods style. Once your paper is typeset, you will receive an email with a link to choose the appropriate publishing options for your paper and our Author Services team will be in touch regarding any additional information that may be required. It is extremely important that you let us know now whether you will be difficult to contact over the next month. If this is the case, we ask that you send us the contact information (email, phone and fax) of someone who will be able to check the proofs and deal with any last-minute problems.

Please note that *Nature Methods* is a Transformative Journal (TJ). Authors may publish their research with us through the traditional subscription access route or make their paper immediately open access through payment of an article-processing charge (APC). Authors will not be required to make a final decision about access to their article until it has been accepted. [Find out more about Transformative Journals](https://www.springernature.com/gp/open-research/transformative-journals)

If you are active on Twitter/X, please e-mail me your and your coauthors' handles so that we may tag you when the paper is published.

Best regards,
Madhura

Madhura Mukhopadhyay, PhD
Senior Editor
Nature Methods

** Visit the Springer Nature Editorial and Publishing website at www.springernature.com/editorial-and-publishing-jobs for more information about our career opportunities. If you have any questions please click here.**

Open Access This Peer Review File is licensed under a Creative Commons Attribution 4.0 International License, which permits use, sharing, adaptation, distribution and reproduction in any medium or format, as long as you give appropriate credit to the original author(s) and the source, provide a link to the Creative Commons license, and indicate if changes were made. In cases where reviewers are anonymous, credit should be given to 'Anonymous Referee' and the source. The images or other third party material in this Peer Review File are included in the article's Creative Commons license, unless indicated otherwise in a credit line to the material. If material is not included in the article's Creative Commons license and your intended use is not permitted by statutory regulation or exceeds the permitted use, you will need to obtain permission directly from the copyright holder.

Response to the reviewers' comments of

NMETH-A49853: "Probe set selection for targeted spatial transcriptomics"

We would like to thank the editor and the reviewers for their comments and suggestions, which we believe helped to improve this manuscript considerably. In the following we give **point-by-point answers (green)** to the questions and **comments (black)** and in parts **copy old parts of the manuscript text or specific panels (blue)** and **updated parts of the manuscript text or specific panels (red)**. We numbered the comments of the editor (0.x) and of the reviewers (1.x, 2.x, 3.x) to allow for referencing between related comments.

Dear Madhura

I hope you are doing fine. Not sure you still remember our spatial probe set design ('Spapros') manuscript - first thank you for sending out the manuscript for review and getting back to us in a timely manner, in particular for allowing revision! We are grateful for the valuable reviewer comments that led to a substantial improvement of the manuscript and usability of the code base. We sincerely apologize for the long delay in our manuscript's revision: we had a cyber attack on our facilities last year shortly after revision came in, and had to recover and in part redo significant parts. On the positive side, this delay gave us the opportunity to add more material on the benchmarking side, both data sets as well as methods. We thus believe that we were able to address every reviewer comment in high detail now.

In response to the reviewers' feedback as outlined in our point-by-point appeal plan we previously sent and discussed, we have conducted a thorough series of revisions that have led to new insights, significantly expanding the breadth and depth of our study. The major new contributions and findings are summarized below:

- Firstly, by **adding a wide range of novel datasets into our benchmark (overall 13 added data sets and 4 new selection methods)**, including a matched dataset of dissociated single cells and a recent MERFISH human brain dataset, we found that our optimizations on dissociated data successfully translate to spatial data while showing clear statistical significance that Spapros is the top performing method (addressing comments 1.1, 3.1, 3.3, 3.6). This **confirms that our design strategies** are not only effective but also **yield biologically relevant spatial signals**. Specifically, we constructed new spatial metrics and discovered gene signals indicative of cell sub-states related to cell-cell interactions, as well as gene signals that manifest distinct spatial patterns. We added an entirely new figure in the main text, which provides an in-depth comparison of the methods.
- Secondly, **utilizing additional validations through newly generated immunofluorescence (IF) measurements**, we reinforced the presence of intra-cell type variation relating to spatial patterns (addressing comments 3.4, 3.5). Our SCRINSHOT data served as the basis for showcasing additional genes that exhibit spatially patterned intra-cell type variation.
- Finally, we have made **several robust enhancements to Spapros' code base, and usability guidance**. Importantly, we introduced a bootstrapping approach in Spapros' benchmark pipeline to robustly assess statistical significance. Further efforts were directed toward measuring performance metrics that focus on scalability and usability (comments 1.8 and 2.4), updating our usage guidelines and code tutorials (comments 2.1, 2.2, 2.5, 3.2, and 3.8), and clarifying previously ambiguous sections of the manuscript (comments 1.2, 1.3, 1.4, 1.6, and 3.2), further we clarified the limitations of our method based on extensive analyses (comments 3.2, 3.6, 3.7).
- As Reviewer 1 raised concerns on the novelty of our work we further compiled a **thorough comparison table of our probe set selection method Spapros with classical gene set selection methods**, which clearly highlights the novelty of our work (see answer to remarks of Reviewer 1). While our method already entailed the unique dual optimisation objective and probe design, the comments of the reviewers lead to novel analyses that distinguishes our work further by showing the **significance of our probe set selection objectives** for capturing spatially relevant signals.

Since the initial submission of our manuscript, Spapros has been employed in multiple internal and external consortia projects. The constructive feedback gleaned from these practical applications has led to several enhancements not explicitly raised in the reviewer comments. Specifically, we have **optimized the memory usage and computation time** of our selection algorithm, **introduced tutorials** that guide the refinement of pre-existing gene panels using our evaluation suite, incorporated an additional Snakemake pipeline for **enhanced reproducibility** in large-scale benchmark experiments, and added analyses based on a streamlined version of Spapros focused solely on cell type identification rather than broader variation recovery (referred to as *SpaprosCTo*).

We feel that we were able to take the feedback on board and address all issues in a revised manuscript and want to thank you and the reviewers again for the detailed review of our initial manuscript. With the exponentially increasing excitement and use of recent targeted spatial omics methods - in particular commercial ones such as Xenium and MerScope - together with available organ atlases, the need for informed panel design as we offer with Spapros is a key experimental design step in any such analysis, and we hope you continue to share our enthusiasm for the revised approach.

Best regards,
Fabian

Editor comments:

Your Article entitled "Probe set selection for targeted spatial transcriptomics" has now been seen by 3 reviewers, whose comments are attached. While they find your work of potential interest, they have raised serious concerns which in our view are sufficiently important that they preclude publication of the work in Nature Methods, at least in its present form.

0.1 As you will see, the reviewers raise concerns about the technical validation experiments as well as a lack of sufficient benchmarking. We agree with the referees that it will be important to demonstrate the advance and novelty in the context of spatial transcriptomics data.

As summarized above we added 13 new datasets, 2 new metrics on spatial data, 4 selection methods, and tested for significance of metric differences using a bootstrapping approach to: 1) extend our technical validations towards spatially pertinent signals and 2) show clear statistical significance of Spapros' improved performance in the method benchmark. See detailed answers below (remarks of reviewer 1 and comments 1.1, 1.2, 1.5, 3.1).

To demonstrate the advance and novelty of our method we added a new validation scenario based on matched public human brain snRNA-seq and MERFISH data showing that Spapros' novel, combined optimisation objectives correctly translate to spatial variation (comments 1.1, 3.1, 3.3, 3.6). Further we validated our intra-cell type variation signal with a novel IF dataset (comments 1.3, 3.4). Lastly, we showcase the performance increase of our novel joint consideration of gene panel selection and probe design for the probe set selection process of spatial experiments (comment 2.3).

0.2 Should further experimental data allow you to fully address these criticisms we would be willing to look at a revised manuscript (unless, of course, something similar has by then been accepted at Nature Methods or appeared elsewhere). This includes submission or publication of a portion of this work somewhere else. We hope you understand that until we have read the revised paper in its entirety we cannot promise that it will be sent back for peer-review.

With the newly added MERFISH and newly generated IF data, as well as further analysis of our newly generated SCRINSHOT data and extending our benchmark with 10 additional public scRNA-seq datasets, we believe that we could fully address the criticisms of the reviewers.

If you are interested in revising this manuscript for submission to Nature Methods in the future, please contact me to discuss your appeal before making any revisions. Otherwise, we hope that you find the reviewers' comments helpful when preparing your paper for submission elsewhere.

As outlined in our appeal, we have revised our manuscript accordingly. Our revision process not only followed the detailed plan we proposed but also went beyond it. Driven by the insightful comments from the reviewers, we undertook additional analyses that have added considerable depth and value to our work. We are confident that these enhancements have notably improved the manuscript.

Reviewers' comments:

Reviewer #1:

Remarks to the Author:

Novel contribution of the work is not clear. There are already work that aimed at capturing variance (gradient) within cell clusters. Also unclear if it is possible or beneficial to streamline the process as a computational pipeline, as there are a wide variety of needs under various design/study scenario, many of which desire to be handled in flexible, human-intelligent ways. Comparison with other programs were not performed comprehensively (mostly against the "basic methods"). In the 2 datasets (lung and heart) where more comparisons were performed against the state-of-the-art tools, the results do not indicate/support a clear advance/benefit of Spapros (even under n=50 condition, n=150 is essentially a tie). The criteria for judgement/comparison are not well reasoned.

We thank the reviewer for the critical evaluation and constructive comments below. We highly appreciate highlighting our lack of clarity regarding the novelty of our work. The two key unique novelties of our approach are that (1) we **perform probe set selection and probe design simultaneously**, which is necessary for a combinatorial probe set selection, and (2) we **for the first time formulate probe set selection as a trade-off** - optimizing for a combination of both recovery of cell types and spatial variation signals beyond cell types. None of the methods the reviewers mentioned, or indeed any of the methods we have compared Spapros against, perform either of these tasks. We extended our previous method comparison and added a new main text figure to adequately demonstrate the novelties of Spapros in contrast to recent related work. Furthermore, with the additional analyses conducted for the revised manuscript **we now show that optimizing for the metrics on dissociated data translates to biologically relevant spatial signals** which has not been done in such detail before. In total, we have clarified the novelty of Spapros and documented its improved performance in the following ways:

- We added new analyses and according text paragraphs to the benchmark results section, showing:
 - Spapros' performance significance over other methods based on a bootstrap sampling approach on **12 datasets (instead of previously 2)** (see details below),
 - the **unique positioning of Spapros** in the performance pareto front of the metrics variation recovery and cell type classification (see comment 1.2),
 - the importance of taking into account the probe design (see comment 2.3),
 - Spapros showing top performance on recovering biologically relevant states associated with cell-cell interactions (see comments 1.1 & 3.1).
- In the results section on quantifying optimal probe set selection we now discuss that optimisation of variation recovery translates to recovering spatially patterned variation based on a new analysis (see comment 1.1)
- Finally, to give an overview of the novelty of our work, we provide an additional supplementary comparison table (see below, table A7).

We thank the reviewer for highlighting the limitations of our previous comparison. This motivated us to do better, and we have now extended this comparison to a more robust and convincing benchmark. On top of the previous 2 datasets we now added 10 more datasets to show that Spapros indeed shows significantly higher performance

in both cases $n = 50$ and $n = 150$. Besides adding more datasets we further increased the statistical robustness by measuring performance distributions per dataset with a bootstrap sampling approach. Accordingly, we adjusted the benchmark results section with the following text:

[...] All methods were run on lung and heart datasets [...]. We find that Spapros is consistently the top performing method while the ranking of other methods varies among the different datasets and panel sizes (Figs 4a,b,d, S9a, S10a, S11a).

To provide context for the observed differences in metric scores, we assessed the statistical significance of performance differences between top-performers from this benchmark using a bootstrapping approach across 12 datasets (Figs 4c, S11b; see Methods). We find that Spapros shows significantly higher performance on our aggregated score for 50 and 150 genes compared to all other methods (except for $n=50$, no significant difference to geneBasis). SpaprosCTo performs best in cell type classification, followed by Spapros and NS-Forest (Fig. S11b), especially for low numbers of genes (Figs 4a, S9a, S11). In contrast, as expected, the variation recovery methods (selfE, SCMER, PCA, geneBasis) achieve the highest variation recovery metric scores (Fig. S11b). However, this does not translate to a significant performance difference to Spapros. Taken together, when investigating pareto-optimal method performance in terms of cell type classification and variation recovery, Spapros has a unique position on the pareto front (Figs 4d, S10a). In contrast, other cell type identification methods do not optimize for variation recovery, and other variation recovery methods do not reach the cell type classification performance of Spapros.

The extended benchmark shows the same trends as our previous results, but we agree that the statistical significance over multiple datasets was important to make this analysis more robust. The performance hierarchy between methods and the statistical significance is shown in the newly added Figure S12b:

Panel b from Extended Data Figure 11:

Fig. S11: Method benchmark significance tables. a, [...] comparing cell type recovery, variation recovery, and the aggregated score of the different selection methods [...]. Methods are ranked by mean performance. b, Paired t-test P-values for the mean scores of the same metrics across 12 datasets on 50- and 150-gene selections.

As mentioned above to further clarify the novelty of our work we now also provide an overall comparison of recent methods with a new supplementary comparison table:

Method	general variation recovery	cell type identification	probe design	evaluation tool-box	spatial robustness constraints	incorporating prior knowledge	validation on spatial datasets	validating spatial pattern recovery	validating CCI recovery	new spatial dataset
Spapros	orange	orange	red	red	orange	red	red	red	red	red
scGeneFit	orange	orange	red	red	orange	red	red	red	red	red
geneBasis	orange	orange	red	orange	orange	red	orange	red	red	red
scPNMF	orange	orange	red	red	orange	red	red	red	red	red
selfE	orange	orange	red	red	orange	red	red	red	red	red
SCMER	orange	orange	red	red	orange	red	red	red	red	red
ASFS/activeSVM	orange	orange	red	red	orange	red	orange	red	red	red
Triku	orange	orange	red	red	orange	red	red	red	red	red
NS-Forest	orange	orange	red	red	orange	red	red	red	red	red
SMaSH	orange	orange	red	orange	orange	red	red	red	red	red
PERSIST	orange	orange	red	red	orange	red	orange	red	red	red
PERSISTus	orange	orange	red	red	orange	red	orange	red	red	red
COSG	orange	orange	red	red	orange	red	orange	red	red	red

Color code:

- green = supported/high performance
- orange = somewhat supported/medium performance
- red = not supported/low performance

Header explanations:

general variation recovery:	Selection of genes for general variation recovery (based on method description and our benchmark results)
cell type identification:	Selection of genes for cell type classification of a defined set of cell types (based on method description and our benchmark results)
probe design:	Incorporation of probe design, enabling combinatorial gene panel selection due to probe design constraint
evaluation tool-box:	Code toolbox/functions for evaluating newly generated gene panels
spatial robustness constraints:	Whether the method incorporates options/features to select signals that translate to spatial technologies
Incorporating prior knowledge:	Whether the method incorporates options to account for prior knowledge (e.g. pre selected genes, or marker lists)
validation on spatial datasets:	Gene sets were evaluated on spatial data in the publication of the method (note: orange if the gene set was not selected on a dissociated ref)
validating spatial pattern recovery:	Spatial validation included the validation whether the method's optimisation objective leads to increased recovery of genes exhibiting spatial p
validating CCI recovery:	Spatial validation included the validation whether the method's optimisation objective leads to increased recovery of cell-cell interaction assoc
new spatial dataset:	A new spatial dataset was measured in the publication of the method

New Table A7: Comparison of probe set selection methods

Major comments:

1.1 The criteria were tailored towards scRNA-seq data with limited consideration (e.g., spatial covariation of gene expression) towards spatial omics data, which are becoming widely available.

While previous validations on our spatial SCRINSHOT dataset indicate that scRNA-seq signals do translate to spatial data, we appreciate that additional evaluations using spatial data would bolster this argument. Therefore we now selected gene subsets of a public MERFISH human brain dataset of 4000 genes on a matched snRNA-seq dataset and evaluated to assess how our metrics translate across modalities. Furthermore, we performed spatial autocorrelation analysis to investigate how strongly the selected genes show informative spatial patterning, and assessed how well cell-cell interactions (CCIs) are captured with a selected gene panel.

Panels b and c from Extended Data Figure 2:

Fig. S2: Variation recovery metrics for different granularity levels and translation to spatial variation. [...]. **b**, Correlation between performance metrics on dissociated and spatial data based on matched snRNA-seq and MERFISH human brain data. **c**, Correlation between spatial variation metric on the MERFISH data and fine clustering similarity on the snRNA-seq data.

Panel e from main text Figure 4:

Fig. 4: Spapros outperforms classical selection strategies and state-of-the-art methods. [...]. **e**, Correlation between variation recovery scores on dissociated data and CCI recovery on spatial data, using matched snRNA-seq and MERFISH data from the human brain. [...]

We find a high correlation between our metrics on snRNA-seq and on spatial data, as well as high correlation between the variation recovery metrics on snRNA-seq and our new spatial metrics (spatial autocorrelation and CCI recovery). The strongest correlation to spatial autocorrelation is observed for the variation recovery metric “fine clustering similarity” which shows that optimizing for general variation recovery (especially more fine-grained variation beyond cell type annotations) improves the likelihood of selecting genes with spatial patterns. For CCI recovery we find the highest correlation to the aggregated variation recovery score. We described these findings in the main text as follows:

We find that cell type identification and variation recovery metrics translate well to matched spatial data ($r = 0.67$ and 0.68 ; see Methods; Fig. S2b). The optimization of fine level variation recovery on the dissociated reference is highly correlated with spatial variation recovery in spatial measurements ($r = 0.79$) (see Methods; Fig. S2c).

Overall, we find a high correlation between variation recovery and CCI recovery ($r = 0.85$), with Spapros among the top performers (Fig. 4e).

While we observe these general trends we also acknowledge that there is no perfect correlation between the performance on the snRNA-seq reference and on MERFISH. The deviation is most prominent for the cell type classification score of the method PERSIST. We investigated this effect with additional experiments, and contextualized it with respect to the recent literature on data quality of spatial transcriptomics technologies. PERSIST applies binarization on count data to achieve robustness for translation to spatial data. Applying the same constraint to Spapros selections lead to even slightly higher cell type classification performance on MERFISH. Such robustness constraints are beneficial for lower quality spatial datasets. We previously highlighted these issues in our discussion:

There are multiple reasons that lead to discrepancies between the modalities. [...] technology-specific effects like optical crowding can result in imprecise measurements. Quantifying these differences and constructing additional robustness constraints are important future directions for probe set selection and reference based down stream analysis. [...] To design ideal robustness constraints future studies need to investigate these effects as well as tissue parameters like organ type, cell density, and tissue quality.

While it is out of scope of our manuscript to provide a full analysis of data quality differences between spatial technologies, recent literature gives additional insights. We discussed the robustness constraints and data quality in an additional Supplementary Note 1. We reference the Supplementary Note 1 in the main text discussion as follows:

Thus, we find that the number of genes needed to reach the same cell type classification robustness can be substantially higher in data from older, high-plex spatial technologies compared to the dissociated reference (Fig. S12b). This has been reported to improve with newer spatial technologies such as Xenium^{40,41}. We also find that spatial robustness constraints like PERSIST’s binarization of counts can enhance performance on such spatial data (see Supplementary Note 1).

Supplementary Note 1:

[...] Based on our evaluation experiments on matched snRNA-seq and MERFISH data we find the general trend that gene sets optimized for variation recovery on dissociated data show increased capture of spatially relevant variation (Figs S2b,c, 4e).

Despite the correlation between dissociated and spatial metrics our validation experiment suggests that additional spatial robustness constraints during selections can improve performance on the spatial data. We find that two methods show improved translation of signals to spatial data compared to the expectation from dissociated metrics. Specifically, PERSIST and scPNMF achieve the highest scores in cell type classification and spatial variation on the MERFISH data respectively, despite not being top performers on scRNA-seq (Fig. S2c,b). These methods account for the challenges in translation between modalities with specific preprocessing and filtering steps. For PERSIST, this is enabled by optimizing on binarized count data, and for scPNMF, by filtering genes based on correlation with library size and non-unimodality of gene score distributions. Indeed, when applying binarization to Spapros selections, we achieve even slightly higher cell type classification performance on the MERFISH data compared to PERSIST (Supplementary Note Fig. 1a).

Our experiments and recent literature show that spatial robustness constraints are beneficial for current low quality high-plex technologies and less important for other technologies. While the additional robustness constraints enhance translatability to spatial data, it is crucial to consider the extent to which our spatial validation generalizes. The scarcity of high-plex spatial datasets (> 3k genes) limits the generalizability of our validation. Our reliance on a single high-plex MERFISH dataset means our observations are more indicative of trends than definitive rankings. Additionally, and more importantly, recent studies have shown significant variability in the quality of spatial measurements^{40,41,75}. These studies suggest that current high-plex solutions (MERFISH, CosMx, STARmap PLUS) exhibit fewer counts per gene and cell, and increased nonspecific signals for individual genes, compared to the higher quality, currently lower-plex methods (MERSCOPE, Xenium). Specifically, as demonstrated with our SCRINSHOT data, individual selected marker genes can effectively recover targeted cell types (Fig. 3a). This was similarly observed in Cook et al.⁴¹, where cell types within delicate spatial structures were recoverable with single marker genes using Xenium, but challenging with high-plex CosMx measurements. Wang et al.⁴⁰ also reported similar findings, with more cell types identified across multiple tissues using lower-plex Xenium compared to higher-plex CosMx. These studies provide strong evidence that with the latest advancements in spatial technologies, there is improved translatability between scRNA-seq and spatial data. Taken together, these findings suggest that for panel designs of current high-plex technologies like MERFISH and CosMx, robustness constraints used in methods like PERSIST and scPNMF can be beneficial, while for lower-plex solutions like Xenium and MERSCOPE, better translatability is expected.

Supplementary Note Fig. 1: Translation of metrics to spatial data including Spapros on binarised counts.
a, Correlation between performance metrics on dissociated and spatial data based on matched snRNA-seq and MERFISH human brain data. **b**, Correlation between spatial variation metric on the MERFISH data and fine clustering similarity on the snRNA-seq data.

1.2 Why should the “aggregate score” be the overall basis of judgement of the approaches? “The aggregated score is the average between variation recovery metrics and the first two cell type classification metrics”, why does aggregate score focus on only cell-type classification and variation recovery, both are well covered by other approaches. What about other metrics, for example “neighborhood similarity”, which is relatively novel in the context of spatial data? In Fig. S8, if the KNN column corresponds to “neighborhood similarity”, it appeared that Spapros did not outperform existing methods.

We now clarified in the main text that the “neighborhood similarity” metric is one of the variation recovery metrics and is therefore included in the aggregated score:

[...] neighborhood similarity, which measures how well the local cell neighborhoods of the gene expression knn-graph are preserved. [...] The overall performance of a probe set is then computed as the average of the dissociated variation recovery metrics coarse/fine clustering similarity and neighborhood similarity and the cell type identification metrics classification accuracy and percentage of captured cell types as these are the main objectives we want to optimize for.

Spapros aims to optimize both variation recovery and cell type classification - as we argue and show two key properties for optimal probe selection. All relevant metrics to measure this dual objective are included in the aggregated score. Even though other methods typically perform well on either variation recovery or cell type classification, Spapros typically reaches the same or a higher score on the individual metrics and outperforms them on the aggregated score as we now showed with higher statistical robustness with our large benchmark extension (see remarks to author).

Of course the two general approaches of optimal cell type recovery vs variation recovery are conflicting i.e. maximizing the one will decrease the other given a fixed budget (i.e. the number of genes to select). Hence we aggregated the two scores. We now show the two scores separately across the various examples, Moreover we show the Pareto front of the two scores to visualize this more detailed method comparison and to avoid this singular weight selection at the start as rightfully criticized by the reviewer. We point to it in the main text:

[...] Taken together, when investigating pareto-optimal method performance in terms of cell type classification and variation recovery, Spapros has a unique position on the pareto front (Figs 4d, S10a). In contrast, other cell type identification methods do not optimize for variation recovery, and other variation recovery methods do not reach the cell type classification performance of Spapros.

Panel d of main text Figure 4

Fig. 4: Spapros outperforms classical selection strategies and state-of-the-art methods. a, [...] d, Pareto front, showing the trade-off between variation recovery and cell type classification scores for 50-gene selections from the Madisson2020 lung data. e, [...].

Panel a of Extended Data Figure 10

Fig. S10: Uniqueness of Spapros on balancing performance metrics and selecting probeable genes. a, Pareto fronts, showing the trade-off between variation recovery and cell type classification scores for 50- and 150-gene selections from the Madisson2020 lung and Litvinukova2020 heart data. b, [...].

1.3 It is not novel to demonstrate the identification of transcription factors such as FOS from within-cluster variation. Other programs such as SCMER (Nature Comp. Sci.) have already demonstrated TFs including FOS as examples (Fig. 3).

As the reviewer suggests, a method that focuses on variation recovery should ideally capture transcription factors such as FOS. We appreciate that SCMER, which also performs well at variation recovery in our benchmark, also selects FOS. The analysis of the FOS signal in the manuscript has the goal of validating that selecting continuous signals from dissociated data (as SCMER also does) is relevant for the analysis of spatial datasets rather than only focusing on cell type markers. FOS variation is indeed often overlooked as it has otherwise been associated with dissociation induced stress of scRNA-seq. To strengthen our analysis on showing the relevance of selecting FOS, we **now additionally validated the transcriptomic intra-cell type variation shown in SCRINSHOT with matched protein measurements.** (see answer to comment 3.4).

Our comparison with SCMER and other variation recovery methods is separate from this analysis. In that comparison we find that while SCMER and similar methods optimize well for variation recovery, they do not reach the same performance of cell type classification as Spapros.

1.4 The comparison appears unfair in that some of the methods being compared also supports preselected genes, but the DE genes was only given to Spapros.

The construction of different DE gene selections within the Spapros decision tree optimization are an inherent part of our method. No prefixed (“DE”) genes were given as a pre-selection in our comparisons. We clarified this in the main text as follows:

To ensure that all **user-defined** cell types can be identified, Spapros compares the classification performance for each cell type to the performance of **reference** trees. **These trees** are generated via a custom approach that

iteratively optimizes for classifying similar cell identities. In each iteration Spapros performs DE selections on critical cell type subsets and retrains the trees on the extended gene pool.

Instead, Spapros takes cell type annotations as input, which is also the case for 7 other methods which we compare Spapros against. Of course, this is not the case for fully unsupervised approaches.

1.5 Is there any particular reason that the authors did not include geneBasis (ref 33) in the comparison?

After unsuccessful trials to implement geneBasis on our HPC cluster - needed for the extensive comparisons - for more than 1 week, we initially decided to not include the method in our benchmark due to usability issues. We now added geneBasis to our comparison, which was enabled through the recent support of singularity containers on our HPC cluster. Its performance profile is similar to variation recovery methods (SCMER, PCA, SelfE). Moreover, we also added the methods scPNMF¹ and PERSIST². scPNMF exhibits lower variation recovery performance compared to top performers. PERSIST has a supervised version (cell type classification) and an unsupervised version (PERSISTus, gene expression reconstruction), both showing lower performance compared to top performers. To put the effect of Spapros' PCA preselection into context we also included a simplified version "SpaprosCTo" as an additional method which optimizes for cell type classification only, by selecting genes from the DE trees instead of the PCA trees. We included the new methods in the updated descriptions of the benchmark results and throughout the benchmark related figures (Figs 4, S9, S10, S11). Spapros still shows the highest performance after extending our benchmark with the additional methods. The previously shown benchmark tables were updated to:

a

selection method	aggr. score	variation recovery			cell type classification		probe redundancy		comp. time
		coarse clustering similarity	fine clustering similarity	neighborhood overlap	cell type classif. acc.	perc. of captured cell types	gene correlation	perc. of highly correlated genes	
spapros*	0.67	0.80	0.73	0.10	0.86	0.75	0.83	0.84	1 h
spaproscto*	0.67	0.78	0.67	0.06	0.87	0.79	0.88	0.86	56 min
DE*	0.65	0.79	0.66	0.06	0.86	0.74	0.88	0.77	2 min
selfe	0.64	0.79	0.74	0.18	0.82	0.61	0.85	0.96	51 min
nsforest*	0.64	0.75	0.63	0.04	0.86	0.75	0.93	0.87	1 h
scgeneft*	0.63	0.78	0.71	0.10	0.82	0.63	0.77	0.70	3 h
scmer	0.63	0.78	0.73	0.15	0.81	0.60	0.80	0.77	2 h
genebasis	0.62	0.78	0.74	0.17	0.81	0.55	0.88	0.97	3 h
persist*	0.60	0.73	0.64	0.04	0.82	0.62	0.90	0.98	6 h
asfs*	0.59	0.73	0.62	0.05	0.82	0.61	0.89	0.99	23 h
pca	0.59	0.73	0.70	0.17	0.78	0.49	0.82	0.79	16 sec
scpnmf	0.53	0.71	0.64	0.11	0.75	0.42	0.84	0.88	57 min
persistus	0.53	0.70	0.62	0.05	0.77	0.42	0.92	1.00	1 d
cosg*	0.30	0.27	0.40	0.01	0.60	0.15	0.97	0.99	3 sec
smash*	0.10	0.05	0.13	0.00	0.26	0.01	0.99	1.00	16 h
triku	0.06	0.03	0.03	0.00	0.13	0.08	0.60	1.00	4 min

* used cell type annotations for selection

Panel a of main text Figure 4

Fig. 4: Spapros outperforms classical selection strategies and state-of-the-art methods. **a**, Table showing mean performances of Spapros and other methods, based on 20 bootstrap samples for selecting 50 genes from the Madisson2020 lung dataset. Methods that use cell type information are annotated with a red star. [...]

Panel a of Extended Data Figure 9

Fig. S9: Spapros outperforms classical selection strategies and state-of-the-art methods. a. Heatmap of our evaluation metrics comparing Spapros with recently published methods as well as DE, and PCA-based selections. We compared selections of 50 and 150 genes for lung and heart data sets. Methods are sorted and

ranked by the aggregated score of variation recovery and cell type classification. Methods that use cell type information are annotated with a red star. **b** [...]

1.6 The method largely relies on clustering/annotation accuracy of the single-cell data, e.g., selection of DE genes. In Fig 2c, the left panel: the bottom right blue cluster is a complex cell type with several subclusters. With 150 genes, the work did not appear to resolve them, likely due to limited cell-type clustering/annotation quality/resolution to begin with.

Below, we added a figure that shows that the substructure of the AT2 cells that the reviewer refers to are batch effects and in the case of 50 selected genes Spapros prioritizes genes not affected by this batch. Some batch effects are still recovered in the case of 150 selected genes. Spapros - as most other single-metric methods - currently does not support a batch aware selection, we argue about this limitation in the discussion as follows:

[...] integrated reference atlases [...] also pose additional methodological challenges such as batch effects [...] batch effects are still a challenge for Spapros and other probe selection methods. Especially selection methods that optimize for variation recovery will also select genes aligned to batch effect variation. While we showed that Spapros can select probe sets and project these across datasets, we did find that also dataset-specific variation (potentially due to batch effects) was captured.

The second row of panels in Figure R1 shows transferred cell type annotations from the recently published human lung cell atlas³ (HLCA) using HLCA's finest level of annotation. Also these high-resolution annotations do not resolve the AT2 cluster into further cell types/states.

Further, the figure shows a comparison with the SCMER method, which is specifically designed for reconstructing the data structure in an unbiased fashion. Also SCMER does not resolve this substructure better than Spapros for a given number of genes.

Fig. R1: Fine level cell type annotations and batch variation within AT1, AT2, and ciliated cells in the Madisson2020 lung data. Included are UMAPs for 8000 reference genes and 50-/150-genes selections of Spapros, SpaprosCTo, and SCMER. The colors show the original annotations from Madisson2020, the finest annotation level of the HLCA, and the sample information.

1.7 At the end, about what percentage of genes are from PCA and DE, respectively? Do the percentages change significantly with different datasets? I would guess DE will take larger portion if the annotations are finer.

This is an excellent point - the main determining factor of the PCA-DE ratio between different datasets is indeed the granularity of the cell type annotations as the reviewer suggests. Therefore, to systematically investigate the PCA-DE genes ratio, we selected probe sets based on the five different levels of annotations of the human lung cell atlas³. As expected the percentage of DE genes increases for finer cell type annotations. Also the percentage of DE genes is higher when fewer genes are selected. We added a percentage plot as a supplementary figure to demonstrate this characteristic of Spapros. We refer to it in the main text when introducing the SpaprosCTo version that neglects the PCA pre-selection:

[...] SpaprosCTo, which exclusively utilizes DE trees for selection. This contrasts with the standard approach, where the majority of selected genes originate from the PCA-based selection (Fig S13d).

Panel d of Extended Data Figure 13

Fig. S13: Technical aspects of Spapros selections. a [...] d, Ratio between selected genes originating from PCA selections or DE trees in Spapros selections. Genes were selected on the human lung cell atlas with different cell type annotation levels.

1.8 The longest time Spapros needed on the test data was 11h (Fig S8). What kind of machine (CPU & RAM) was used? Can the authors mention which step is the most time consuming, and do the authors envision ways (e.g., parallel computing) to accelerate it? The authors may also want to briefly discuss scalability in general.

The implementation of Spapros in our python package already includes parallelization over cpu cores. However, we encountered a memory leak that slowed down the selection. Upon fixing this bug we measured a significant speed up and improved memory usage.

We added details about the machine specifications in the methods section: **For all selections 12 CPU cores and 64 GB of memory were allocated.**

The main factors that affect time and memory usage of Spapros is firstly the number of cell types, and secondly the number of cells. We ran multiple selections to measure time and memory usage under different dataset conditions for the individual steps. The DE trees training is the most time consuming step as it entails an iterative optimization step, the step is followed by the optimization of PCA trees, and lastly PCA. We added the new results as a supplementary figure, and referred to them in the main text discussing scalability and providing usage recommendations:

[...]. Spapros shows comparable computation times as other fast methods with high performances on our evaluation scores (Figs 4a, S9a). Importantly, in contrast to methods like SelfE, ASFS, and geneBasis, it also maintains constant time consumption for varying gene set size. Instead, Spapros scales linearly with the number of cell types in both memory and time (Fig. S13). Thus, Spapros can be run locally, especially for selections on coarse cell type annotations.

C
Panels a,b,c of Extended Data Figure 13

Fig. S13: Technical aspects of Spapros selections. a, Computation time and b, memory of Spapros selections for datasets with different numbers of cell types and cells per cell type. c, Computation time of different steps in the Spapros gene set selection. d [...].

Reviewer #2:

Remarks to the Author:

The manuscript describes the method Spapros, an end-to-end pipeline for probe selection for targeted single-cell spatial transcriptomics experiments. This is done by first identifying the technologically feasible probes combined with computationally optimal probes that preserve cell type markers, variation, and reduced redundancy. The authors demonstrate that standard algorithms for feature selection are inconsistent across priority metrics and their proposed method generally out-performs recent state-of-the-art approaches, although only slightly in some cases. Their results are supported by public data as well as an experimental validation dataset. The paper is well-written, with appropriately scoped introduction and discussion sections. Overall, the method seems useful and the package relatively user-friendly.

We thank the reviewer for the summary and positive evaluation of the text, our selection approach, and our code base, as well as for comments below.

Major comments:

2.1 As mentioned, the technical probe selection step is currently tailored to SCRINSHOT. It would be helpful to highlight specifically which steps are most likely to be technology specific in a workflow/tutorial so that users are very aware of that.

We agree that with the recent popularity of related assays, this needed to be extended. While Spapros can be run without the probe design constraint filter for all targeted spatial technologies, we aim to support the full selection pipeline for all popular technologies. We now expanded our probe design pipeline to also support the technologies SeqFISH⁴ and MERFISH⁵ besides SCRINSHOT. Further we provide a supplementary table in the manuscript **A6 Comparison of probe design specifications of different technologies** which compares the features of the three design pipelines in detail. We linked that table in our probe design related code tutorials so that users are pointed to it in both the code documentation and the manuscript. We did not include the table here, as it is too large to be displayed properly. Further we adjusted the probe design method section to capture the three technologies **SCRINSHOT, MERFISH and SeqFISH+**.

Based on the recently published design details of 10x Xenium probes we are currently implementing a pipeline configuration to support that technology. Further we plan to extend the probe design pipeline to other recent in situ based methods (Nanostring CosMx, Resolve Bioscience Molecular Cartography) as soon as the relevant companies publish details on their probe design requirements. The probe design package has a highly modular set up which enables easy extension to technology specific pipelines that differ slightly from already supported technologies.

2.2 The tutorial for the end-to-end pipeline incorporating the technical considerations does not exist yet.

We now added two versions of the end-to-end selection pipeline tutorial:

- A short tutorial that captures the minimal commands to run the pipeline with the technology specific default parameters
(https://spapros.readthedocs.io/en/latest/tutorials/spapros_tutorial_end_to_end_selection_short.html)
- and a second tutorial which provides a detailed walk through the different steps of the design pipeline including the description of the parameters in each step.
(https://spapros.readthedocs.io/en/latest/tutorials/spapros_tutorial_end_to_end_selection.html)

2.3 Is it worth looking at the top 50 or 150 genes in Figure S8 selected by the pca or scmer methods and show how many of those genes/probes would be deemed technologically infeasible/suboptimal? Or even how many of those are ruled out due to high correlation by spaprose? This could certainly help bolster the significance of the method.

Good suggestion - we added a graph that shows for each method the percentage of “selected genes with probes” and further conducted comparative evaluations of gene sets after filtering genes that do not pass the probe design constraint filter. We adjusted the main text to point out the importance of the probe design filter, a key novelty of our approach:

[...] experimental design for targeted spatial transcriptomics involves both selecting optimal gene sets and designing probes for these gene sets. Yet, beside Spapros, no other method considers the technical probe design constraints in the selection, leading to a significant number of genes lacking adequate probes (Fig. S10b). Removing genes for which probes cannot be designed from the selected probe pool for each method, significantly reduces the performance of these methods (Fig. 4f) and again requires manual adaptation of the probe set leading to non-optimized experimental designs.

Interestingly, the important genes that are selected can show significantly different proportions of feasible probe candidates in different tissue types. For well performing methods we find proportions of ~0.6 - 0.8 for lung and ~0.95 for heart data.

Panel b of Extended Data Figure 10

Fig. S10: Uniqueness of Spapros on balancing performance metrics and selecting probeable genes. a, [...] selections from the Madisson2020 lung and Litvinukova2020 heart data. **b,** Proportion of genes that pass the SCRINSHOT probe design constraints for the same datasets.

Panel f of main text Figure 4

Fig. 4: Spapros outperforms classical selection strategies and state-of-the-art methods. a, [...] f, Performance benefit of probe design constraint: Comparison of the aggregated scores for different methods after excluding genes failing probe design criteria, using 50-gene selections from the Madissoon2020 data.

2.4 Given the long computation time in some cases, how to you recommend using the package? Was it done on a single standard laptop? Or should it be on a HPC system? Can steps be parallelized by the user?

Spapros' gene panel selection time and memory usage mainly increase with the number of cell type clusters in the data set. For selections on coarse cell type annotations computations can easily be run locally. For finer cell type annotations we recommend single nodes of HPC systems. For fixed computational resources Spapros shows similar selection times as other fast high performing methods. We show this in our method comparison on a dataset with fine cell type annotations in Fig 4a (see below). The full genome probe design of Spapros requires high memory usage and also takes longer than the panel selection. We therefore recommend running the design on HPC systems, at least if many genes are included. All steps of Spapros as well as the evaluation suite leverage multi processing, therefore larger systems are always preferable. We added multiple supplementary figures and usage recommendations in the manuscript, please see comment 1.8.

a

selection method	aggr. score	variation recovery			cell type classification		probe redundancy		comp. time
	score	coarse clustering similarity	fine clustering similarity	neighborhood overlap	cell type classif. acc.	perc. of captured cell types	gene correlation	perc. of highly correlated genes	run time
spapros*	0.67	0.80	0.73	0.10	0.86	0.75	0.83	0.84	1 h
spaproscto*	0.67	0.78	0.67	0.06	0.87	0.79	0.88	0.86	56 min
DE*	0.65	0.79	0.66	0.06	0.86	0.74	0.88	0.77	2 min
selfe	0.64	0.79	0.74	0.18	0.82	0.61	0.85	0.96	51 min
nsforest*	0.64	0.75	0.63	0.04	0.86	0.75	0.93	0.87	1 h
scgeneffit*	0.63	0.78	0.71	0.10	0.82	0.63	0.77	0.70	3 h
scmer	0.63	0.78	0.73	0.15	0.81	0.60	0.80	0.77	2 h
genebasis	0.62	0.78	0.74	0.17	0.81	0.55	0.88	0.97	3 h
persist*	0.60	0.73	0.64	0.04	0.82	0.62	0.90	0.98	6 h
asfs*	0.59	0.73	0.62	0.05	0.82	0.61	0.89	0.99	23 h
pca	0.59	0.73	0.70	0.17	0.78	0.49	0.82	0.79	16 sec
scpnmf	0.53	0.71	0.64	0.11	0.75	0.42	0.84	0.88	57 min
persistus	0.53	0.70	0.62	0.05	0.77	0.42	0.92	1.00	1 d
cosg*	0.30	0.27	0.40	0.01	0.60	0.15	0.97	0.99	3 sec
smash*	0.10	0.05	0.13	0.00	0.26	0.01	0.99	1.00	16 h
triku	0.06	0.03	0.03	0.00	0.13	0.08	0.60	1.00	4 min

* used cell type annotations for selection

Panel a of main text Figure 4

Fig. 4: Spapros outperforms classical selection strategies and state-of-the-art methods. a, Table showing mean performances of Spapros and other methods, based on 20 bootstrap samples for selecting 50 genes from the Madisson2020 lung dataset. Methods that use cell type information are annotated with a red star. [...]

2.5 The 10X protocol for targeted spatial experiments appears to just enrich target genes for sequencing after performing an untargeted capture. Is spapros still relevant for use in that scenario?

We are not familiar with a spatial 10X method that does enrichment after an untargeted capture. Potentially this refers to a combination of selecting a gene panel based on the untargeted method Visium and exploring the selected panel in high spatial resolution with the targeted method Xenium? This scenario is supported by Spapros: In our comparison (Fig. S9, previously S8) with the selected panel of the ISS study of Asp et al.⁶ we ran a Spapros selection on Visium data and scRNA-seq data that outperformed the selection of the publication that was also based on the two datasets. We describe this in the method section as:

For our comparison with an ISS panel we took the original gene set from Asp2019³ which contains 69 genes that were selected based on an scRNA-seq and an untargeted spatial transcriptomics data set. To generate a comparable selection with Spapros we selected 34 genes on the untargeted dataset and used these as prior knowledge selection [...] for a selection of 69 genes on the scRNA-seq data set.

We describe the performance comparison in the main text as follows:

[...] when comparing Spapros with manual feature selection based on literature markers, Spapros outperformed the manual list for both heart³ and lung³⁸ scenarios. The method surpasses manually selected probe sets and curated marker lists on cell type identification and variation recovery (Fig. S9b,c).

As Xenium is the refinement of the ISS technology we expect this approach to transfer well to Xenium, this is also expected for other targeted spatial methods.

Minor comments:

1. I don't see anywhere on the github on documentation pages that explicitly say the package is Python based.

We adjusted the "About"-text in github to "Python package for Probe set selection for targeted spatial transcriptomics."

Further we adjusted the first sentence of the documentation main page to: "Spapros is a python package that provides a pipeline for probe set selection and evaluation for targeted spatial transcriptomics data."

Reviewer #3:

Remarks to the Author:

The study reported in the manuscript entitled "Probe set selection for targeted spatial transcriptomics" by Kuemmerle et al., introduces a novel computational method for selecting a probe set to analyze spatial transcriptomics data from different tissues. This method combines unsupervised and supervised combinatorial selection features to account for cell type identification and gene expression variation within a cell type, which resolve its spatial and temporal location within a tissue. The authors compared their method to the existing methods aiming at targeting key features within tissue and described the advantages of Spapros over these methods. To ensure that Spapros produces credible results, the authors performed validation experiments using the in situ spatial method SCRINSHOT on two tissues: the lung and the heart. Specifically, Spapros was utilized to identify not only the lung's cellular subsets but also to account for expression variation and technical noises that emerged from the single cell data sets obtained from different studies. By utilizing their new probe set of 50 or 150 genes, the authors showed their new method's ability to capture the cell type landscape of the tissue while also measuring their expression variability in different locations, using the FOS gene as proof of concept in tracheal basal cells. Lastly, the method is well described, the code is well written, and the open-source software works fairly quickly with a friendly interface. Indeed, Spapros, a new spatial transcriptomics probe selection, fulfills the open and accessible source so that it can be adopted by the community around the world.

We thank the reviewer for the positive evaluation and appreciation also of benchmarks and accessibility, as well as for the comments below.

Spapros was evaluated by selecting probes from 6 different single cell datasets and utilizing 12 quality metrics to show its superior selection in cell type and expression variation recovery. Of note, Spapros relies heavily on single cell genomics analysis of tissues. Therefore, it might be biased toward abundant cell subsets and may be guided by false signals arising from cell dissociation. To overcome that, the authors utilized only the in situ-based SCRINSHOT assay for validation and showed FOS expression variation along the basal cell spatial organization. Is this an actual signal? Why do basal cells express FOS?

Indeed the selection is biased towards the single cell reference including the cell type clusters as defined by the user. However, in the design of Spapros this is considered as a feature by which the probe set selection can be flexibly tailored to the planned spatial experiment. To account for cell type proportion differences Spapros includes sampling strategies such that cells of rare cell types can be classified with the same accuracy as cell types that were enriched e.g. due to the dissociation protocol of scRNA-seq. Besides cell type proportion biases, the selection can be biased towards variation that could be technical scRNA-seq variation, as we alluded to in the discussion section.

We agree that it is an interesting question if the spatial FOS variation in basal cells is a biological or technical effect. With our SCRINSHOT experiments we showed that the signal of high FOS variation can be found in spatial assays. We **now further evaluated that this is not an artifact of SCRINSHOT** by measuring the protein level of an adjacent slide with immunofluorescence (IF, see answer to comment 3.4 below). There can be different reasons for the observed variation of FOS. We leave the investigation of the cellular mechanism of FOS expression to further studies and regard it as out-of-scope for our method paper.

Therefore there is a need to test the success rate of Spapros prediction on other in situ-based spatial datasets such as Slide-seq. This examination will reduce noise and technical issues associated with dissociation protocols used in scRNA-seq.

We agree that it is important to test the success rate of Spapros predictions on other in situ-based spatial datasets. We tested this now with two strategies: 1. showing that the signals identified in SCRINSHOT translate to other modalities and technologies, and 2. validating Spapros selections additionally on MERFISH:

We show that the reported intra-cell type variation in SCRINSHOT is highly correlated with an orthogonal protein measurement (IF). This provides a validation using a different molecular readout (see answer to comment 3.4).

We tried to validate our spatial signals in Visium data, however, due to the current low spatial resolution of Visium (and Slide-seq) and limitations of cell type deconvolution methods we couldn't confidently attribute spatial patterns of intra-cell type variation to specific cell types.

We have incorporated extensive evaluations on MERFISH and matched snRNA-seq data. Our results demonstrate that optimizations tailored to the matched reference yield superior performance on the spatial data. This is evident not only in our metrics related to cell type classification and variation recovery but also in the newly introduced metrics that gauge the capture of variation linked to spatial patterns and cell-cell interaction states (see answers to comments 3.1, 3.2, 3.3, and 1.1).

We believe our current evaluations using multiple technologies and orthogonal measurements offer a comprehensive assessment of Spapros' robustness and reliability.

Next, it is imperative to include basic guidelines to perform Spapros analysis, e.g., how many single-cell datasets should be used? Is it important to include at least one in situ hybridization-based dataset?

The selections are solely done on scRNA-seq references, no in situ hybridization-based datasets are needed. Spapros is flexible enough to leverage additional information if given though (e.g. from a Visium dataset as shown in our comparison with the ISS selection from Asp et al.⁶). We now extended our evaluations on scRNAseq from 2 to 12 datasets, showing broad applicability over different tissues. With our cross-dataset evaluation we showed that probesets selected on sufficiently large datasets translate well between each other. The number of required datasets is highly dependent on the captured variation within each dataset. We now added additional evaluations of selections on subsets that provide guidelines for the minimal number of cells and genes that are needed for robust selections (see answer to comment 3.7).

In addition, the authors claim that their method is broadly applicable to different tissues. However, I was not convinced that this method is robust enough to account for cell-cell interactions from different cellular lineages (e.g., epithelial-immune and fibroblast-immune interactions). In addition, a step-by-step validation of this algorithm by showing its specificity and success rate in a controlled environment (known interactions between cells) is missing from the paper.

Besides our quantitative evaluation that shows how much cell-cell interaction related variation is captured with a given probe set we use intermediate results of the evaluation to investigate specific interaction states of cell type pairs. We show that significant state variation related to interactions of similar and of different lineages are well captured with Spapros (see answer to comment 3.3. below)

Overall, Spapros provides an elegant solution method for choosing a targeted gene set for spatial transcriptomics by utilizing single-cell RNA-seq datasets. However, the method's robustness and validations are still missing and need more work to pinpoint the robustness of the technique in identifying cell-cell interaction signals within tissues as other non-supervised spatial transcriptomics methods.

We again thank the reviewer for the positive evaluation and the constructive comments. In our initial manuscript version, we have not delved deeply into the nuances of cell-cell interaction related variation. The reviewer's comments illuminated a crucial perspective for assessing our method. We concur that this dimension was pivotal and its inclusion has considerably enriched our manuscript by underscoring that our selections indeed target gene expression variation of significance in spatial measurements.

Specific comments:

Major comments:

3.1 Cell-cell interactions might change the gene expression of the interacting cells (changing the cell state) so that cells would have a different expression profile than their singlet constituents. How does the method deal with these changes, and how can the algorithm assess this? This is one of the most challenging issues that Spapros should deal with.

One rationale for targeting variation recovery in Spapros selections is that dissociated data variation represents patterns of spatial variation. So we should recover variation between cells due to their interaction patterns. To show that this is actually the case we investigated if the general variation recovery objective of Spapros leads to an increased coverage of cell-cell interaction correlated expression profiles. We selected probesets subsets of a MERFISH human brain dataset of 4k genes⁷ (selection on the single cell reference⁸) and applied the unbiased graph neural network approach NCEM⁹ to measure the number of significant gene to cell-cell interaction relationships. We find a clear correlation between variation recovery on the reference dataset and the cell-cell interaction recovery score on MERFISH. This shows that optimizing for variation recovery translates to capturing more CCI related state variation. We added the analysis as an additional main figure and expanded the main text results section accordingly:

To assess whether the performance differences in our benchmark translate to improved utility of Spapros to address biological questions, we investigated how well cellular interactions are recovered in different probe sets. Specifically, we compared the gene set selection methods on recovery of cell state variation associated with CCIs on a matched snRNA-seq and MERFISH brain dataset (see Methods). Overall, we find a high correlation between variation recovery and CCI recovery ($r = 0.85$), with Spapros among the top performers (Fig. 4e).

Panel e of main text Figure 4

Fig. 4: Spapros outperforms classical selection strategies and state-of-the-art methods. a, [...]. e, Correlation between variation recovery scores on dissociated data and CCI recovery on spatial data, using matched snRNA-seq and MERFISH data from the human brain. f, [...].

3.2 Related to the previous comment, the strength of this method should be in trying to understand the cellular program changes occurring due to cell-cell interactions in health and, most importantly, in disease, as many of the single cell datasets report these transcriptional changes but with no spatial context.

A central goal of measuring Spapros probe sets in spatial experiments is the investigation of cell-cell interactions and spatial modules/niches that are associated with the transcriptional variation that translates from scRNA-seq measurements to spatial measurements. In the answers to comments 3.1 and 3.3 we showed that Spapros optimizes for recovering transcriptional variation associated with cell-cell interactions and does so broadly by recovering CCIs of different cell type interaction pairs.

Spapros enables users that are interested in particular biological processes to target these processes while also selecting probes that enable users to get an overview of cellular heterogeneity within the tissue. Importantly the reference data set should be selected accordingly. To further increase the focus on specific cell-cell communication or disease-relevant genes the preselection feature of Spapros enables more fine tuning of the selection.

We clarified this in detail in the main text discussion section as follows:

While Spapros is a flexible pipeline that can be tuned to emphasize sub-types of a specific cell type, it inherently focuses on capturing the major sources of variation within a dataset. However, the strength of these signals does not always correspond to their relevance to a given scientific question. This can be particularly true in disease studies, where subtle transcriptional differences may be of greater interest than global sources of variation. In these cases, incorporating prior knowledge through Spapros' pre-selection feature can be especially beneficial. This approach enables users to specify genes to focus on the exploration of disease mechanisms and the identification of relevant spatial cell niches and their associated cell-cell interactions.

3.3 The authors should also address the possibility of two similar cell subsets (same kind) interacting with another cell type (such as T-B or T-DC interactions). Could Spapros identify this?

With our NCEM evaluation approach we measured for each gene if it is significantly related to cell-cell interactions of specific cell type pairs. With the highest scoring (Spapros) gene set ($n = 150$) of our analysis in 3.1 we capture significant CCI genes for most cell type pairs (84%). The top 16 CCI pairs (highest ratio of selected significant genes) include cell-cell interactions of various cell types interacting with the same/similar cell types and more distant types (see plot below; genes of the selected set are shown as stars), e.g.:

- same/similar cell types: glia (IOGC-IOGC), neurons (eL23IT-eL45IT)
- different cell types: excitatory and inhibitory neurons (iSST-eL6ITCAR3), glia-neuron (IOPC-eL5IT), immune-glia (IMGC-IOGC), ...

Further we find distinct CCIs for similar neuronal cell identities and a third cell type. These results show that Spapros gene sets broadly capture cell-cell interaction state related genes in an unbiased fashion. We added the volcano plots for the top 16 cell type pairs as a supplementary figure and extended the main text accordingly:

[...] Spapros gene sets capture cell type interaction pairs in an unbiased fashion: We detect interactions between all cell type lineages. Furthermore, CCIs detected by Spapros are also able to distinguish between similar cell subtypes. We find distinct CCIs for similar excitatory neuronal cell identities with a third cell type (e.g., SST+ inhibitory neurons or excitatory neurons in layers 4/5; interactions iSST-eL6ITCAR3 vs iSST-eL5IT, and eL45IT-eL23IT vs eL45IT- eL6ITCAR3; Fig. S12a).

Panel a of Extended Data Figure 12

Fig. S12: Cell-cell interaction and cell type recovery of gene set selections for MERFISH human brain data. **a**, Volcano plots of the 16 cell type interaction pairs with the highest number of significant genes affected by cell-cell interaction of the given cell type pair (based on Wald-tests of the NCEM model on MERFISH human brain data). Significant hits are shown for a 150 genes Spasros selection on snRNA-seq human brain data. Genes of the selected gene set are highlighted by star symbols. P-values of 0 were set to the minimal non-zero observed p-value of $\sim 10^{-16}$. **b**, [...].

3.4 The authors show FOS expression in tracheal basal cells. FOS expression may be due to low-quality single cell measurements and therefore represent technical noise expression. Could the authors utilize the protein levels to validate their results? And show a different gene with a similar variation in basal cells?

As the reviewer suggested, we validated the observed FOS signal on the protein level using newly generated immunofluorescence (IF) stainings of the same samples. Two additional adjacent slices of the tracheal sample were taken, one for IF of FOS and basal markers and one for an additional SCRINSHOT run of just the three genes FOS, KRT15, S100A2. We registered the epithelia of the two new slices based on expert annotations of landmarks and the path along the epithelium, and measured the mean FOS signal. We find a clear correlation between the increasing FOS signal in SCRINSHOT and IF. We report the validation analysis in a supplementary figure and extended the main text with:

[...] FOS exhibits up- and down-regulated regions along the epithelium orthogonal to the interior-to-exterior epithelial variation of KRT15 and S100A2 (Fig 3d,e). Investigating FOS variation along the airway epithelium with immunofluorescence stainings validated this finding (Fig. S8a).

Panel a of main Extended Data Figure 8

Fig. S8: Intra-cell type variation and validation with IF. a, Validation of the spatially variable FOS signal in tracheal basal cells. FOS expression of adjacent IF and SCRINSHOT samples are correlated along the registered annotated tracheal epithelium. **b,** [...].

In our SCRINSHOT panel there was no gene with high correlation to FOS as the panel only included 64 genes and was designed with reduced gene redundancy to capture more variation. However, we'd like to point out that we showed the additional genes KRT15 and S100A2 that exhibit interior to exterior variation in tracheal basal cells.

3.5 It will be important to show gene expression variation to other cell types from a different compartment. For example, expression variation in stromal or immune cells?

We agree that it is important to provide more evidence that Spapros selects genes that exhibit interesting spatial intra-cell type variation, especially focusing on cell types of other lineages besides epithelial cells. We leveraged the scRNA-seq lung data and our presented intralobar SCRINSHOT sample to screen for genes with high gradients in non-epithelial cell types. We found the strongest candidates in endothelial cells and macrophages: In ECs the genes IGFBP7 and RGCC are up-regulated in different spatial clusters of ECs. For macrophages we found that APOE is more up-regulated in the center part of local macrophage clusters while IFITM1 shows higher expression in macrophages that surround these clusters. We added an additional supplementary panel highlighting the spatial variation of these genes and referenced the results in the main text as follows:

[...] multiple other genes selected by Spapros exhibit spatial intra-cell type variation, such as anti-correlation between IGFBP7 (associated with larger vessels) and RGCC (capillary marker) in endothelial cells, as well as between APOE and IFITM1 in macrophages (Fig. S8b).

Panel b of main Extended Data Figure 8

Fig. S8: Intra-cell type variation and validation with IF. a, [...]. b, Spatial intra-cell type variation of genes in the intralobar SCRINSHOT lung sample.

3.6 The authors should state more clearly the limitations of their method. It would be important to know if Spapros may not work well in deciphering cell-cell interactions. It is important to expand on this and other limitations so that researchers who adopt the Spapros method are aware of possible biases or limitations.

While we discussed the main limitations of Spapros in the discussion section, we agree that a more fine grained description is needed, especially on the previously missing topic of selection bias in cell-cell interaction variable genes. With the results presented in comments 3.1 and 3.3 we found that Spapros indeed selects genes associated with cell-cell interactions in an unbiased manner, meaning that interactions of different cell type lineages and specific differences between similar interaction pairs are captured.

We believe the previously covered limitations updated with the new insights based on the reviewer's comments inspired analysis builds a comprehensive limitations description for our method. Here we list up all limitations discussed in our updated discussion section:

- there are various possible discrepancies between spatial and scRNA-seq data:
 - [...] equality assumption between spatial and scRNA-seq data is currently not strictly met and we find discrepancies between the modalities. [...] some genes are uncorrelated to the signal expected from scRNA-seq, possibly due to non-functional probes or individual tissue section anomalies.
 - [...] Different sample processing between scRNA-seq and targeted ST lead to the enrichment or exclusion of certain cell types and states.
 - based on the analysis for comment 3.7: [...] the number of genes needed to reach the same cell type classification robustness can be substantially higher in data from older, high-plex spatial technologies compared to the dissociated reference (Fig. S12b)
 - We also added a more detailed discussion about translation to spatial data, and the consideration of data quality and robustness constraints in an additional Supplementary Note 1: [...] for panel designs of current high-plex technologies like MERFISH and CosMx, robustness constraints used in methods like PERSIST and scPNMF can be beneficial, while for lower-plex solutions like Xenium and MERSCOPE, better translatability is expected.
- In specific situations Spapros is "too unsupervised":
 - Variation prioritized by Spapros might not be the most relevant variation for the given study: [...] Spapros [...] inherently focuses on capturing the major sources of variation within a dataset. However, the strength of these signals does not always correspond to their relevance to a given scientific question.
 - While Spapros optimizes for the selection of genes encoding state information of cell-cell interactions it might not capture the ones most important for the given scientific question. Instead prior knowledge might be important to capture these: [...] in disease studies, where subtle transcriptional differences may be of greater interest [...] Spapros' pre-selection feature [...] enables users to specify genes to focus on the exploration of disease mechanisms and the identification of relevant spatial cell niches and their associated cell-cell interactions.

- While recovery of dissociated variation is correlated with recovery of biologically relevant spatial variation, there is no guarantee that specific signals translate to spatial patterns or instead randomly distributed variation: (from Supplementary Note 1) [...] In a spatial context, such variation could manifest as either locally patterned signals (spatial variation) or randomly distributed variation over space.
- Spapros might select genes due to batch effect variation: [...] batch effects are still a challenge for Spapros and other probe selection methods. Especially selection methods that optimize for variation recovery will also select genes aligned to batch effect variation.
- Inclusion of variation recovery genes can reduce cell type separation: [...] including genes like FOS reduces the separation of cell types when clustering cells. Thus, such genes introduce an additional challenge for downstream analysis and should be excluded for cell type clustering.
- Spapros' probe design does not support all technologies yet: [...] probe design component supports a range of technologies, including SCRINSHOT, MERFISH, SeqFISH+ and HybISS [...]
- Spapros is only tested on transcriptomics datasets: [...] Further extensions to Spapros include experimental design for other modalities like spatial proteomic measurements (e.g. Codex)

3.7 The authors should provide a guideline on how many genes are necessary to identify most cell subsets and their expression variation within a tissue.

To properly address this question we investigated cell type classification performance on the matched mouse brain MERFISH and snRNA-seq datasets. Gene sets with different numbers of selected genes were selected on the single nucleus dataset. We find a significant difference between spatial and dissociated data in the trends of required genes to reach saturated cell type classification performance. While for single nucleus data 2-4 genes per cell type are sufficient for the highest classification performance, we find that 20-60 genes per cell type are needed for the spatial data to reach the same. It is important to note that it is questionable how far these results generalize to different technology, tissue, and experiment parameters. In our SCRINSHOT data we successfully identify cell types based on the scRNA-seq derived cell type classification rules that contain 2-5 genes per cell type (Fig 3a,b,S7). With our analysis we only scratch a more thorough comparison between technologies and tissues about how well the signal from sc/snRNA-seq translates to spatial data. A more extensive comparison is however out of scope of our manuscript. We included a supplementary figure panel on the cell type classification comparison with MERFISH and raise the limitation in the main text discussion:

[...] the number of genes needed to reach the same cell type classification robustness can be substantially higher in data from older, high-plex spatial technologies compared to the dissociated reference (Fig. S12b)

Panel b of main Extended Data Figure 12

Fig. S12: Cell-cell interaction and cell type recovery of gene set selections for MERFISH human brain data. a, [...]. b, Cell type classification accuracy on snRNA-seq and MERFISH human brain data of Spapros selections on snRNA-seq with different numbers of genes.

Further we discussed this spatial data quality aspect in our Supplementary Note 1:

[...] recent studies have shown significant variability in the quality of spatial measurements^{40,41,75}. These studies suggest that current high-plex solutions (MERFISH, CosMx, STARmap PLUS) exhibit fewer counts per gene and cell, and increased nonspecific signals for individual genes, compared to the higher quality, currently lower-plex methods (MERSCOPE, Xenium). Specifically, as demonstrated with our SCRINSHOT data, individual selected marker genes can effectively recover targeted cell types (Fig. 3a). This was similarly observed in Cook et al.⁴¹, where cell types within delicate spatial structures were recoverable with single marker genes using Xenium, but challenging with high-plex CosMx measurements. Wang et al.⁴⁰ also reported similar findings, with more cell types identified across multiple tissues using lower-plex Xenium compared to higher-plex CosMx. These studies

provide strong evidence that with the latest advancements in spatial technologies, there is improved translatability between scRNA-seq and spatial data. [...]

3.8 Is it possible to create combinatorial gene probes that are not unique but will combine a few genes to account for cellular programs such as consensus non-negative matrix factorization (cNMF)?

This is indeed an intriguing suggestion. Spapros currently works under the premise that users require unique gene probes and we thus filter out all potential probes that are shared by multiple genes. Thus, building non-unique probes that target multiple genes is out of scope for this publication.

We would further hypothesize that designing such probes would not always be possible for genes in the same pathway or that are involved in the same biological function. Probes are based on gene sequence, which finally refers to protein structure. While this structure is indicative of a protein's molecular function (e.g., kinases), we are not aware that this is also indicative of the pathways a gene or protein is involved in. Yet, building a non-unique probe that targets multiple genes in a pathway or cellular program requires that these probes are still unique to the genes in this pathway. This may be possible for pathways that involve complexes formed of related genes with similar sequence, but may be hard to get to work generally.

We added a sentence on this in the outlook: Further extensions to Spapros include [...] extended probe design schemes of combinatorial gene probes that target multiple genes contributing to the same cellular program.

List of citations:

1. Song, D., Li, K., Hemminger, Z., Wollman, R. & Li, J. J. scPNMF: sparse gene encoding of single cells to facilitate gene selection for targeted gene profiling. *Bioinformatics* **37**, i358–i366 (2021).
2. Covert, I. *et al.* Predictive and robust gene selection for spatial transcriptomics. *Nat. Commun.* **14**, 1–14 (2023).
3. Sikkema, L. *et al.* An integrated cell atlas of the human lung in health and disease. *bioRxiv* 2022.03.10.483747 (2022) doi:10.1101/2022.03.10.483747.
4. Eng, C.-H. L. *et al.* Transcriptome-scale super-resolved imaging in tissues by RNA seqFISH+. *Nature* **568**, 235–239 (2019).
5. Chen, K. H., Boettiger, A. N., Moffitt, J. R., Wang, S. & Zhuang, X. RNA imaging. Spatially resolved, highly multiplexed RNA profiling in single cells. *Science* **348**, aaa6090 (2015).
6. Asp, M. *et al.* A Spatiotemporal Organ-Wide Gene Expression and Cell Atlas of the Developing Human Heart. *Cell* **179**, 1647–1660.e19 (2019).
7. Fang, R. *et al.* Conservation and divergence of cortical cell organization in human and mouse revealed by MERFISH. *Science* **377**, 56–62 (2022).
8. Hodge, R. D. *et al.* Conserved cell types with divergent features in human versus mouse cortex. *Nature* **573**, 61–68 (2019).

9. Fischer, D. S., Schaar, A. C. & Theis, F. J. Modeling intercellular communication in tissues using spatial graphs of cells. *Nat. Biotechnol.* (2022) doi:10.1038/s41587-022-01467-z.

Response to the reviewers' comments after AIP of

NMETH-A49853B: "Probe set selection for targeted spatial transcriptomics"

We would like to thank the editor and the reviewers for their comments and suggestions, which we believe helped to improve this manuscript considerably. In the following we give **point-by-point answers (green)** to the questions and **comments (black)** and in parts **copy old parts of the manuscript text or specific panels (blue)** and **updated parts of the manuscript text or specific panels (red)**. We numbered the comments of the editor (0.x) and of the reviewers (1.x, 2.x, 3.x).

Dear Madhura

Thank you again for the acceptance in principle for publishing our manuscript "Probe set selection for targeted spatial transcriptomics". We were very happy about the decision. Please find below our answers to the remaining comments of reviewer 1. We wish to participate in transparent peer review.

We feel that we were able to take the feedback on board and address the raised issues of reviewer 1 in a revised manuscript and want to thank you and the reviewers again for the detailed review of our manuscript.

Best regards,
Fabian

Reviewers' comments:

Reviewer #1:

General remarks:

The authors clarified the novelty of their work, and it looks like the uniqueness of this work is still being an ensemble of multiple smaller steps, which remains as a concern. The authors added more benchmarking and validation datasets, which is appreciated.

We thank the reviewer for the appreciation of our novelty clarifications and benchmarking efforts of the revised manuscript as well as the critical evaluation and constructive comments below.

Major comments:

1.1 The updated manuscript seems to restrict the application of this method to a few sequencing-based technologies. This makes the context clearer. It would be appreciated if the authors can state more clearly about which technologies they recommend applying this method to (and which are not)

We appreciate the reviewer's suggestion for clarification regarding the application of our method to different sequencing-based technologies. We would like to emphasize that Spapros' gene panel selection is applicable to all imaging-based spatial transcriptomics (ST) methods that measure subsets of genes. On the other hand Spapros' probe design aspect is currently restricted to 4 technologies, which we highlighted

- in the abstract: Spapros enables [...] probe design for **currently 4 spatial technologies**, as a freely available Python package.

- the method description results section: This probe design component supports a range of technologies, including SCRINSHOT, MERFISH, SeqFISH+ and HybISS, and is easily extensible.
- and the method section: To design probes for a set of given genes, we developed three custom probe design pipelines for the spatial transcriptomics protocols: SCRINSHOT (HybISS), MERFISH and SeqFISH+.

While the gene panel selection can be applied to all imaging-based ST methods we find that the performance gain of Spapros compared to other selection methods is especially pronounced for small gene panels. Therefore the optimization is most relevant for methods with smaller gene panels. We added the following sentence to the discussion section: Our gene set selection approach is broadly applicable to all imaging-based ST methods that use gene subsets, with a pronounced advantage for those with smaller gene panels.

1.2 The authors showed an advantage of optimizing gene selection and probe design simultaneously and many genes selected by other methods does not make good targets because they cannot be well-probed. This point is well taken. The question is: is it feasible that people can filter-out these genes beforehand and only retain the ones that can be probed, and apply other methods?

Absolutely, Spapros' probe design pipeline can be used to generate genome-wide gene filters and apply them to other methods as well. We added a sentence that highlights the independent value of the probe design pipeline: [...] Spapros' probe design filter can be used independently of the gene set selection process, making it compatible with other selection methods.

1.3 While discussing cell type classification and variation preservation, the authors need to clarify under what circumstances each aspect is important. In many cases, even cell types cannot be well-defined.

The main factor to decide to focus more on cell type classification or variation preservation is the purpose of an experiment. If the scientific hypothesis is solely built on cell type proportion ratios, the detection of compositional niches, or occurrence of specific cell types then cell type classification should be weighted higher. We provide the recommendation in the results section that describes the Spapros method: To facilitate downstream analysis in studies that solely focus on detecting cell type frequencies, it may be of interest to select only genes for cell type recovery rather than detecting additional spatial signals. For this, we provide SpaprosCTo, which exclusively utilizes DE trees for selection. This contrasts with the standard approach, where the majority of selected genes originate from the PCA-based selection [...]

On the other hand if the variation of interest refers to subtle state changes more weight should be put on variation recovery. For exploratory studies we recommend to optimize for general variation recovery to not miss prominent intra-cell type variation. This refers to the default Spapros configuration. We agree with the reviewer that for certain spatial datasets the cell type classification turns out to be much harder compared to the single cell reference (e.g. due to segmentation issues in high cell density regions). If prior knowledge about these issues is available a stronger focus on cell type classification is also recommended. We now include this consideration in the discussion: [...] the equality assumption between spatial and scRNA-seq data is currently not strictly met and we find discrepancies between the modalities. [...] the number of genes needed to reach the same cell type classification robustness can be substantially higher in data from older, high-plex spatial technologies compared to the dissociated reference (Fig. S12b). This has been reported to improve with newer spatial technologies such as Xenium^{40,41}. [...]. When anticipating lower data quality, prioritizing cell type identification over variation recovery during gene selection (as implemented in SpaprosCTo) can enhance the experimental design's robustness.

1.4 The authors did not address the question regarding the requirement of clustering/annotation of the data for Spapros.

We thank the reviewer for specifying the question below such that we can extend our previous answer. As mentioned in the previous response, Spapros relies on clustered single-cell datasets to serve as reference for probe set selection. The probe sets will be targeted towards the clusters provided.

1.4.a The authors discussed that Spapros cannot deal with batch effects as a limitation. However, with only a single-batch dataset, one could hardly reach high cell type resolution (given the small cell number).

The datasets we used in our comparison have multiple batches. However, the batch effects are comparably low within each multi-batch dataset. Indeed, we test the robustness of our probe set selection pipeline to selections on different batches in a multi-batch dataset and on all batches together and find comparable performance (see Fig S6). In such datasets a high cell type resolution covering rare cell types can be achieved and batch variation does not affect the selection negatively. E.g. the Krasnow2021 lung dataset contains 46 annotated cell types. We agree that integrated atlases with stronger batch effects are additional valuable resources to achieve high cell type resolution. We clarified this limitation in the discussion section as follows:

Further improvements to probe set robustness can be derived from integrated reference atlases. [...] Yet, strong batch effects are still a challenge for Spapros and other probe selection methods. Especially selection methods that optimize for variation recovery will also select genes aligned to batch effect variation. While we showed that Spapros can select probe sets and project these across datasets, we did find that also dataset-specific variation (potentially due to batch effects) was captured. Users that choose to select probes using reference atlases that contain robust consensus cell type annotations such as the HLCA (ref), may want to consider testing whether selected probes are consistent across batches.

Panel b of Extended Data Figure 6

Fig. S6: Spapros selections show robust cross dataset performance. a, [...] b, Cross dataset evaluations of selections on the lung data sets and on the donor samples within each data set. Cell type cls. perform. is the average of the metrics cell type classification accuracy and percentage of captured cell types. Variability recovery is the average of the metrics coarse and fine clustering similarity, and neighborhood overlap.

1.4.b What about cases that cells are in transitioning/development/tumor that are hard to annotate? In developmental data, for example, progenitor cells are possibly in different stages to differentiate but shall we annotate all as progenitor cells?

We appreciate the reviewer's concern regarding the annotation of cells in transitional, developmental, or tumor states. However, Spapros does not mandate any particular granularity of annotation, or indeed an annotation at all. We thus regard giving any guidance on cell type annotation as out of scope for this manuscript. As long as distinct clustering exists, the exact naming or annotation of these clusters is not relevant to the algorithm. Clusters can be labeled generically, such as "cluster 1". The granularity of this clustering is entirely up to the user and should be targeted towards what the users would aim to see in the spatial transcriptomic dataset they are generating.

We should note that datasets containing developmental transition states, such as hematopoietic stem cells from bone marrow data, which were clustered into different progenitor states, were included in our benchmark. In these scenarios, Spapros demonstrated the highest performance compared to other methods, showcasing its capability to manage such complex data effectively.

We clarified the label issue in our method description as follows: **Exact labels for cell type or state clusters are not necessary; once a cluster is identified as interesting, it can be included in the cell type classification task with a generic label to ensure it is distinguishable from other cell types or states.**

Reviewer #3:

The revised version of the manuscript offers novel analysis and statistical confirmations and, as such, enhances the method's credibility for the selection of specific probes to in situ-based ST. The authors have addressed my main concerns and modified their conclusions accordingly.

Overall, this study provides an important guideline for spatial analysis probe selection, which could be of interest to the broad scientific community and may help uncover additional layers of complexity within cells found in tissues under homeostasis and disease.

We thank the reviewer for the positive feedback and the appreciation of our study's value to the scientific community.